



# Development of a multi-method chronology spanning the Last Glacial Interval from Orakei maar lake, Auckland, New Zealand

Leonie Peti[1], Kathryn E. Fitzsimmons[2], Jenni L. Hopkins[3], Andreas Nilsson[4], Toshiyuki Fujioka[5,^], David Fink[5], Charles Mifsud[5], Marcus Christl[6], Raimund Muscheler[2], Paul C. Augustinus[1]

5    1 School of Environment, The University of Auckland, New Zealand

2 Research Group for Terrestrial Palaeoclimates, Max Planck Institute for Chemistry, Mainz, Germany

3 School of Geography, Environment and Earth Sciences, Victoria University of Wellington, New Zealand

Department of Geology, Lund University, Lund, Sweden

Australian Nuclear Science and Technology Organisation (ANSTO), Lucas Heights, Australia

6 Laboratory of Ion Beam Physics, ETH Zurich, Switzerland

^ current address: Centro Nacional de Investigación sobre la Evolución Humana, Burgos, Spain

*Correspondence to*: Leonie Peti (lpet986@aucklanduni.ac.nz)

**Abstract.** Northern New Zealand is an important site for understanding Last Glacial Interval (LGI) paleoclimate dynamics, since it is influenced by both tropical and polar climate systems which have varied in relative strength and timing of associated

changes. The Auckland Volcanic Field maar lakes preserve these climatic influences on the regional paleoenvironment, as well as past volcanic eruptions, in their sedimentary infill. The sediment sequence infilling Orakei maar lake is continuous, laminated, high-resolution and provides a robust archive from which to investigate the dynamic nature of the northern New Zealand climate system over the LGI. Here we present the chronological framework for the Orakei maar sediment sequence. Our chronology was developed combining Bayesian age modelling of combined radiocarbon ages, tephrochronology of

known-age rhyolitic tephra marker layers, $^{40}$Ar/$^{39}$Ar-dated eruption age of a local basaltic volcano, luminescence dating (using post infrared-infrared stimulated luminescence, or pIR-IRSL), and the timing of the Laschamp paleomagnetic excursion. We also investigated the application of meteoric (cosmogenic) Beryllium-10 variability to improve the age-depth model by complementing relative paleointensity measurements. However, the results were apparently influenced by some unaccounted catchment process and unable to reach satisfactory interpretation, apart from confirming the presence of the Laschamp

excursion, and therefore the $^{10}$Be data are not used in the production of the final age model. We have integrated our absolute chronology with tuning of the relative paleointensity record of the Earth's magnetic field to a global reference curve (PISO-1500).

The maar-forming phreatomagmatic eruption of the Orakei maar is now dated to >130,120 yr (95% confidence range 128,665 to 131,560 yr). Our new chronology facilitates high-resolution paleoenvironmental reconstruction for northern New Zealand

spanning the last ca. 130,000 years for the first time as most NZ records that spall all or parts of the LGI are fragmentary, low-resolution and poorly dated. Providing this chronological framework for LGI climate events inferred from the Orakei sequence is of paramount importance in the context of identification of leads and lags in different components of the Southern Hemisphere climate system as well as identification of Northern Hemisphere climate signals.

## 1 Introduction

A better understanding of past climatic change events, and coeval responses of the terrestrial ecosystems, is fundamental for quantifying the nature, magnitude, rates and drivers of these changes. However, temporal and spatial uncertainties remain in our understanding of the generation and transmission of climate events in a global context (Alley et al., 2003; Broecker, 2003;

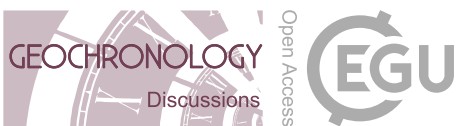

NGRIP Members, 2004; Seelos et al., 2009; Shanahan et al., 2009; Vandergoes et al., 2005). The incomplete picture of the spatial and temporal variability of terrestrial responses to climatic forcing globally is partly due to two factors: 1) a paucity of

complete, long and high-resolution terrestrial records of past climatic change in the Southern Hemisphere mid-latitudes, and 2) large chronological uncertainties associated with the few records that span the Last Glacial Interval (LGI; 115,000 – 12,000 BP). It is in these contexts that the maar lakes in the Auckland Volcanic Field (AVF) are significant as several of them contain laminated sediment sequences that span the LGI and beyond (e.g., Augustinus et al., 2012; Augustinus, 2007; Pepper et al., 2004; Stephens et al., 2012).

The Auckland Volcanic Field (AVF) is located in the City of Auckland at the top of the North Island of New Zealand's (Fig. 1) and consists of 53 basaltic volcanic centres that erupted since ca. 200,000 yr ago (e.g., Lindsay et al., 2011; Leonard et al., 2017). Of these 53 AVF volcanoes, 13 centres are extant or paleo maar crater lakes, ranging from ca. 200,000 yr to ca. 100,000 yr in age (Augustinus, 2007, 2016). Seven of the maar lake sequences have been drilled to date and all contain well-laminated lacustrine sediment intercalated with tephra from a range of AVF and other volcanic centres in the Taupo Volcanic Zone

(TVZ). This study focuses on Orakei Basin, an AVF paleo maar crater lake currently connected to the sea (Fig. 1). The sediment record from the Orakei maar paleolake is of unprecedented quality, length, resolution, and completeness in the context of the terrestrial south-west (SW) Pacific. The Orakei archive has the potential to facilitate reconstruction of the most detailed picture yet of past climatic changes in this part of the Southern Hemisphere, a region bridging the westerly-dominated mid-latitudes and the subtropics (Peti and Augustinus, 2019).

This study focusses on the integration of a range of absolute dating techniques (tephrochronology, radiocarbon, luminescence) with correlative dating (tuning of paleomagnetic field variations established by the relative paleointensity and meteoric [10]Be) for the development of the Orakei maar lake sediment sequence age-depth model.

Detailed tephrochronology is possible since several volcanic centres from the Taupo Volcanic Zone (TVZ), North Island, New Zealand, frequently ejected large volumes of rhyolitic material throughout the late Quaternary, repeatedly mantling the

Auckland region with volcanic ash providing isochronous tephra marker layers (e.g., Shane and Hoverd, 2002a; Molloy et al., 2009; Zawalna-Geer et al., 2016) in the AVF maar lake sequences (Fig. 1). Radiocarbon dating was used to refine the tephrochronology-based age estimates of the Orakei sequence, since it is a well-established technique for dating organic macrofossil samples younger than ca. 50,000 years (Bronk Ramsey, 2008). Luminescence dating, a technique which determines the time elapsed since crystalline minerals such as quartz and feldspar were last exposed to sunlight (Aitken, 1990),

was also applied. In this case, feldspar was used for luminescence dating, since New Zealand's relatively young geological history precludes effective dating of quartz due to reduced sensitisation of the luminescence signal (Almond et al., 2007). Feldspar is common in the Orakei sediments, its comparatively brighter infrared-stimulated luminescence (IRSL) signal was used to date deposition of the lake sediment. The most stable known signal from feldspar, known as post-infrared infrared-stimulated luminescence (pIR-IRSL), was used for the Orakei sediments >50,000 yr since it was recently demonstrated to be

effective for dating lacustrine feldspars (Roberts et al., 2018).

Synchronous environmental changes observed in multiple proxies allow the transfer of chronological information from one record to another. For example, correlation of changes preserved in polar ice cores and sediment archives can be made on the basis of fluctuations in geomagnetic field intensity (reconstructed from paleomagnetic measurements such as the ratio of natural to anhysteretic remanent magnetisation; relative paleointensity) and cosmogenic [10]Be atmospheric nuclide production

(Stockhecke et al., 2014). With regard to terrestrial sediment archives, [10]Be-inferred dating is a relatively recent approach and its utility for chronology development for sediment records beyond ca. 50,000 years (limit of radiocarbon dating) remains challenging due to overprinting by local catchment processes may which disturb the [10]Be inventory based solely on paleointensity changes (Carcaillet et al., 2004). Paleomagnetic measurements down core can also be used to identify magnetic excursions of known age, for integration into an absolute age model. Here we apply and assess the utility of each of these

approaches in the context of the development of the Orakei maar lake sediment sequence chronology.





A preliminary age model had been presented based on rhyolitic tephra marker layers in Peti and Augustinus (2019). In comparison to the earlier model, this study re-evaluates the tephra identification and adds radiocarbon and luminescence dates, as well as paleomagnetic measurements and variations in [10]Be production. This study develops a robust chronological framework for the long lacustrine sediment sequence retrieved from Orakei maar (Peti and Augustinus, 2019). A robust

independent chronology for this record will significantly improve temporal constraints on regional of paleoclimatic and – environmental reconstructions from the New Zealand sector of the SW Pacific, thereby enhancing our ability to identify leads and lags in climate teleconnections between the poles and tropics.

## 2 Regional setting

Orakei maar lies adjacent to the sea at the mouth of the Waitematā Harbour where it joins the Hauraki Gulf through the Tamaki
Strait (Fig. 1). The Orakei maar was formed as the result of a highly explosive, phreatomagmatic eruption ($126,150 \pm 3,320$ yr; Hopkins et al. (2017)). Since then it has hosted a lake that persisted for most of its history protected by a ca. 50 m-high tuff ring (Peti and Augustinus, 2019).

Sedimentation in the Orakei maar lake was dominated by autochthonous biogenic processes with contributions from distal aeolian transport and mass flows from erosion of the steep crater rim (Fig. 1d), since the catchment was mostly constrained by
the crater rim (Fig. 1d). A short period of fluvial sedimentation interrupted endorheic lacustrine conditions (Peti and Augustinus, 2019). Postglacial sea level rise breached the crater rim abruptly at ca. 9,000 cal yr BP, after which 18.6 m of estuarine muds were rapidly deposited, controlled by tidal inflow given Orakei Basin's proximity to the sea (Fig. 1; Hayward et al., 2008; Peti and Augustinus, 2019). Orakei Basin is presently connected to the sea by artificially-controlled inflow and outflow following the construction of a railway embankment in the 1920s (Hayward and Hayward, 1999) and is surrounded
by residential areas (Fig. 1c, d).

Orakei maar and the Auckland area is dominated by a warm temperate to subtropical climate (Hessell, 1988) with a mean annual temperature of 15°C. The mean annual rainfall is 1000-1400 mm, with drier summers and wetter winters (Hessell, 1988; Hurnard, 1979). On a larger scale, New Zealand experiences a spatially variable climate dependent on synoptic-scale pressure systems, wind regimes, latitude, topography and marine proximity. These dynamic climate subsystems likely have
expanded and contracted through time, so influencing the long-term climate of the Orakei area. Kidson (2000) described six regional climate zones in New Zealand from north to south, which experience very different climate parameters. Persistent mid-latitude westerly circulation brings moisture laden air masses to much of New Zealand (Hessell, 1988), although the northern parts of the archipelago, including the Orakei area, are also seasonally-influenced by the subtropical anticyclonic belt (Alloway et al., 2007; Barrell et al., 2013). The modern landscape and vegetation of Auckland have been heavily modified by
human activity (Shane and Sandiford, 2003) following Maori arrival in the late 13[th] century (Newnham et al., 2018).

## 3 Material and Methods

The Orakei sediment sequence is a composite of several overlapping cores (Peti and Augustinus, 2019). Two overlapping cores with 50 cm vertical offset were retrieved from Orakei Basin in 2016 using wireline drilling in 1 m-length sections. The sediment sections collected in 2016 were combined with a shorter core collected in 2007 to produce a complete composite
stratigraphy as described in Peti and Augustinus (2019). The Orakei sediment sequence consists of 14 sediment facies units overlying the Waitemata sandstone basement and basaltic volcanic ejecta from the maar-forming eruption, and is summarised in Fig. 2 (Peti and Augustinus, 2019). The Orakei maar core chronology presented here was established based on the event-corrected depth (ECD; Peti and Augustinus, 2019), in which all visible events of instantaneous deposition (tephra layers and mass movement deposits) have been removed.





### 3.1 Tephrochronology

Tephra identification was undertaken by major element analysis of glass shards via electron microprobe analyses (EMPA) following standard protocols as described in Peti et al. (2019). EMPA were performed by a JEOL JXA 8230 Superprobe at Victoria University of Wellington. A minimum of 10 individual glass shards in carbon-coated epoxy mounts were analysed using wavelength diffraction spectroscopy (WDS) with a static defocused beam at 10 µm and 8 nA. Major element oxide concentrations were determined using the ZAF[1] correction method and normalised to 100% to account for variable degrees of post-eruption hydration (Shane, 2000). Compositional data, such as major element percentages, are non-normally distributed and highly constrained by the closed-sum-effect (Aitchison, 1986). For this reason, we transformed the major element oxide concentrations using the centralised-log-ratio transformation (clr in the R package "compositions 1.40-2"; van den Boogaart et al. (2018)) prior to principal component analysis (PCA) using the prcomp command in the "stats" package as part of R version 3.5.3 (R Core Team, 2020). Additional details are included in Appendix A.

### 3.2 Radiocarbon dating

macrofossils (twigs and bark) were carefully extracted for radiocarbon dating, rinsed with de-ionized water and dried at 40°C. 28 bulk sediment samples were also extracted from 1 cm-thick slices, dried at 40°C and thoroughly homogenised. Anomalous old radiocarbon ages in lacustrine sediment samples can arise due to a freshwater reservoir effect, resulting from water rich in dissolved ancient calcium carbonates (Philippsen, 2013). Marine reservoir age corrections are routinely addressed in the marine realm but more difficult to assess in lake basins due to their different sizes, variable regional lithologies, depths and movement of water masses (Philippsen, 2013). We made efforts to restrict radiocarbon dating to short-lived terrestrial material, but the scarcity of macrofossils often obviated this approach so that we also used bulk sediment $^{14}$C dates to increase the age resolution at the risk of obtaining anomalous old ages via incorporating old carbon. To determine facies-dependent freshwater reservoir-effect corrections, we compared paired radiocarbon ages from terrestrial macrofossils and bulk sediment taken from the same depth. For facies unit 4, we selected the age of the Okareka tephra layer instead of a macrofossil and sediment sample OZX348, and samples OZX341 (wood) and OZX347 (bulk sediment) for facies 8 following the recommendations of Philippsen (2013).

Radiocarbon dating of organic macrofossils and bulk sediment was conducted at the Australian Nuclear Science and Technology Organisation (ANSTO) (Fink et al., 2004), apart from one macrofossil sample at the Radiocarbon Dating Laboratory (Department of Geology, Lund University) and one published date (NZA28865) from the Rafter Radiocarbon Laboratory (GNS, New Zealand). Samples for Accelerator Mass Spectrometry (AMS) measurement were pre-treated with 2M HCl to remove carbonate, 0.5-4% NaOH to remove humic and fulvic acids, followed by 2M HCl to remove absorbed $CO_2$ from the atmosphere during the preparation process. After drying, the samples were combusted, graphitised and analysed at the respective AMS-facilities. Ages were reservoir corrected and calibrated using the SHCal13 calibration curve (Hogg et al., 2013) as part of the age-depth modelling process (see section 3.6.1).

### 3.3 Luminescence dating

Luminescence dating of feldspar from ten samples was undertaken for the Orakei core. The fine-grained fraction was targeted in order to maximize the percentage component of non-volcanic, distally-sourced dust, which is more likely to yield a reliable luminescence signal. The sediment cores were stored wrapped in light-proof plastic at 4˚C until opened and subsampled for luminescence dating under subdued orange light at the University of Auckland (New Zealand), taking care to only sample the innermost parts of the core for dating. Samples were taken to ensure maximal spread over the >45,000 year interval of the

---

[1] The ZAF correction method refers to the consideration of the atomic number (Z), absorption of X-rays (A), and the fluorescence (F).





Orakei sequence, based on a preliminary age model which extrapolates the sedimentation rate between the two oldest tephra layers of known age (Tahuna and Rotoehu).

Luminescence dating sample pre-treatment for polymineral fine grains (4-11 µm) was undertaken at the Max Planck Institute for Chemistry (Mainz, Germany) following published methods (Mauz and Lang, 2004) and involved HCl and $H_2O_2$ digestion, sieving and settling. Additionally, coarse-grained K-feldspar (90-115 µm) was extracted from sample L20 (A0113) by density separation using lithium heterotungstate for comparison with the fine-grained signal. The other samples yielded insufficient coarse-grained material for dating. Samples were loaded using a pipette onto 10-15, 1 cm diameter stainless steel discs

depending on the amount of material remaining after chemical preparation. Additional details are included in Appendix B. The equivalent dose ($D_e$) was measured using the post-infrared infrared (pIR-IRSL$_{290}$) protocol, stimulated at 290˚C to minimize the potential for feldspar signal fading (Buylaert et al., 2012; Thiel et al., 2011, 2014). Measurements were undertaken using an automated Risø TL-DA-20 reader using infrared (IR) LEDs for illumination, with the resulting signal detected by an EMI 0235QA photomultiplier tube with a D410 filter for feldspar (Bøtter-Jensen, 1997; Bøtter-Jensen et al.,

1999). Irradiation was provided by a calibrated $^{90}Sr/^{90}Y$ beta source (Bøtter-Jensen et al., 2000). An internal alpha contribution of 0.12 was assumed for the samples, in order to account for the lower luminescence efficiency of alpha radiation relative to the beta and gamma components (Rees-Jones, 1995; Rees-Jones and Tite, 1997). A higher stimulation temperature was chosen based on equivocal results of the preheat plateau test which showed substantial inter-aliquot scatter irrespective of measurement temperature (Fig. B1). The accepted aliquots of all samples yielded either normal distributions, which were

typically very wide, or skewed distributions indicative of incomplete bleaching. In the first case, equivalent dose values were calculated using the Central Age Model (CAM) (Galbraith et al., 1999). In the latter, equivalent dose was calculated using the Minimum Age Model (MAM) (Olley et al., 2004) (Figure B2). Further details can be found in Appendix B.

Sub-samples were collected for dose rate analysis from the same depth in the core as each dating sample. These were dried and homogenised. A weighed part of this sample was ashed at 950°C and K concentrations were obtained by XRF at the

University of Waikato, Hamilton (New Zealand, see Appendix B for further details). The other weighed part was treated with $HNO_3$ and HF at ANSTO, from which U and Th concentrations were obtained by inductively-coupled plasma mass spectrometry (ICP-MS) at the University of Auckland (see Appendix B for further details). The measured activities of the radioactive elements K, Th, and U were converted to dose rates using published factors (Adamiec and Aitken, 1998; Guerin et al., 2011). Dose-rate attenuation was estimated using published values (Mejdahl, 1979) and the sediment moisture content.

Water content was estimated assuming saturation of the lacustrine sediments, taking into account water loss in the laboratory subsequent to core extraction (Lowick and Preusser, 2009). Substantial uncertainties in water content of 20 ± 10% were included in order to mitigate potential inaccuracies in the dose rate calculations due to desiccation. Cosmic-ray dose rates were calculated based on sediment density estimates, altitude, latitude, longitude, and depth, following Prescott and Hutton (1994). Cosmic-ray dose rate values were negligible (<0.01 Gy/ka) due to the depths of the samples within the lake sediments.

**3.4 Paleomagnetism**

Sediments below ~40 m depth (representing ages > ca. 35,000 cal yr BP) were used for paleomagnetic measurements. The upper (younger) interval of the Orakei sequence was not used for paleomagnetism as it can be reliably dated with radiocarbon and we note that the presence of numerous basaltic tephra layers can obscure a reliable paleomagnetic signal (Nilsson et al., 2011). This is a problem especially around the age of the Mono Lake Excursion, which correlates with a flare-up of the basaltic

volcanoes of the AVF around 30,000 cal yr BP (Molloy et al., 2009). Measurements of the Orakei maar sediment Natural Remanent Magnetisation (NRM), Anhysteretic Remanent Magnetisation (ARM) and Saturation isothermal remanent magnetisation (SIRM) were conducted using a 2G-Enterprises model 760-R SQUID magnetometer equipped with an automatic three-axis alternating field (AF) demagnetisation system at Lund University (Sweden). Oriented standard (volume = 7 cm$^3$) plastic paleomagnetic sampling cubes were used, with 48 samples treated as pilot samples and progressively demagnetised in





5 mT increments from 0 to 40 mT, and in 10 mT increments up to a maximum AF of 80 mT. The remaining samples were demagnetised in increments of 5 mT up to 30 mT, and in increments of 20 mT from 40 mT to a maximum AF field of 80 mT. Preliminary analyses on the pilot samples indicated that a stable single component magnetisation, demagnetising towards the origin, could usually be isolated between 15 mT and 80 mT (see Appendix C). This interval was therefore used to isolate the characteristic remanent magnetisation (ChRM) using principal component analysis (Kirschvink, 1980). Samples with

Maximum Angular Deviations MAD >15, mostly associated with weak NRMs, were rejected.

ARMs were induced in a peak AF of 80 mT with a DC bias field of 0.05 mT. The ARMs were measured and AF demagnetised in increments of 5 mT from 0 to 30 mT and in increments of 20 mT from 40 to 80 mT in the pilot samples and in increments of 10 mT from 20 to 40 mT in all remaining samples. SIRMs were induced with a DC field of 1 T using a Redcliffe 700 BSM pulse magnetizer. Volume specific magnetic susceptibility ($\kappa$) was measured using a AGICO MFK1-FA Kappabridge. Relative

paleointensities (RPIs) were estimated by normalising the $NRM_{20mT}$ with the magnetic concentration dependent $ARM_{20mT}$. The application of the Orakei RPI for chronology development is described in section 4.7.

### 3.5 Beryllium

Meteoric (i.e., atmospheric), cosmogenic $^{10}Be$, once produced in the atmosphere via nuclear reactions of cosmic ray particles with nuclei such as nitrogen and oxygen, readily attaches to aerosols and dust, and with a short residency time of ~1 yr, is

deposited on the Earth surface mainly via precipitation (Willenbring and von Blanckenburg, 2010). In open depositional environments (i.e., deep-sea and polar-ice cores), to a first order, variability in $^{10}Be$ concentrations can be used as a proxy of past geomagnetic field variation (e.g., Frank et al., 1997; Muscheler et al., 2005). In terrestrial settings, such as lakes, catchment processes and lithology (i.e., erosion, soil geochemistry, aeolian input, grain size) complicate $^{10}Be$ provenance, delivery and accumulation (Nilsson et al., 2011). To overcome such complexity, one approach is to normalise $^{10}Be$ measurements to an

element or isotope with similar geochemical properties, such as $^{9}Be$, where it has been demonstrated that authigenic $^{10}Be/^{9}Be$ ratio can compensate effects of catchment processes and grain size variations (e.g., Bourles et al., 1989; Wittmann et al., 2012). Although the Orakei catchment is small, grain size and lithology variability throughout the core is problematic due to in-wash events of clastic grains. In this study, we extract both authigenic $^{10}Be$ and $^{9}Be$ to compensate the effect of grain size and possible catchment process variation. Meteoric $^{10}Be$ and authigenic $^{9}Be$ was measured in order to detect variations in

cosmogenic nuclide production rates down-core to supplement the paleomagnetic measurements (section 3.4).

Laminated sediment intervals below ~40 m-depth in the Orakei sediment sequence, which cannot be dated with radiocarbon, were targeted for cosmogenic $^{10}Be$ measurements with a focus on the lower half of the sequence.

samples were processed for AMS measurements at ANSTO. Sample preparation (detailed in Appendix D) followed White et al. (2019) and involved 1-cm thick core slices (representing ca. 25-year duration; average sedimentation accumulation rate

(SAR) of 0.04 cm/yr) from which 0.5 to 1 g of dried, homogenised sediment was treated with 6M HCl for 2 hours to leach out the authigenic $^{10}Be$ and $^{9}Be$. A sub-sample of each leachate was extracted to provide a HCl-extractable $^{9}Be$ concentration (White et al., 2019) as measured on an Agilent 7700 ICP-MS at the Mass Spectrometry Equipment Centre (University of Auckland (UoA), New Zealand). $^{9}Be$ spike, derived from $^{10}Be$-free beryl mineral, was added to the remainder of the leachate prior to treatment with HF, $HClO_4$ to remove silicon and boron complexes. Beryllium was separated from other cations by

passage through anion and cation exchange columns, converted to $Be(OH)_2$, oxidised to BeO at 800°C and finally mixed with Nb powder. AMS $^{10}Be/^{9}Be$ measurement of these 59 samples were performed at the Centre for Accelerator Science, ANSTO (Australia) on the 6MV SIRIUS accelerator (Wilcken et al., 2019). The measured $^{10}Be/^{9}Be$ values were normalised to either the 2007-KN5-2 (nominal ratio of 8.558 x $10^{-12}$) or the 2007-KN5-3 (6.320 x $10^{-12}$) standard (Nishiizumi et al., 2007). Full chemistry procedural blank $^{10}Be/^{9}Be$ ratios range from 1.1-13.7 x $10^{-15}$ and are ≤~1% of the measured sample ratios.

Ten additional samples were prepared at Lund University for a high-resolution study of cosmogenic $^{10}Be$ variation during the Laschamp Excursion. Each sample consists of a 10-cm interval, representing ca. 250-year duration (SAR = ~0.04 cm/yr).





Samples were weighed, homogenised, spiked with $^9$Be and leached with $H_2O_2$, HCl, and $HNO_3$. Separation and precipitation of Be(OH)$_2$ was achieved by treatment with ammonia solution and NaOH for pH adjustment and followed by oxidation to BeO at 600°C (See details in Appendix D). AMS $^{10}$Be/$^9$Be measurements of these 10 samples were performed at ETH

(Eidgenössische Technische Hochschule) Zurich. Authigenic $^9$Be was not analysed for these ten samples and was considered negligible compared to the $^9$Be spike mass following measurements at UoA and ANSTO (see above). The measured $^{10}$Be/$^9$Be ratios were normalised to the ETH Zurich in-house standard material S2007N, which has a nominal $^{10}$Be/$^9$Be ratio of 28.1 x $10^{-12}$ (Christl et al., 2013) and is consistent with the 2007-KN$^{10}$Be/$^9$Be standards used at ANSTO. All measured $^{10}$Be concentrations were several orders of magnitude higher than the background of full chemistry procedural blanks, so that blank

corrections were negligible.

### 3.6 Correlation of magnetic relative paleointensity for relative dating

We applied relative dating using magnetic relative paleointensity (RPI) to Orakei maar lake sediments older than ca. 45,000 years to supplement the absolute age information provided by luminescence (pIR-IRSL) dating (see section 4.3). As highlighted by Blaauw (2012), it is crucial to avoid circularity in tuning climate proxies (such as pollen percentages, isotopes

or elemental compositions) based on assumed synchroneity, when the presence or absence of this possible synchroneity is actually an overarching study objective. Hence, we make use of the RPI of the Earth magnetic field strength which is considered to be a globally synchronous signal over millennial timescales (e.g., Channell et al., 2000; Laj et al., 2004) unlike climate signals. This approach was successful in Lake Van (Turkey) where several RPI minima were aligned to provide a chronologic framework (Stockhecke et al., 2014).

We use a dynamic programming algorithm to align the Orakei RPI with the Virtual axial dipole moment (VADM) reference curve from the marine PISO-1500 stack (Channell et al., 2009). Dynamic time warping (DTW) uses generalized dynamic programming, in which a complex problem is divided into smaller problems and their solutions are stored for later use. DTW aligns time series datasets through generalized dynamic programming and has been used for paleomagnetic data by Hay et al. (2019) and Hagen et al. (2020). The DTW algorithm calculates every possible match between the reference and query time

series at every strata (data point), producing a cost matrix. This cost matrix quantifies the Euclidean distance between each point in the query time series and each point in the reference time series. Under step-pattern constraints, an optimal alignment path is then found through the cost matrix which represents the optimal alignment path between both time series (Hagen et al., 2020). The step pattern only allows two directions in the path. This prevents the creation of age inversions, and heavily penalises alignments between points with very different indices, so that major jumps and/or hiatuses are very unlikely. The

DTW approach relies on (mostly) complete time series with a substantial component that overlaps between the reference and query time series (Hay et al., 2019).

### 3.7 Bacon age model development

The final age-depth model for the entire Orakei sediment sequence was produced using the Bacon package (Bayesian Accumulation model, using rbacon 2.4.2 (Blaauw and Christen, 2011, 2020) with a prior accumulation rate of 15 yr/cm

(assumed from a maximum core age of ca. 130,000 yr and a core length of 8000 cm, i.e., ~16.25 yr/cm). Prior accumulation shape, memory strength and memory mean were suggested by the Bacon package following Blaauw and Christen (2011) based on the age data provided (see Table 1). The age-depth model is further constrained by the provided rhyolitic tephra, radiocarbon and luminescence ages from the core (Tables 2, 3 and 4 respectively), the U/Th-age of the Laschamp Excursion (41,100 ± 350 (1 σ) years BP; Lascu et al., 2016), the Ar/Ar-age of the locally-sourced basaltic Mt Albert eruption (119,200 ± 2,800 (1 σ)

years; Leonard et al., 2017), and the dated tuning points obtained from matching the Orakei paleomagnetic data to the PISO-1500 global reference curve (see section 3.6, Table 5). Reservoir correction and calibration with the Southern Hemisphere calibration curve SHCal13 (Hogg et al., 2013) of radiocarbon ages was conducted by Bacon as part of the age modelling





procedure (Blaauw and Christen, 2020). A Student's t-distribution, essentially a symmetric distribution with longer tails than a typical Gaussian distribution, was used for the radiocarbon ages since it is more robust against outliers (Blaauw, 2010;

Blaauw and Christen, 2011; Christen and Pérez E, 2009). Gaussian distributions were assumed for tephra and pIR-IRSL ages, the Laschamp Excursion age and tuning point ages.

## 4 Results

### 4.1 Tephrochronology

The identification of the rhyolitic tephra layers follows Peti et al. (2019) with the addition of the Rotorua tephra and is

demonstrated in Figure 3 using multivariate statistics. PCA was first performed on published tephra EMPA data (black and grey symbols in Fig. 3), which serve as reference tephra in this study. The same scaling and rotation of the reference PCA were then applied to the unknown compositional data and the tephra layers were thus assigned to their source eruptive events through multivariate similarity (Fig. 3). The identification of tephra layers and the correlation to their eruption events is supported by the geochemical composition of the glass shards and stratigraphic position in relation to the AVF framework

(Hopkins et al., 2015; Molloy et al., 2009). The resulting identifications then allowed the direct transfer of published ages (or adjusted ages in this study) onto the Orakei sediment sequence. Further details can be found Appendix A.

Tephra sample T-74 could not be clearly identified solely from its major oxide geochemistry (Fig. 3, A1), since many layers show similar geochemical composition. However, its stratigraphic position and substantial thickness (>30 cm) suggest that this layer is the Rotoehu tephra. Tephra sample T-73 displays the same geochemistry as the Tahuna tephra, an interpretation

supported by its stratigraphic position (Figs. 3, A1). Tephra sample T-71 can confidently be assigned to the eruption pair of Hauparu and Maketu (both from Okataina Volcanic Centre (OVC)), which occurred within 300 years of each other (Molloy et al., 2009) and show similar major oxide compositions (Fig. 3). Given the similarity between Hauparu and Maketu reference major oxide data (Fig. 3), we cannot confirm from the tephra geochemistry whether T-71 represents the Maketu or Hauparu eruptions. Since only one layer has been found, we assign sample T-71 to Maketu given that this was the larger eruption with

more widespread and voluminous ashfall (estimated to be 15 km$^3$ for Maketu compared to 10 km$^3$ for Hauparu by (Froggatt and Lowe, 1990)). From a chronological perspective, this uncertainty in identification is irrelevant, since the two tephra are essentially the same age (Hauparu dates to $36,000 \pm 1,000 - 2,000$ years and Maketu to $36,300 \pm 1,000 - 2,000$ years; ages and generic error ranges of 1,000 to 2,000 years for rhyolitic tephra older than 21,000 years from Molloy et al. (2009)).

The geochemical compositions of tephra T-02 (25 mm-thick) and T-04 (5 mm-thick) are very similar (Fig. 3, A1). They are

positioned between the Okareka and Maketu tephras (above T-02 and below T-04). Stratigraphically-feasible candidates include the Te Rere, Kawakawa/Oruanui tephra (KOT), Poihipi, and Okaia tephra (Lowe et al., 2013). Te Rere and Poihipi can be rejected, since neither T-02 or T-04 show geochemical overlap with reference data for the products of these two eruptions (Fig. 3, A1). Stratigraphic ordering suggests that T-02 may be the younger KOT, whilst T-04 may correlate to the older Okaia tephra. However, similar depth in the stratigraphic framework developed for the Orakei sequence (Peti and

Augustinus, 2019) and the position of T-04 between several basaltic AVF tephra layers (Peti et al., 2019) suggests an identification of sample T-04 as the KOT as identified by (Molloy et al., 2009). This would imply a previously unknown eruption as the source of the thicker T-02 layer (25 mm thick), which is unlikely given the well-studied eruptive history of the Taupo Volcanic Zone (e.g., Lowe et al., 2013). Layer T-02 is well preserved, fining upwards and shows no signs of reworking (Peti et al., 2019). Its thickness supports identification as KOT, since the Kawakawa/Oruanui eruption was much larger than

Okaia (Froggatt and Lowe, 1990) and thus a thicker deposit would be expected in the sediment sequence. Most importantly, the glass shards of both samples show very different morphology in backscatter images (Fig. A2). Shards in T-02 are very large, cuspate, and bubbly with preserved thin bubble walls signifying a very large, explosive eruption. On the other hand, small, blocky shards in T-04, suggest a smaller, less explosive eruption (Heiken, 1972; Liu et al., 2015). This observation links





sample T-02 to the Kawakawa/Oruanui super-eruption and out of stratigraphic necessity T-04 to the smaller Okaia eruption.

It follows that Molloy et al. (2009) may have mis-correlated the apparent KOT in light of very similar geochemical compositions between sample and reference data and following the same assumption that the larger eruption should have been more easily preserved over the smaller eruption. The true KOT had been missed in a core break in the 2007 Orakei sequence. Other known major tephra, including Rerewhakaaitu, Te Rere and Poihipi, were not observed as macroscopic event layers within the Orakei sediment sequence in this study, although they may be present as cryptotephra. Samples T-01 and T-15 can

be clearly identified as the Okareka and Rotorua tephras based on geochemistry (Fig. 3) and stratigraphic position.

Basaltic tephra T66 was sourced from Mt Albert (119,200 ± 2,800 (1 σ) years; Leonard et al., 2017). This tephra was documented here for the first time in the AVF tephra framework (Hopkins et al., 2015, 2017; Molloy et al., 2009). We have termed it "AVFaa" (see Appendix A for details).

### 4.2 Radiocarbon dating

All 54 radiocarbon dates from the Orakei core (53 new and the NZA28865 date previously published in Hayward et al. (2008)) are summarised in Table 3.

The Bacon age model recognises 13 ages as outliers (24%), which is in agreement with visual identifications in comparison to the model (Fig. 6). The four wood sample ages OZW884-887, and 3 bulk sediment sample ages OZX869, OZX888 and OZX889 are considered outliers being much older than bracketing ages and thus the age-depth model (Fig. 6). These outliers,

all between 12.5 and 16.5 m-depth, are interpreted as representing fluvial in-wash containing reworked organic matter and, in the case of bulk sediment ages, carry additional uncertainties from necessary reservoir corrections. The ages for OZX873 and OZX882 are also older than anticipated. This is most likely due to the unknown reservoir correction, since no double-dating of these facies units (5-7) was possible due to an absence of plant macrofossils. The ages OZW874, OZX340, OZX343 and OZX878 are too young and probably explained by the first three yielding very small masses of organic carbon, whereas the

reason for the latter age is unclear. The Bacon age model recognises all 13 outliers and hence there was no need to remove them manually.

The facies-dependent reservoir effect in facies 4 is 1,026 ± 117 yr and 410 ± 170 yr in facies 8. Where the reservoir correction is stated with a question mark in Table 3, no reservoir correction for these specific facies could be calculated. In that case we used the reservoir correction from the stratigraphically closest facies (i.e., reservoir correction from facies 4 also applied to

facies 3 and 5, marked with a question mark in Table 3).

### 4.3 Luminescence Dating

Luminescence data were obtained for six of the ten fine-grained samples measured from the Orakei series, and for one coarse-grained sample, L20. Samples L18, L20 and L29 yielded insufficient natural signal for equivalent dose estimation. These are also the youngest luminescence samples according to depth and therefore had accumulated less charge during burial. The

sensitivity of the Orakei samples was low for feldspars and may suggest a primary igneous origin.

The remaining six samples provided ages, and these results are summarised in Table 4. The results broadly conform (within 2σ error) to the expected ages based on the other approaches presented here but suffer from limitations as described below and in Appendix B.

The luminescence characteristics of the Orakei samples were not optimal for age determination. Ages could only be calculated

by relaxing the acceptance criteria for analysis. The samples are typically very dim, yielding generally low signal counts (Fig. B4a) and low sensitivity. Late background subtraction was used to optimize the signal measured. The sensitivity to test dose over the regenerative cycles of the pIR-IRSL$_{290}$ protocol typically decreased to 20-30% of the test dose sensitivity following the natural signal measurements and remained fairly constant over subsequent cycles (Fig. B4b). Dose-response curves could often, but not always, be fitted to an exponential or exponential-plus-linear function, but rarely passed through the origin due





to the high degree of thermal transfer (Fig. B4c). Recuperation was very high, yielding values of up to 50%, and recycling ratios were variable but often exceeding 20% divergence from unity (Table B1). In order to obtain results, aliquots were accepted if dose-response curves could be reasonably fitted; threshold criteria for other quality control measurements such as recycling ratios and recuperation were relaxed such that values of 1.0 ± 0.4 and <67% (mostly <25%) were accepted for analysis respectively. The residual dose test run on sample L15/A0111 provided usable signals only in 1/3 aliquots, and so the

residual correction was made on the basis of that lone aliquot. Despite these limitations, the residual-corrected ages yield results more closely fitting with the age predicted by the matching of the paleomagnetic RPI to the PISO-1500 stack.

### 4.4 Paleomagnetism

Down core paleomagnetic measurements of the Orakei sediment sequence were undertaken to support the development of the chronology with the identification of known-age excursions of magnetic field inclination and reduced intensity of the Earth

magnetic field. Additionally, relative changes in the strength of the Earth's magnetic field were estimated to allow the transfer of chronological information from dated sediment sequences showing corresponding variations in the relative paleointensity.

Mineral magnetic data (see Appendix C) of the Orakei sediment sequence is consistent with the detrital magnetite as the main magnetic carrier. Following Peters and Thompson (1998) we classify the magnetic assemblage of the Orakei samples as magnetite/titanomagnetite (Fig. C5a). We identify both detrital and extracellular magnetite components based on the definition

of Egli (2004; Fig. C5b), with the latter mostly found in facies 10. Variations in magnetic grain size, estimated following Thompson and Oldfield (1986), suggest that most Orakei sediment samples used for paleomagnetic measurements fall in the pseudo-single domain range of magnetite (Fig. C5c), consistent with a detrital origin.

The coring procedure at Orakei prevented useful declination determination. The 1 m-long core sections were not oriented with respect to magnetic north/south and several segments were further rotated as they broke along coarser clastic layers such as

tephra.

The ChRM inclination in the lower ~40 m of the Orakei sediment sequence mostly varies from -50 to -60°, which is consistent with the geocentric axial dipole (GAD) prediction of -56° for the site latitude. A reversed polarity direction with inclination reaching +66° is observed at 43.24 m (Fig. 4A). Its position between the Tahuna and Rotoehu tephra layers (see section 4.1) implies that this short-lived reversal can be assigned to the Laschamp Excursion (41,400 ± 350 yr; Lascu et al., 2016).

Significantly shallower inclinations are recorded between ~63 and 68 m, as well as around the facies unit boundary 13/12 at ~73 m (Fig. 4). The inclination flattens to -4.1° at 73.42 m, 10 cm below the tephra "AVFaa" from the Mt Albert eruption (119,200 ± 2,800 yr; Leonard et al., 2017; Fig. 4)). This age constraint in proximity of this near-reversed inclination, along with pronounced and sustained decrease in RPI starting at the time of deposition of "AVFaa", suggests that it corresponds to the Blake Excursion (116,500 ± 700 to 112,000 ± 1,900 years, (Osete et al., 2012)). Other occurrences of reversed/transitional

(positive or near-zero) inclinations at 49.77 m, 51.23 m, 63.38 m, and shallower inclinations between ~63 and 68 m, occur in intervals of coarser grain size more likely indicating thin mass movement deposits (King, 1955). No data was obtained between 65.8 and 67.78 m due to sand bands preventing sampling of the intercalated laminated sediment as the sample cubes are larger (2.2 cm) than samples taken for other techniques (e.g., 1-cm slices for Beryllium, see below).

The NRM and ARM variation of the Orakei sediment sequence show the same broad pattern in each demagnetisation step

(Fig. 4B, C). Very low NRM and ARM in facies unit 14 are followed by a sudden increase at ~76 m (Fig. 4B, C), then decreasing stepwise until very low values are reached again at ~60 m. Up to 40 m, NRM and ARM values remain very low, although the ARM curve is interrupted by slightly elevated values in facies unit 9 (~55-50 m; Fig. 4C).

To calculate the relative paleointensity - expressed by the NRM/ARM ratio - we used the NRM and ARM after 20 mT AF demagnetisation, as a similar pattern was obtained from all demagnetisation steps (different colours in Fig. 4D). Very low

values are associated with the partially-reversed polarity direction tentatively identified as the Laschamp Excursion (Fig. 4). The strength of the global magnetic field is known to have been exceptionally low during the Laschamp Excursion (Channell





et al., 2009), supporting this interpretation. Equally low RPI values are also observed in facies unit 9 and facies unit 11 (Fig. 4). These facies unit represent coarser grain size facies, suggesting that the low RPI values could be artefacts associated with less efficient NRM recording in a higher energy depositional environment (compare Fig. 2). This may indicate a possible link

between sedimentary facies and the efficiency to acquire a NRM signal. The down-core variation in NRM/ARM (Fig. 4D) is assumed to be proportional to the scatter around a linear trend-line in a scatterplot (Fig. C6). However, the offset between the linear trend-lines fitted to the data points of the coarse and the fine-grained facies units indicates only a very small difference in efficiency of NRM acquisition. Hence, no facies-dependent correction is applied to the paleomagnetic intensities.

The Orakei RPI shows marked variation with a short-lived peak at 78.35 m (Fig. 4D) and troughs around 70, 73, and 76 m.

Pronounced low RPI values between ca. 65 and 61 m are followed by high-frequency variability up to ~55 m, then a pronounced decrease to reach a short-duration trough around 52 m. RPI values then increase toward a maximum around 49 m, followed by a step-wise decrease to a RPI minimum coincident with the Laschamp Excursion (Fig. 4D).

### 4.5 Beryllium

Relative changes in the $^{10}$Be production rate in the atmosphere, at first order, should reflect fluctuations in the relative strength

of the Earth's magnetic field, whilst on shorter time scales, it should reflect solar magnetic shielding assuming a constant galactic cosmic ray flux (e.g., Frank et al., 1997; Muscheler et al., 2005). As such, the direct-fallout meteoric $^{10}$Be record can be expected to provide an inverse record to the relative paleointensity time-series described in section 4.4. However, in this case, the authigenic $^{10}$Be signal may have contributions from non-direct-fallout component(s) such as associated with catchment-derived erosional influx and material of aeolian origin, and thus may be modulated by associated grain size

variability. Authigenic (HCl-extracted) $^9$Be serves as the normalizing factor for $^{10}$Be geochemistry to compensate such effects but may also indicate shifts in the sedimentary regime on its own.

### 4.5.1 Beryllium-9

$^9$Be shows nearly constant, long-term increasing values from the core base up to large step-wise increase between 50 and 45 m reaching its maximum of ~6 x $10^{17}$ atoms/g just below the Rotoehu tephra layer, then returning to constant (near pre-peak)

values up core (Fig. 5A). The large peak between 47 and 45 m may reflect accelerated weathering and erosion in the catchment and/or wind-blown deposits with a different $^9$Be inventory although the catchment is confined to the crater rim environment and enhanced weathering and erosion are not apparent at this depth in the Orakei core.

### 4.5.2 Beryllium-10

The $^{10}$Be record (from ANSTO, black in Fig. 5B, D) increases up-core, peaking at the top of unit 10 (~57 m), interrupted by a

prominent trough in facies unit 9 and followed by relatively constant values across facies unit 8, punctuated by discernible peaks at ~44 and ~42 m. The Lund-ETH $^{10}$Be data largely follows the ANSTO $^{10}$Be but deviates considerably for three of its deepest four samples (i.e., a below ~44 m (Fig. 5B, 5D)).

The trough in $^{10}$Be in the coarser facies unit 9 (50 – 55 m depth) is interpreted to be caused by a higher proportion of coarser (sandy) grains and hence, reduced surface density of $^{10}$Be/g, reflecting apparent reduction of surface area per unit of mass in

coarser grains. The same behaviour of the $^{10}$Be profile is observed in facies unit 11 (~64 – 69 m depth, sand and silt dominated) but the trough is not as distinct as in facies unit 9 (Fig. 5B). This is likely a response to the more effective separation between distinct coarse-grained sand bands, which have been removed on the ECD, and intercalated finer silt from which the Beryllium samples originate. On the other hand, in facies unit 9, the sand bands are less distinct, and the Beryllium sample material likely originates from a multi-modal grain size distribution in which particles of different grain sizes (clay, silt, sand) are mixed.




### 4.5.3 Beryllium-10/ Beryllium-9 ratio

In order to correct for catchment processes that modulate the $^{10}$Be concentration (e.g., sediment influx variability, grain size variation and geochemical variation in Beryllium transport and deposition) the $^{10}$Be/$^9$Be ratio (Carcaillet et al., 2004) is plotted in Fig. 5C. The data incur high analytical uncertainties originating from low level $^9$Be ICP-MS measurements near detection limits (ranging from <1 to 13 ppb, with most data <2 ppb to which we assigned 30% errors). Very low ratios in facies unit 14 are followed by elevated ratios in unit 13, which are largely maintained to ~48 m-depth (Fig. 5C). The uppermost studied interval is characterised by a pronounced $^{10}$Be/$^9$Be trough (due to the strong $^9$Be peak at 45.5 m) followed by an abrupt increase near coincident with facies unit boundary 8b/8a (dashed grey line in Fig. 5C) and associated with elevated values up to ~40 m-depth (Fig. 5C).

The conspicuous saw tooth profile in the $^{10}$Be/$^9$Be ratio in facies unit 9 originates from $^9$Be measurement uncertainties (Fig. 5A, C) rather than from the bimodal grain size distribution of sand and silt, which we avoided. Samples were measured for $^{10}$Be and $^9$Be in two campaigns in 2018 and 2019 with the former having larger uncertainties. Thus, we attribute the scatter mostly to measurement uncertainties rather than a true signal of the geomagnetic field strength, cosmogenic nuclide production or sediment influx. In summary, $^{10}$Be/$^9$Be ratios are associated with large uncertainties preventing any correlation to paleomagnetic variations with high confidence. On the other hand, $^{10}$Be data, except for unit 9 where effects of larger grain size cannot be ruled out, are probably more reliable and discussed further below.

### 4.5.4 Paleomagnetic Excursions in the Beryllium record

Enhanced galactic cosmic-ray production of $^{10}$Be during a paleomagnetic excursion, such as the Laschamp Excursion (McHargue et al., 1995) enables the use of peaks in the $^{10}$Be concentration record to identify dated excursions and hence to include them in the age modelling process. Based on the estimated age range covered by the lower ~40 m of the Orakei sediment sequence, for which we have obtained a Beryllium isotope record, we expected to see elevated $^{10}$Be (and/or $^{10}$Be/$^9$Be) during the Laschamp and Blake Excursions. The magnitude of the peak above baseline would depend on the magnitude of the decrease in paleomagnetic field intensity and the impact of competing terrestrial processes which can also alter Beryllium delivery and deposition.

The Tahuna (37,800 ± 900 cal yr BP) and Rotoehu (45,100 ± 3,300 yr BP) tephra layers serve as upper and lower age markers, respectively, for the Laschamp excursion in the Orakei sediment sequence. For a detailed, high resolution study of this interval, we use $^{10}$Be data from ANSTO and Lund/ETH (Fig. 5D). Aside from one data point at ~44.5 m and two data points below 46 m-depth, the two curves overlap well despite the Lund/ETH samples above 46 m representing mixed sediment from 10-cm intervals (i.e., ~250 yr) and the ANSTO samples representing 1-cm sediment slices (i.e., ~25 yr) of sediment only. Three Lund/ETH samples below 46 m, which deviate the most from ANSTO data, represent mixed material from 2-cm intervals (~50 yr). Hence, the ANSTO and Lund/ETH $^{10}$Be data may reflect short-term field intensity fluctuations (e.g., 11-yr solar cycles; (Beer et al., 1990)) resulting in the inconsistency between both datasets (Fig. 5D).

$^{10}$Be shows increasing values from ~44 to 42 m with a distinct peak at 42.5 m and return to lower $^{10}$Be at 41 m-depth (Fig. 5D). This peak may represent the Laschamp Excursion at Orakei and the onset may be placed at 43.3 m-depth which aligns well with the sharp reversal in paleomagnetic inclination (Fig. 4A) and RPI minimum (Fig. 4D; see section 4.4).

The $^{10}$Be increase in the lower part of the section (facies unit 13) is interrupted by smaller peaks. None of these stand out above background level as would be expected for the Blake Excursion (116,500 ± 700 to 112,000 ± 1,900 years (Osete et al., 2012)). However, the "AVFaa" tephra (119,200 ± 2,800 years (Leonard et al., 2017)) provides a suitable time marker in proximity to where the Blake Excursion was expected (Fig. 5). Small peaks in $^{10}$Be at 73-75 m-depth correspond to the inferred level of the Blake Excursion (Fig. 5C). However, the Blake Excursion is not independently identifiable in the Orakei sediment sequence, neither in the paleomagnetic inclination or intensity (RPI) records (see section 4.4, Fig. 4) nor in the cosmogenic $^{10}$Be record. Consequently, the Blake Excursion is not used to inform the chronology development.





In the present study, the [10]Be data, although showing trends and variability correlated with different facies units, is limited in its ability to directly reflect geomagnetic field variations, whilst the [10]Be/[9]Be data suffered from large analytical uncertainties. For example, the Laschamp reduction in geomagnetic field strength approximately doubled the globally-averaged atmospheric

production rate of [10]Be as observed in polar ice core [10]Be records (e.g., Wagner et al., (2000)). In this study, the inferred Laschamp [10]Be is no larger than x1.3 above adjacent base level [10]Be and the Blake Excursion barely stands out. Variations in the [10]Be/[9]Be data suggest strong catchment and/or regional influences on isotopic Beryllium records that may not easily be compensated via normalisation with [9]Be. Therefore, the interpretation of [10]Be in terms of geomagnetic field changes and its utility for dating appears to be rather limited in the context of the Orakei maar sediment sequence.

### 495 4.6 Tuned Paleomagnetic RPI curve

Visual inspection of the Orakei RPI and the variation of the PISO-1500 VADM (Channell et al., 2009) over the period ca. 30,000 – 140,000 years suggests strong similarities between both curves (Fig. 6A, B). This correlation facilitates an improvement of the absolute age model by tuning the Orakei RPI to the PISO chronology. Records were scaled by subtracting the mean of the RPI time-series and dividing by its standard deviation. The PISO-1500 stack has a temporal resolution of one

datapoint per 1000 years. To reach a similar resolution in both time series, the PISO-1500 stack was linearly interpolated to a 200 year-resolution. The interval between the Laschamp Excursion (ca. 41,100 yr) and the 'AVFaa' tephra (ca. 119,200 yr) was extracted from the Orakei RPI record and compared to the PISO-1500 stack between the equivalent ages. DTW was then performed in R (package dtw (Giorgino, 2009)) with closed start and end points on the two time-series, and using a Euclidean distance-based cost matrix and symmetric step pattern. The remainder of the Orakei RPI curve was then matched from the

'AVFaa' tephra to its base with an open end to the PISO-1500 stack up to 140,000 years (estimated maximum age of the sequence by extrapolating the sedimentation rate). We integrated 13 tuning points into a Bacon age model. We selected 10 random tuning points (TuP in Table 5) from the alignment path (solid line in middle panel of Fig. 6C, D) over the interval between the Laschamp Excursion and the 'AVFaa' tephra. Two additional tuning points (TuPex) from the alignment path over the interval from the 'AVFaa' tephra to the Orakei core base were extracted randomly and supplemented by the basal tuning

point (TuPex115 in Table 5). The depth and age values of these 13 tuning points were incorporated into the Bacon age modelling process (see section 3.7 and Table 5). A standard deviation of ±1000 years is assumed for all tuning points which equals the temporal resolution of the PISO-1500 stack.

The fit of both curves following DTW alignment is presented in Fig. 6E. Neither [10]Be concentration nor [10]Be/[9]Be curves display a comparable pattern, likely due to an unidentified catchment-influx overprint (Carcaillet et al., 2004). We therefore

do not incorporate the Beryllium results into the tuned chronology.

### 4.7 Integration of multiple chronologies: The Orakei age model

All ages from radiocarbon dating, tephrochronology, the Laschamp Excursion, luminescence dating and the RPI tuning points obtained in the Orakei sediment core are presented in Fig. 7. The age of the Blake Excursion has not been used for the chronology development, since its identification in the Orakei sequence is equivocal in both the paleomagnetic and [10]Be data,

and the chronology for the lower part of the Orakei sequence is mainly guided by the "AVFaa" tephra. The age model obtained using Bacon is shown in Fig. 8. The mean 95% confidence age range is 1,599 years, with a minimum of 227 years at 2.58 m ECD and a maximum of 3,084 years between 69.65 and 69.68 m ECD. Only 65% of all ages overlap with the 95% range of the age-depth model.

The Orakei age-depth model defines the age of onset of lacustrine sedimentation (at 79.24 m ECD) as 130,120 yr (95%

confidence range 128,665 - 131,560 yr). Following that, the age-depth model is predominantly guided by the RPI tuning points (Fig. 8) despite not always passing through or near the mean. This occurs as the modelling process is designed to avoid strong or abrupt changes in the accumulation rate unless supported by a hiatus (which has not been observed). The Orakei age model





agrees well with the "AVFaa" tephra age (Fig. 8). Except for sample L32, the Orakei age model lies well within the error ranges of the accepted luminescence ages (L15, L14, L35, L65; Fig. 8), despite their limitations. Aside from the outliers (see

section 4.2), most radiocarbon ages, all known-age rhyolitic marker tephras and the Laschamp Excursion age agree well with the Orakei age-depth model in its upper half (Fig. 8). The facies units 6 and 7, as well as (parts of) 2 and 3, are however, much less well constrained. The radiocarbon dating density is less than ideal given the lack of *in-situ*, datable terrestrial macrofossils in these facies units. Furthermore, the absence of macrofossils adds uncertainty to the freshwater reservoir/input of old carbon correction estimates and potentially introduces additional error in the bulk sediment radiocarbon ages of the bulk sediment

(Fig. 8).The shape of the age-depth model (steepness, inflection points, angles) represents changes in the sedimentation rate which are described in section 4.8 and shown in Fig. 9.

### 4.8 Sedimentation rate variability

Our description and interpretation of the sedimentation rate follows its mean (red line in Fig. 9). At the Orakei core base the sedimentation rate is ~ 0.2 cm/yr and maintaining a fairly constant rate, expect a short-term drop within the finely laminated

facies unit 10 to ~45 m ECD in facies unit 8 (Fig. 9). Stepwise accelerations in sedimentation rate follow and reach ~0.6 cm/yr around the boundary between the facies units 8 and 7. An interval with slightly decreased sedimentation rates in facies unit 4 is followed by a jump to maximum sedimentation rate of ~ 0.9 cm/yr at ~20 m ECD (Fig. 9). High but slowly decreasing sedimentation rate towards the Orakei core top follows in facies units 3 and 2 with an abrupt drop to ~0.4 cm/yr at 8 m ECD (Fig. 9). Another strong decrease in sedimentation rate occurs at the onset of the peat interval (facies unit 1) which records

values below 0.2 cm/yr (Fig. 9).

### 5 Discussion

### 5.1 Strengths and weaknesses of the Orakei age-depth model

A strength of the Orakei chronology lies in its multi-method approach and high dating density and resolution in the post-45,000 yr BP range using Bayesian age-depth modelling. The Orakei age model agrees with the well-constrained ages of all rhyolitic

tephra markers layers found in the sequence highlighting another strength of the presented chronology. The fit between the reference PISO-1500 stack VADM (Channell et al., 2009) and the Orakei RPI is robust for the pre-45,000 yr BP section of the record (see Fig. C7). Using the Earth's magnetic field strength for relative dating, the Orakei age-depth model is independent of presumptions of climate event synchroneity which is one of the central questions of studies using this tuning approach to chronology development (Blaauw, 2012). Avoiding this circularity allows reliable intra- and inter-hemispheric comparison of

the Orakei sequence to independently-dated proxy records of environmental, climatic and oceanographic change, for example Greenland and Antarctic ice cores (EPICA Community Members, 2006; NGRIP Members, 2004; Rasmussen et al., 2014), marine sediment cores (Carolin et al., 2013) and terrestrial records from Australia (Kemp et al., 2020), South America (Zolitschka et al., 2013) and central and eastern Europe (Lézine et al., 2010; Pickarski et al., 2015; Seelos et al., 2009).

Relative dating using DTW alignment and absolute age control does not require any correction for down core compaction,

which is notoriously difficult to estimate and would have been necessary if the basal age was primarily derived from sedimentation rate extrapolation (Allen, 2000; Bird et al., 2004; Stanley and Hait, 2000). The integration of the DTW-derived tuning points into the Bayesian Bacon age model allows for some flexibility, since the age model does not have to pass directly through chosen mean of the tuning point if this does not agree with other calculated ages and/or violates the assumptions of smoothness.

The weaknesses of the Orakei chronology include difficulty in assessing the sedimentological context of facies units 5 to 7, and the unknown freshwater reservoir correction for [14]C measurements due to an absence of datable macrofossils. This results in higher age uncertainty, complicating the validation of sedimentation rates for this interval. Luminescence dating and the



[10]Be variation were not as robust as hoped, most likely due to the inferred detrital volcanic source of the pIR-IRSL samples and unaccounted for variability in catchment sediment processes impacting the Beryllium isotope influx.

The DTW tuning approach only works successfully if a long overlap between the reference and query time series is present and is dependent on the absence of (large) gaps in the query record. The Orakei core is characterized by continuous sedimentation since ca. 130,000 years ago. However, the Orakei RPI record shows a gap between 65.8 – 67.89 m ECD where no samples could be collected. This has potential implications for the alignment of the RPI to the PISO- stack, since it is not known how much of the RPI record is missing. The resolution (and quality) of the alignment achieved through DTW is

restricted by the resolution (and quality) of the reference and original data sets. Whilst well-suited to the task thanks to continuous sedimentation, the PISO-1500 stack has a resolution of 1,000 years per measurement (Channell et al., 2009), whereas the Orakei RPI record averages one measurement per 168 years. Furthermore, our approach relies on an approximate age at or near the core base, which is necessary to estimate the total time interval contained in the record and thus to select the corresponding section in the reference data for reliable tuning of both curves. In our study, this is achieved by the "AVFaa"

tephra for most of the Orakei sediment sequence. Our random selection of a finite number of DTW tuning points may influence the resulting age model. However, additional tuning points reduce the flexibility of the Bacon age modelling process, which may result in overfitting of the age-depth model and unrealistically low confidence ranges for the final model.

**5.2 Validation of the final age model**

In order to assess the validity of the Orakei sediment core chronology, we expect variability in sedimentation rate to be in

agreement with the observed lithological changes. The major difference in sedimentation rate between the upper ~40 m of the core, which has higher rates than the lower ~40 m of the Orakei core (Fig. 9), supports this hypothesis, since we observe a dominance of finely- laminated sediment in much of the lower facies units 8-14, and much thicker laminations (unit 4) and fluvially in-washed sand (unit 3) in the upper part of the core. However, there is no significant link between lithology and sedimentation rate in other parts of the core as observed in the near constant sedimentation rate from the core base into the

sandy facies unit 11 at ~65 m-ECD, and again in the alternating silts and sands of facies unit 9, and up to ~45 m-ECD in facies unit 8. All visible mass movement deposits have been removed from the sequence on the ECD used in this study; after removal of the sand bands (visible mass movement deposits), a constant sedimentation rate for the remaining matrix of laminated clay and sand does not conflict with observed lithological changes. The slowest sedimentation rate of the entire sequence is observed in an interval of very fine laminations (facies unit 10; Fig. 9), which is consistent with slow sedimentation. Fine laminations

generally indicate slower accumulation rates because only small amounts of sediment are slowly deposited at the lake bottom. However, fine laminations observed below 70 m ECD in facies units 12 and 13 (Figs. 2, 9A) also include multiple (mostly thin) mass movement deposits. Whilst these have been removed from the sequence, it is plausible that some thin mass movement deposits (< 1 mm thickness) could have been missed, thereby explaining the faster sedimentation rate compared to facies unit 10 (Fig. 9).

The laminated sediment interval above ~45 m ECD is characterised by a uniform acceleration in sedimentation consistent with the thicker laminations in this section of the core. Comparing sedimentation rate with the lithology is hampered by our poor understanding of the depositional mechanisms associated with facies units 5 to 7. A fast sedimentation rate is consistent with facies unit 6, a massive unit containing reworked basaltic tephra which may indicate near-instantaneous deposition and thicker laminations in facies unit 5. Generally, higher sedimentation rates agree well with thick laminations in facies unit 4, and fluvial

sand bands in facies unit 3, as well as massive deposition in a shallow lake indicated by bioturbation in facies unit 2 (Fig. 9). Very slow sedimentation inferred for facies unit 1 is reasonable considering that this is peat.

Overall, whilst limitations in any age-depth model are inevitable, we nevertheless find that the Orakei sedimentation rates agree reasonably well with the observed changes in sedimentary facies types and properties. Furthermore, our study improves





substantially on the simple extrapolation age models previously employed in the AVF maar sequences, especially >40,000

years BP (e.g., Molloy et al., 2009).

**6 Conclusions and outlook for chronology development of long sediment sequences**

The importance of reliable chronologies for paleoenvironmental studies in lake sediment sequences cannot be overemphasized. For this reason, we developed a detailed chronology for the Orakei maar lake sediment sequence, the highest-resolution and most complete lake sediment record spanning the LGI from the terrestrial SW Pacific region. The chronology developed here

significantly improves upon previous age models for AVF maar lakes, which until now have largely relied on sedimentation rate inter- and extrapolation, especially beyond the radiocarbon dating limit (e.g., Molloy et al., 2009; Hopkins et al., 2017). Our robust and high-resolution age model provides the essential chronological framework for event-based paleoenvironmental and -climatic reconstruction over the LGI without which such work would not be feasible.

Our study also highlighted difficulties involved in combining absolute ages with relative dating of RPI variability in Bayesian

age-modelling. Estimating a realistic error for the tuning points so as to incorporate them into the Bayesian age-model is problematic, and potentially underestimates the 95% confidence range of the age model.

Several recommendations for future development of chronologies in comparable sediment sequences are proposed from this study:

(1) RPI variations are suitable for correlative dating with other records due to their independence from assumptions on climatic

a-/synchroneity between regions, as long as independent age control is provided. In different depositional settings this may be successfully supplemented or replaced by Beryllium-10 variability, since both are indicators of magnetic field strength.

(2) We cannot over-emphasize the need for continuous and high-resolution sediment sequences for relative dating (tuning). Gaps in the proxy record complicate the alignment and increase age uncertainties. An improved uncertainty assessment of the DTW alignment and resulting tuning points is necessary.

(3) Sediment facies without datable, *in-situ* terrestrial macrofossils for radiocarbon dating are not well dated and suffer from the unknown estimation of the freshwater reservoir correction.

Despite the issues encountered with the development of the Orakei maar lake chronology, such an approach is necessary to enable robust records of past climate, especially at the event level, during the LGI that are well known from Northern

Hemisphere equivalents such as Eifel maar lakes (Seelos et al., 2009). The ability to develop lake sedimentation chronologies independently of inferred links with climate events is of paramount importance for the reliable extraction of paleoclimate signals that can then be correlated with similar and well-dated time-series of climate change in both intra- and interhemispheric contexts.  Only then can we produce the critical missing links between past climate drivers and subsequent events from the poles to the Equator from which to develop a robust understanding of the teleconnections and associated leads/lags within the

Southern Hemisphere and global climate system.

The Orakei maar lake work is ongoing with further refinement of the Orakei chronology and paleoenvironmental and -climatic record expected and application of a similar approach to other select maar lake records from the AVF. It is anticipated that the Orakei maar chronology will form the master chronology against which those developed from the other AVF maar lakes will be compared. Such a comparison is particularly feasible given the widespread and excellent preservation of marker tephra

layers in each maar lake sediment sequence that has already been used to develop the complete record of distal tephra to have reached the Auckland area, both from the TVZ and from the local AVF volcanic centre eruptions. The ability of the multi-method approach detailed here to produce robust ages for tephra and hence volcanic events beyond the limit of the radiocarbon method is of importance for reliable determination of past eruption frequencies and associated hazards associated with both





the local AVF volcanic centres and those sourced from the TVZ. This is of particular relevance given the fact that Auckland
       City is built around 53 basaltic volcanoes, the last of which erupted only $504 \pm 5$ cal yr BP (Needham et al., 2011).

**Appendices**

**Appendix A: Tephra**

**Re-calibrated radiocarbon ages for three tephra layers**

The ages for the Okareka, Maketu and Tahuna tephra layers were re-calculated using [14]C ages from (Molloy et al., 2009) as
follows: the ages OZK291 and OZK293 (bulk sediment samples 10 mm above Maketu and Tahuna, respectively) were
       reservoir-corrected with the correction of $410 \pm 120$ years as both layers are located in facies unit 8 of the composite Orakei
       sediment sequence (section 4.2). The reservoir-corrected ages, as well as OZK292 and OZK294 (macrofossil charcoal samples
       10 mm below Maketu and Tahuna, respectively), were calibrated with the SHCal13 calibration curve (Hogg et al., 2013). The
       pairs from above and below the tephra (OZK291, OZK292 for Maketu; OZK293, OZK294 for Tahuna) were then combined
in OxCal 4.3 (Bronk Ramsey, 2009) with the "C_combine" command to obtain a new age for the tephra layers.
       Only [14]C ages from leaf fragments were accepted for the age determination of the Okareka tephra layer from the Molloy et al.
       (2009) [14]C-dataset. The [14]C ages OZK262 and OZK263 were combined in OxCal 4.3 (Bronk Ramsey, 2009) with the
       "R_combine" command to obtain a combined conventional radiocarbon age for the Okareka tephra (used in section 3.2.1 for
       reservoir-correction determination), which was then calibrated with SHCal13 (Hogg et al., 2013).

**ID of sample T66**

An unidentified basaltic tephra layers "T66" could not be matched stratigraphically or geochemically to any other Auckland
       Volcanic Field (AVF) tephra layer previously documented in Molloy et al. (2009) or Hopkins et al. (2015; 2017). Its occurrence
       between the layer AVF1 and the basal eruptive material sourced from the Orakei Volcanic Centre (Fig. A3) suggests that it is
       a new, previously undocumented tephra layer from a different AVF source. In order to test this hypothesis, electron microprobe
analyses (EMPA) were conducted on sample T66 glass shards at Victoria University of Wellington following the method
       outlined in section 3.1. The results are presented in figures A3-A5. We confirmed that T66 is not a correlative to the Orakei
       Volcanic Centre and/or AVFa tephra layer (Fig. A3). Furthermore, glass shard geochemistry also obviates correlation to the
       AVF1 tephra layer (Fig. A4). We consider that T66 constitutes a basaltic tephra that has not so far been encountered in any
       AVF maar cores (Hopkins et al., 2015, 2017).
Based on the position of T66 between AVF1 (correlated to Domain/Grafton eruption) and AVFa (correlated to Orakei eruption;
       Hopkins et al. (2017)), several AVF volcanoes are potential sources for the T66 tephra (Hopkins et al., 2017; Leonard et al.,
       2017). Despite the difficulties in comparing whole rock and tephra material based on major element oxides, we compare data
       from sample T66 to the potential AVF source centres (Fig. A5): Grafton Park, Mt Albert, Mt Roskill, as well as the correlatives
       for AVF1 and AVFa, Domain and Orakei, respectively. Low FeO and $SiO_2$% in Mt Albert whole rock composition as well as
in sample T66 suggests strongly that Mt Albert is the source of the T66 tephra layer (Fig. A5). The formative eruption of Mt
       Albert has been Ar-Ar dated to $119.2 \pm 2.8$ (1 sd) ka by (Leonard et al., 2017) which falls between the ages obtained for AVFa
       ($126.15 \pm 3.32$ (2 sd), Hopkins et al., 2017) and AVF1 ($106.17 \pm 4.3$ (2 sd) ka (Hopkins et al., 2017); 83.1 ka (Molloy et al.,
       2009)) giving further support for this identification. Hence, we conclude that sample T66 is sourced from Mt Albert with an
       eruptive age of $119. \pm 2.8$ ka. In order to maintain nomenclature consistency, we assign the name "AVFaa" to this tephra layer.



**Appendix B: Luminescence dating**

**Methods**

We undertook luminescence dating of polymineral fine-grained (4-11 μm) material from selected samples of the Orakei core, focusing on the feldspar signal. The fine-grained fraction was targeted in order to maximize the percentage component of non-volcanic, distally sourced dust, which is more likely to yield a reliable luminescence signal. Additionally, coarser grained (63-

90 μm) aliquots of two selected samples (L15 and L20) were run.

The sediment cores were stored wrapped in light-proof plastic at 4˚C until opened and subsampled for luminescence dating under subdued orange light at the University of Auckland (New Zealand) taking care to only sample the innermost parts of the core for dating. Samples were taken to ensure maximal spread over the >45,000 years interval of the Orakei sequence following a preliminary age model by extrapolation of the sedimentation rate calculated between the two oldest identified and dated

tephra layers (Tahuna and Rotoehu, see Table 2). Luminescence dating sample pre-treatment for polymineral fine grains (4-11 μm) was undertaken at the Max Planck Institute for Chemistry (Mainz, Germany) following published methods (Mauz and Lang, 2004), involving hydrochloric acid and hydrogen peroxide digestion, sieving and settling. Enough material remained after digestion, sieving and settling for production of 15 aliquots of the polymineral fine-grained component, except for sample L69 (A0119) which produced 10 aliquots. Each aliquot was pipetted onto 10 mm diameter stainless steel cups for equivalent

dose ($D_e$) measurements. Additionally, K-feldspar was extracted from the coarser grained material (90-115 μm) if sample L20 (A0113) by density separation using lithium heterotungstate. Large aliquots (8 mm diameter) were prepared from this material for equivalent dose measurement.

Equivalent dose (De) was measured on the fine-grained samples using the post-infrared infrared (pIR-IRSL290) protocol, stimulated at 290°C to minimize the potential for feldspar signal fading (Buylaert et al., 2012; Thiel et al., 2011; Thiel et al.,

2014), and based on equivocal results of the preheat plateau test which showed substantial inter-aliquot scatter irrespective of measurement temperature (Figure 1). For the coarse-grained sample L20 (A0113), 12 aliquots were run using the pIR-IRSL290 protocol using an infrared (IR) preheat temperature of 50°c, and another 12 aliquots using the same protocol with an IR preheat temperature of 200°C, in order to compare luminescence characteristics in response to changing parameters. $D_e$ measurements were undertaken using an automated Risø TL-DA-20 reader using infrared (IR) LEDs for illumination, with the resulting

signal detected by an EMPI 9235QA photomultiplier tube with a D410 filter for feldspar (Bøtter-Jensen, 1997; Bøtter-Jensen et al., 1999). Irradiation was provided by a calibrated $^{90}Sr/^{90}Y$ beta source (Bøtter-Jensen et al., 2000). An internal alpha contribution of 0.12 was assumed for the samples, in order to account for the lower luminescence efficiency of alpha radiation relative to the beta and gamma components (Rees-Jones, 1995; Rees-Jones and Tite, 1997). The accepted aliquots of all samples yielded either normal distributions, which were typically very wide, or skewed distributions indicative of incomplete

bleaching. In the first case, $D_e$ values were calculated using the Central Age Model (CAM) (Galbraith et al., 1999); in the latter, $D_e$ was calculated using the Minimum Age Model (MAM) (Olley et al., 2004) (Fig. B2). For the coarse-grained sample L20 (A0113), 2 out of 12 aliquots run with an IR50 preheat were not saturated with respect to dose, in comparison with 6 out of 12 aliquots from the IR200 preheat. Whilst this suggests better results using the higher preheat, the resulting equivalent doses were similar, as illustrated in the radial plot in Figure 3.

Samples for dose rate determination were digested with $HNO_3$ and HF at the Australian Nuclear Science and Technology Organisation (ANSTO, Sydney, Australia). U, Th concentrations were then obtained from these solutions on an Agilent 7700 inductively-coupled plasma mass spectrometer (ICP-MS) at the Mass Spectrometry Equipment Centre (University of Auckland, New Zealand). K concentrations were obtained from XRF measurements on fusion disks on a SPECTRO X-LAB 2000 polarizing energy dispersive X-ray fluorescence spectrometer at the University of Waikato (New Zealand). The measured

activities of the radioactive elements K, Th, and U were converted to dose rates using published factors (Adamiec and Aitken, 1998; Guerin et al., 2011). Dose-rate attenuation was estimated using published values (Mejdahl, 1979) and the core sediment



moisture content. Water content was estimated assuming saturation of the lacustrine sediments, taking into account water loss in the laboratory subsequent to core extraction (Lowick and Preusser, 2009). Substantial uncertainties in water content were included in order to mitigate potential inaccuracies in the dose rate calculations as a result of desiccation. Cosmic-ray dose

rates were calculated based on sediment density estimates, altitude, latitude, longitude, and depth, following (Prescott and Hutton, 1994); these values were negligible (<0.01 Gy/ka) due to the depth within the lake sediments.

**Results**

*Finer-grained samples*

Results were obtained for the six samples measured in the series, and for the coarse-grained sample L20 (A0113). Samples

L18, L20 and L29 (A0112-A0114 respectively) yielded insufficient natural signal for equivalent dose estimation (note that these are the youngest samples according to the depth, which explains why less charge has accumulated to date). The remarkably low sensitivity for feldspars may suggest a primary igneous origin. The remaining samples provided ages, although the luminescence characteristics were not optimal for age determination, as discussed below. Ages could only be calculated by relaxing the acceptance criteria for analysis; the results, however, broadly conform (within 2σ error) to the The polymineral

fine-grained samples from the Orakei core do not exhibit optimal characteristics for the pIR-IRSL290 protocol. The samples are typically very dim, yielding generally low signal counts (Figure B4a) and low sensitivity. Late background subtraction was used to optimize the signal measured. The sensitivity to test dose over the regenerative cycles of the pIR-IRSL290 protocol typically decreased to 20-30% of the test dose sensitivity following the natural signal measurement, and remained fairly constant over subsequent cycles (Figure B4b). Recuperation was very high, yielding values of up to 50%, and recycling ratios

were variable but often exceeding 20% divergence from unity (Table B1). Dose-response curves could often, but not always, be fitted to an exponential or exponential-plus-linear function, and rarely passed through the origin due to the high degree of thermal transfer (Figure B4c). In order to obtain any results at all, aliquots were accepted if dose-response curves could be reasonably fitted; threshold criteria for other quality control measurements such as recycling ratios, recuperation and sensitivity change were (by necessity) relaxed. expected ages based on the projected age model.

Additional quality control measurements were undertaken on sample L15 (A0111), which yielded the largest amount of dateable material and therefore the possibility to make extra aliquots for these tests. A dose recovery test on two aliquots of this sample provided a dose recovery ratio of 1.03 (Fig. B5), although it should be noted that thermal transfer remained high and the applied dose (c. 56 Gy) was substantially lower than the measured equivalent dose (c. 230 Gy) (the magnitude of applied dose can result in different dose recovery behaviours (Doerschner et al., 2016 and references therein)). There was no

appreciable residual dose measured in sample L15 (A0111) (Fig. B5), therefore indicating that correction for inherited prior dose in the less easily bleached pIR signal is not necessary (Buylaert et al., 2012).

Residual dose tests were undertaken on samples L15. Residual dose in the order of 24 Gy was observed in one aliquot of sample L15 (A0111). Negligible residual signal, but also poor dose-response characteristics, were observed in the remaining two aliquots measured. Residual dose corrections of 24 Gy was made on the Orakei samples on the basis of the L15 aliquot,

resulting in younger ages than first calculated. The residual-corrected age results are listed in italics in Table 4 of the main text.

*Coarse-grain sample L20 (A0113)*

The results for the comparative study of different IRSL preheat temperatures for the coarse-grained sample L20 (A0113) are summarized in Table B2.

By comparison with the fine-grained measurements, the coarse-grained sample exhibited better characteristics; namely, a brighter signal, overall better recycling and thermal transfer. However, sensitivity changes between regenerative doses remained substantial. While the equivalent doses were comparable between the aliquots measured with both preheats (Figure B3). Given the small number of aliquots accepted, we cannot confirm if this younger result is systematic or otherwise. Perhaps





most critically, both sets of measurements yield ages that substantially overestimate the predicted age for this sample of c. 70

ka, and overestimate all ages calculated from fine grains. This result may suggest that the coarser grains were not completely

bleached during transport, and therefore suggests two different sediment transport mechanisms for the fine and coarse grains.

Ultimately, we cannot use the coarse-grained results for dating the Orakei core.

**Appendix C: Paleomagnetism**

Additional figures with paleomagnetic data.

**Appendix D: Beryllium**

10-Beryllium extraction from sediment samples at ANSTO:

All chemical reagents used are AR or Ultrapure grade.

1)   Label and weigh empty and cleaned 50ml centrifuge tube (+lid)
2)   Fill known weight (approx. 0.5g) dried, powdered and homogenized sediment into centrifuge tube and weigh
3)   Slowly add 10ml 6M HCl to sediment (Adjust to 20ml 6M HCl for 1g samples).
4)   Weigh together after reaction subsides.
5)   Gently swirl and allow digesting for 2-3 hours.
6)   Centrifuge at 3600rpm for 10 minutes and extract 2ml of supernatant into a labelled and rated 15 ml centrifuge tube and weigh.
7)   Dilute sample in small centrifuge tube with MilliQ water to ca. 10 ml total volume and weigh accurately. This sample is sent for ICP-MS or ICP-AES for measurement of native $^9$Be concentration.
8)   Add 0.5 ml of Be carrier to remaining leachate in large centrifuge tube weighing carrier accurately
9)   Swirl and let stand for ca. 2 hours
10)  Centrifuge for 10 minutes at 3600 rpm and carefully pipette supernatant into new, labelled and cleaned 50 ml
centrifuge tube.
11)  Wash residue with 1 ml MilliQ water (2 ml for 1 g samples).
12)  Centrifuge again and add supernatant to centrifuge tube from step 10. Repeat washing, centrifuging and pipetting.
13)  Fill centrifuge tube with MilliQ water up to volume of 25 ml.

All further steps use the supernatant collected in the new centrifuge tube (see step 10). Sediment residue can be

discarded after successful sample preparation.

The following preparation is identical to the steps used in the standard procedure for in situ samples (Child et al.,

2000; White et al., 2019).


10-Beryllium extraction from sediment samples at Lund University/EAWAG method:

1)   Add 9Be carrier to dried and homogenised sample in large beaker.
2)   Add 1 ml 30% H2O2, 1 ml 37% HCl and 4 ml H2O
3)   Mix and wait over night
4)   Centrifuge and decant into beaker
5)   Add 2 ml 37% HCl and 2 ml H2O to centrifuge tube, centrifuge and decant into same beaker. Discard precipitate
6)   Add 1 ml 65% HNO3 and 1 ml 37% to beaker. Mix and wait until sample dissolved.
7)   Transfer solution into small centrifuge tubes.
8)   Add 2-4 ml 25% ammonia solution to reach hydroxide precipitation at pH 10. Mix and wait until precipitation.
9)   Centrifuge and discard solution, keeping the precipitate. Wash sample and repeat.
10)  Add 4 ml 40% NaOH (fresh!) to sample and wait >5h. Removal of Fe at pH 14.
11)  Centrifuge and decant into small centrifuge tube discarding the precipitate.
12)  Add 2-4 ml 37% HCl to solution to pH3. Wait 30 min at room temperature. Centrifuge and decant into small centrifuge tubes.
13)  Add 2-4 ml 25% ammonia solution to precipitate 10Be at max pH 10. Wait 1-2 h, then centrifuge and wash sample twice.





14) Transfer precipitate to quartz crucible and dry.
15) Oxidize to BeO in muffle furnace (110°C, 600°C).
16) Press with Nb into AMS cathode.


**Data availability**

All datasets will be made available on Pangaea.de after publication. The doi will be provided as soon as it is available.

**Author contributions**

**Leonie Peti:** Formal Analysis, Funding acquisition (AINSE), Investigation, Visualization, Writing – original draft; **Kathryn Fitzsimmons:** Investigation, Supervision, Validation, Writing – review & editing; **Jenni Hopkins:** Investigation, Writing – review & editing; **Andreas Nilsson:** Methodology, Supervision, Writing – review & editing; **Toshiyuki Fujioka:** Supervision, Validation, Writing – review & editing; **David Fink:** Supervision, Writing – review & editing; **Charles Mifsud:** Methodology, Writing – review & editing; **Marcus Christl:** Investigation, Methodology, Writing – review & editing; **Raimund Muscheler:** 830 Supervision, Validation, Writing – review & editing; **Paul Augustinus:** Conceptualization, Funding acquisition (Marsden), Supervision, Writing – review & editing

**Competing interests**

The authors declare no competing interests.

**Acknowledgements and funding**

LP thanks Hannah Marley for tephra, luminescence, radiocarbon and paleomagnetic sampling support, Phil Shane for discussing tephras and Craig Woodward for a discussion about age models.
The Orakei Basin drilling campaign in 2016 was co-funded by DEVORA (funded by the New Zealand Earthquake Commission and the Auckland Council and led by Jan Lindsay, University of Auckland, and Graham Leonard, GNS Science), and a grant from the Royal Society of New Zealand Marsden Fund (UOA1415 to Paul C. Augustinus). The paleomagnetic 840 work at Lund University was funded by the same Marsden Fund UOA1415. We acknowledge the financial support for the Centre for Accelerator Science, at ANSTO, through the Australian National Collaborative Research Infrastructure Strategy (NCRIS). LP would like to thank AINSE Limited for providing financial assistance (Award – PGRA 12196) to enable work at the ANSTO Centre for Accelerator Science (radiocarbon dating and 10-Beryllium). This work was partially supported by the Swedish Research Council (grant DNR2013-8421 to RM).

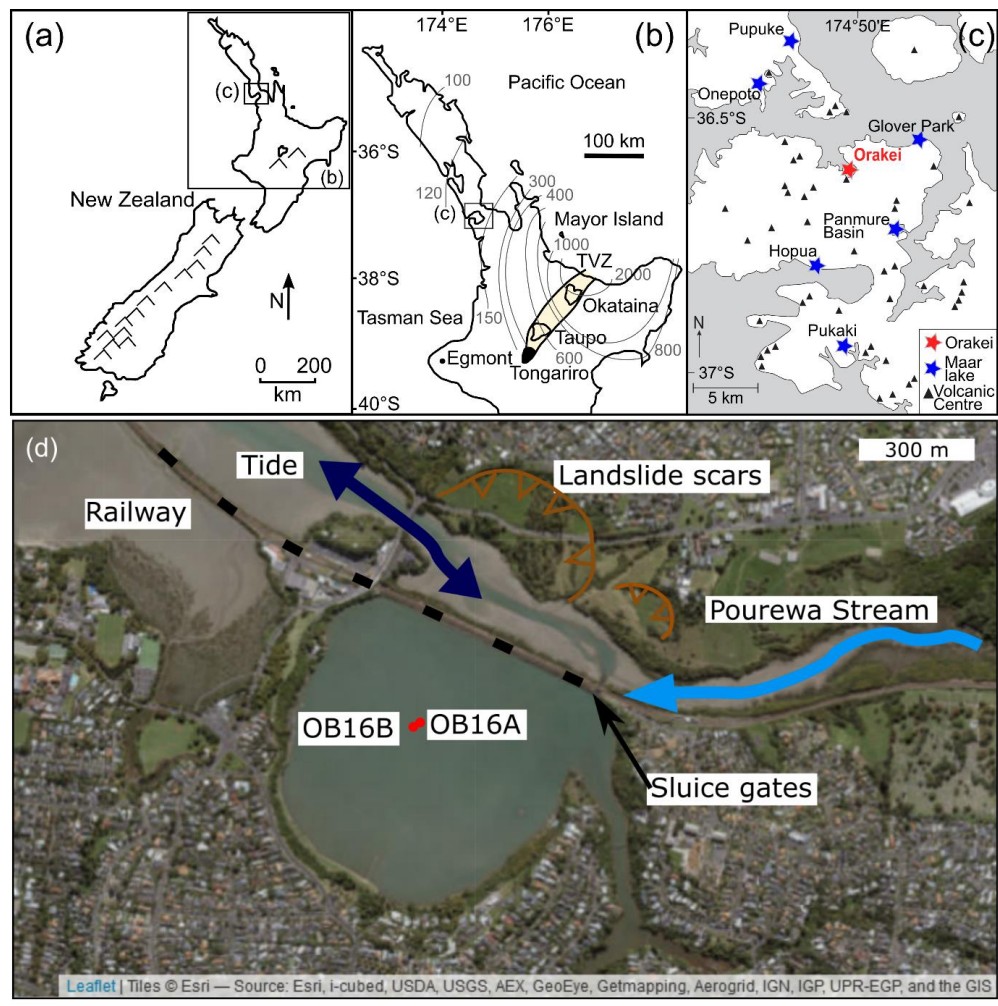

Figure 1: Map of the Orakei maar study area. (a) New Zealand with insets (b) and (c) marked. (b) New Zealand's North Island with Auckland (inset c) and the major volcanic source centres marked, the extent of the TVZ is shaded in yellow. Thin grey lines and numbers are isopachs (in mm) of the widespread Rotoehu tephra (modified from Lowe (2011)) highlighting the importance of the TVZ tephras as age markers in sediment records from New Zealand's North Island. (c) Auckland Volcanic Field showing the position of Orakei Basin. (d) Satellite image (ESRI world imagery) show the coring locations with red dots and other important features of the surroundings of the maar. Modified after Hopkins et al. (2017) and Peti and Augustinus (2019).



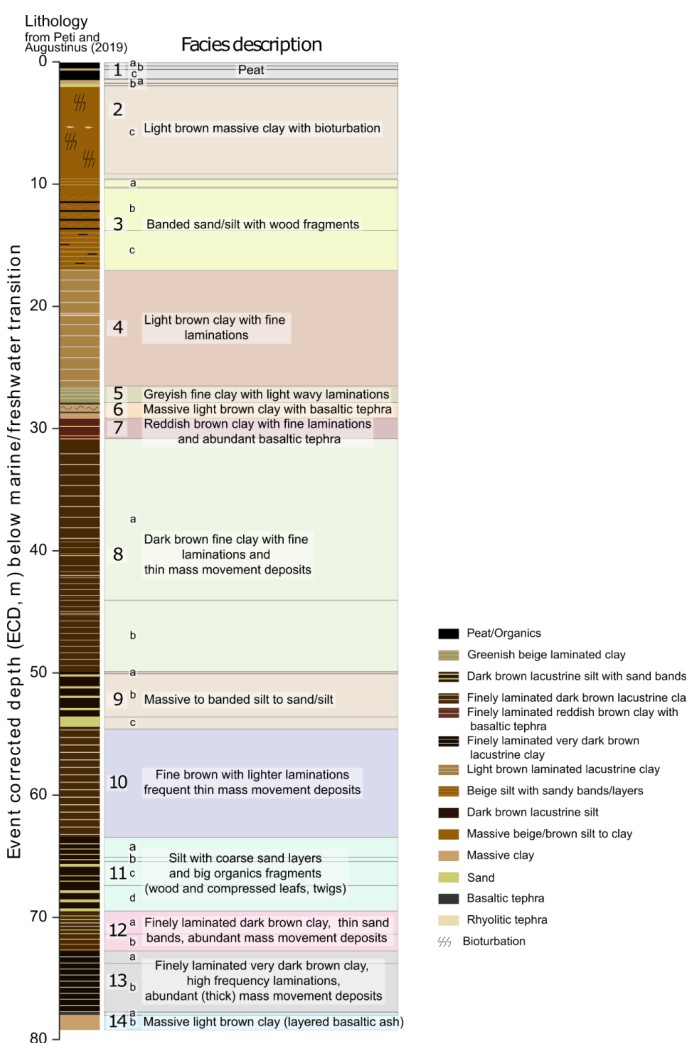

**Figure 2: Simplified lithology and summarised facies unit description of the Orakei sediment sequence. After Peti and Augustinus**
**(2019).**





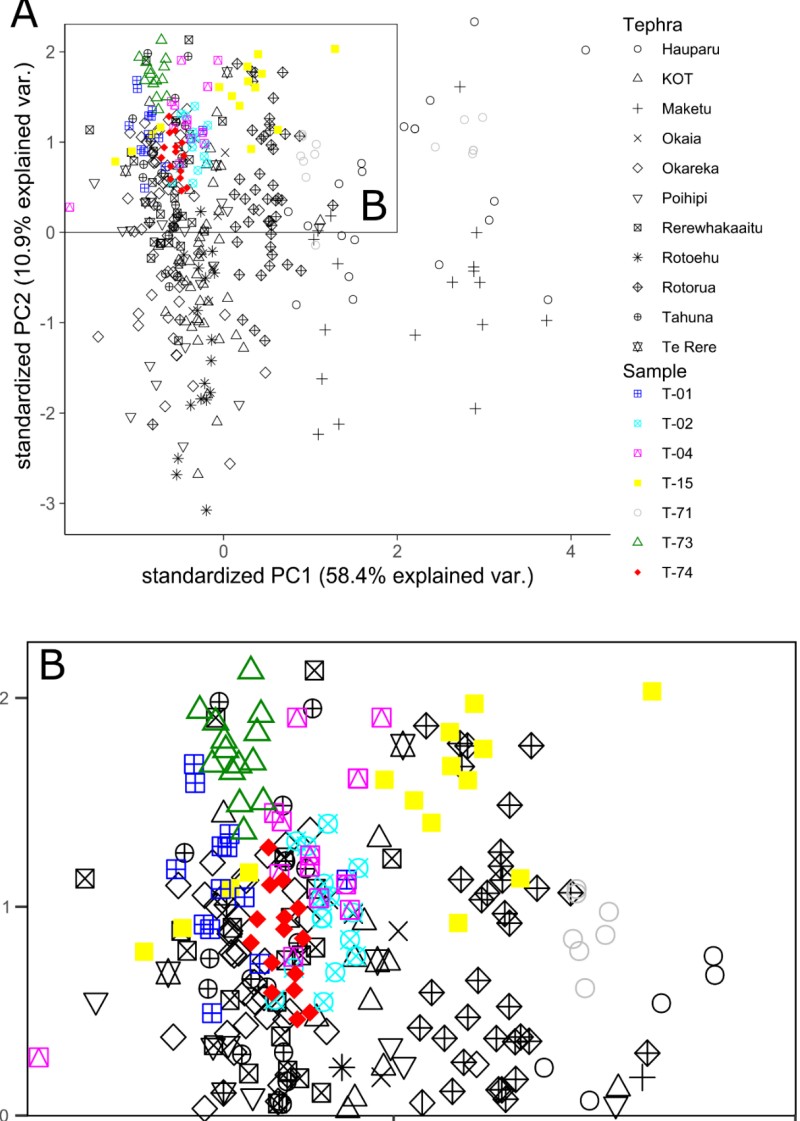

**Figure 3: A: PCA individuals of reference tephra EMPA data (in black symbols from (Lowe et al., 2013; Molloy et al., 2009)) and Orakei samples (in coloured symbols). B: Enlarged inset from A. Symbols and references as in A.**

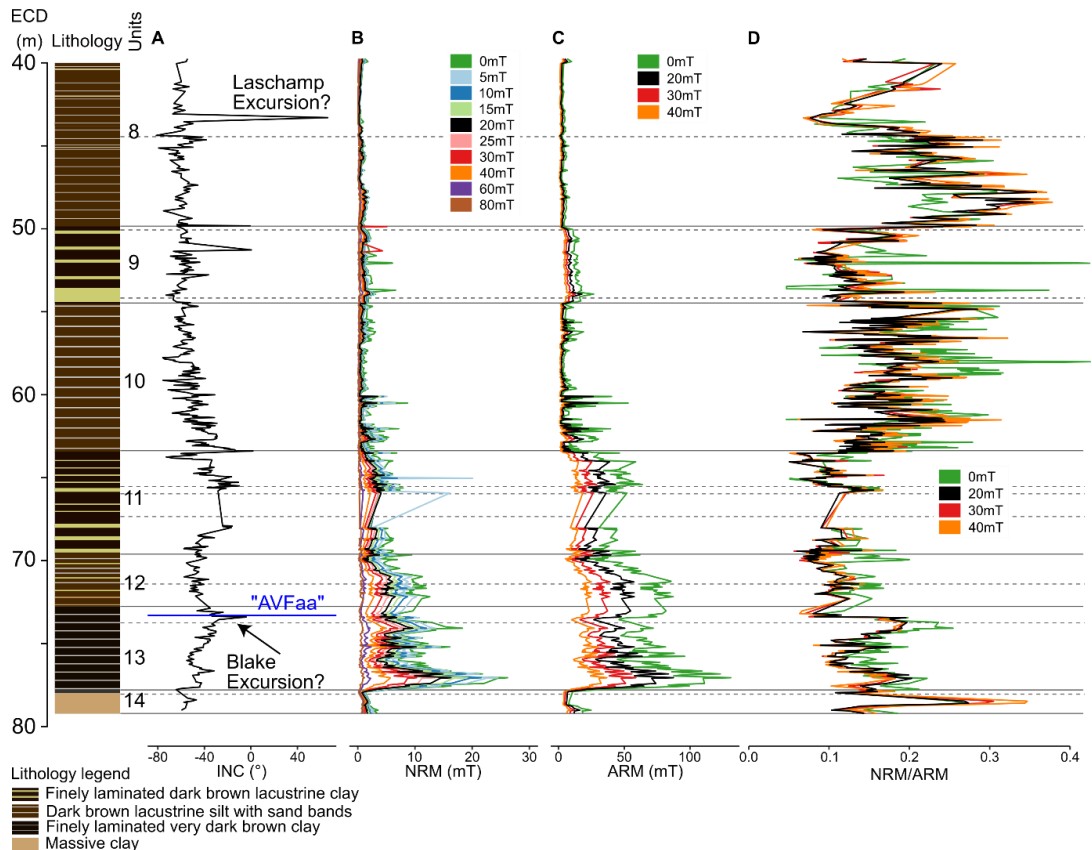

**Figure 4: (A) Inclination, (B) NRM at different demagnetisation steps, (C) ARM at different demagnetisation steps and (D) NRM/ARM at different demagnetisation steps of the Orakei sediment sequence. Horizontal gray lines mark facies unit boundaries (solid) and sub-unit boundaries (dashed) from Peti and Augustinus (2019).**



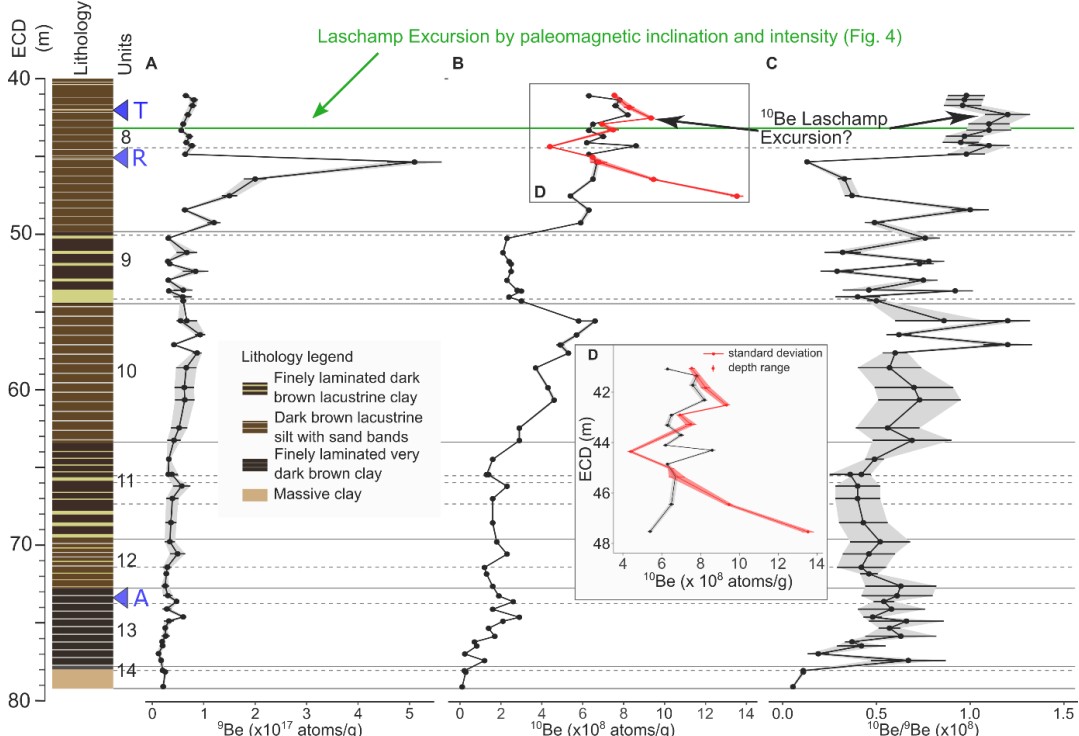

**Figure 5: Beryllium variation in the lower ~40 m of the Orakei sediment sequence alongside the simplified lithology and facies units**
**boundaries (solid grey lines) and sub-unit boundaries (dashed grey lines) from Peti and Augustinus (2019). Green line marks the**
**position of the Laschamp Excursion according to the paleomagnetic inclination and intensity (see Fig. 4 and section 4.4). Blue**
**triangles mark positions of the tephra marker layers T: Tahuna (37,485 ± 932 cal yr BP), R: Rotoehu (45,100 ± 3,300 yr), A: "AVFaa"**
**from Mt. Albert (119,200 ± 2,800 yr). A: $^9$Be concentration ± 1 standard error; B: $^{10}$Be concentration ± 1 standard error from Lund**
**(red) and ANSTO (black), enlarged interval in D; C: $^{10}$Be/$^9$Be ratio ± 1 standard error. Grey/red shading to highlight the uncertainty**
**interval. Where error bars are not visible, they are within the size of the datapoints.**



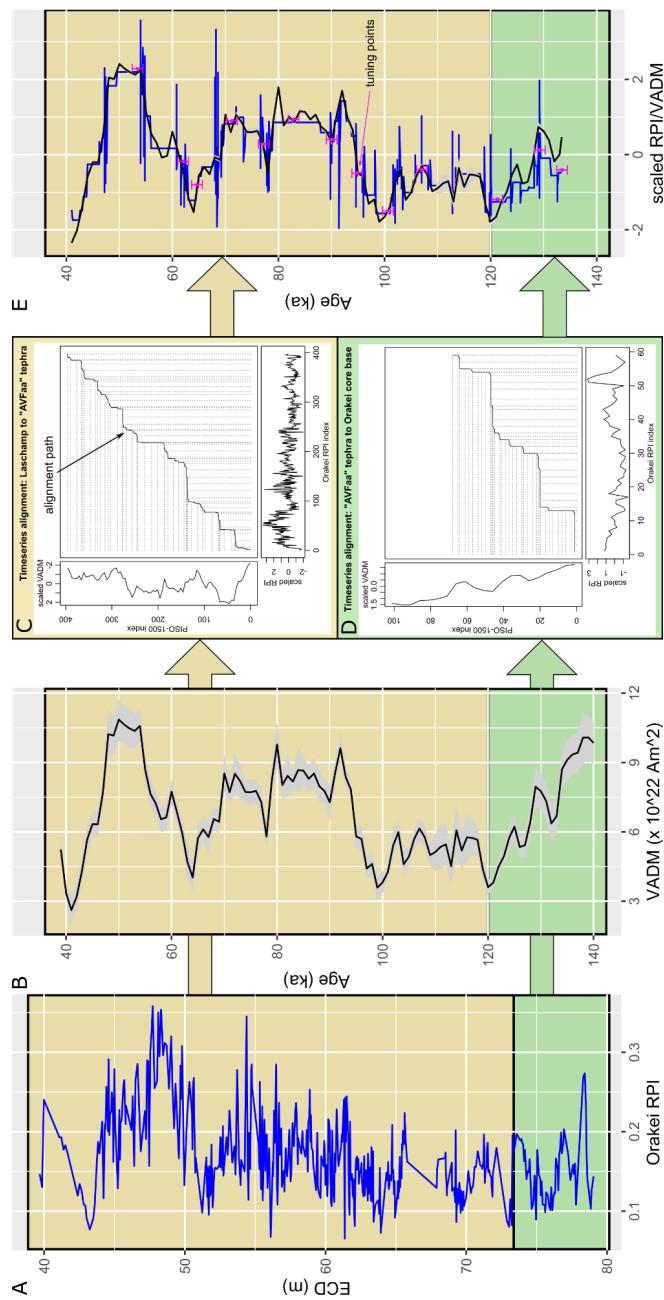

**Figure 6 [Landscape]: A: Orakei RPI (NRM/ARM at 20mT), B: PISO-1500 (Channell et al., 2009), C: DTW alignment path (solid black line) for the upper core section (Laschamp Excursion to "AVFaa" tephra, orange background in A, B) with scaled Orakei RPI as the query curve below and scaled PISO-1500 VADM as the reference curve to the left. Dotted grey lines highlight aligned points between both curves. D: Same as C for the lower core section ("AVFaa" tephra to the Orakei core base, green background in A, B). E: Both curves scaled after DTW alignment on the same age scale**





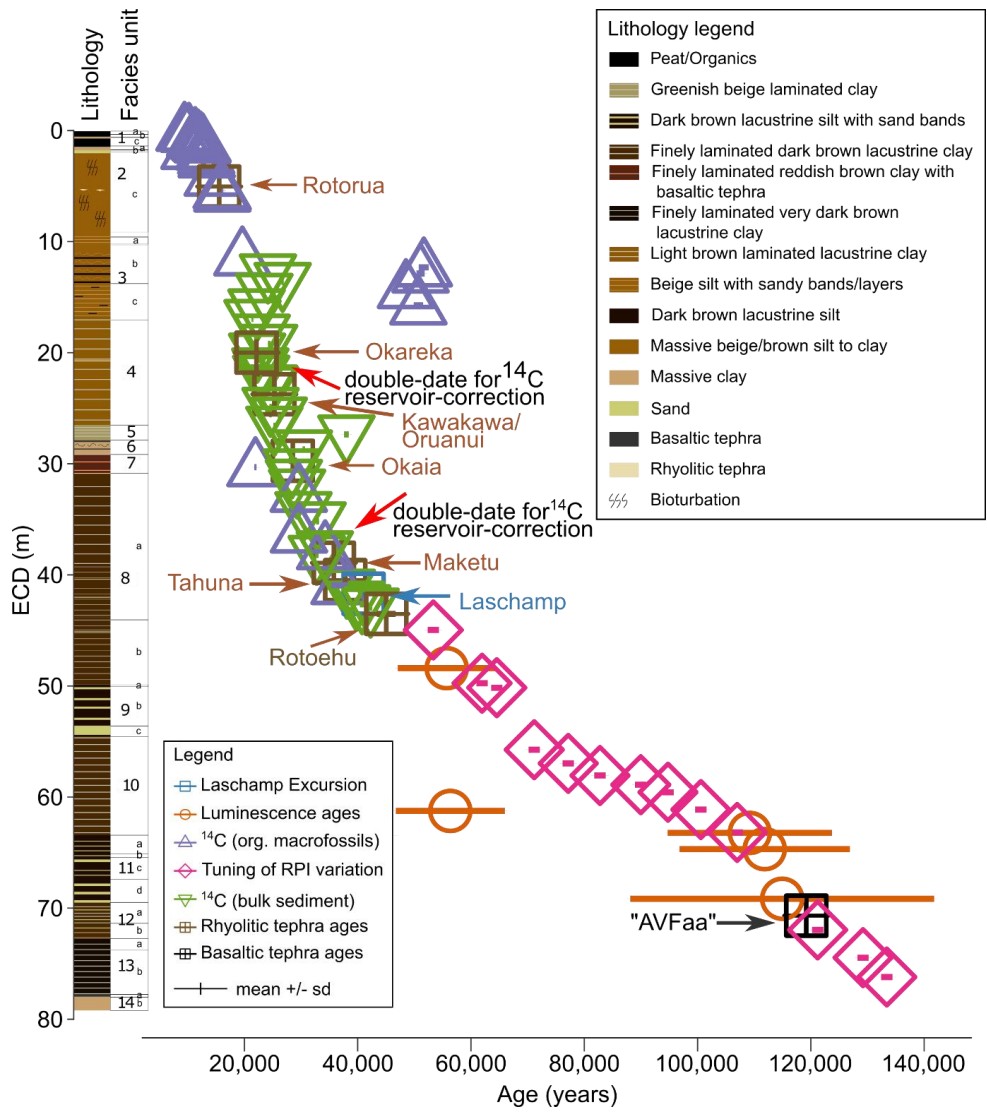

**Figure 7: Simplified lithology and summarised facies units of the Orakei sediment sequence (from Peti and Augustinus (2019)) with**

**ages used in the current chronology study (identified tephras marked with names – for published and adjusted (this study) ages see**

**Table 2). Red arrows mark double dates undertaken for [14]C reservoir-correction (see section 3.2). Note that radiocarbon dates have**

**been calibrated and reservoir-corrected where applicable as noted in Table 1.**







**Figure 8 [Landscape]: Orakei age model spanning the entire sediment sequence using the event-corrected depth obtained by Bacon age modelling. Red dotted line marks the mean age model, grey dotted lines mark the 95% confidence limits. Inset enlarges the interval ca. 45,000 cal yr BP. Blue symbols mark calibrated radiocarbon ages, brown symbols mark rhyolitic tephra ages (marked by name), green symbol marks the age of the Laschamp Excursion, orange symbols mark luminescence ages (marked by sample codes), pink symbols mark tuning points from DTW alignment of Orakei RPI to PISO-1500 VADM, the black symbol marks the age of the basaltic "AVFaa" tephra identified as sourced from the Mt. Albert volcano. Lithology and facies units from Peti and Augustinus (2019).**





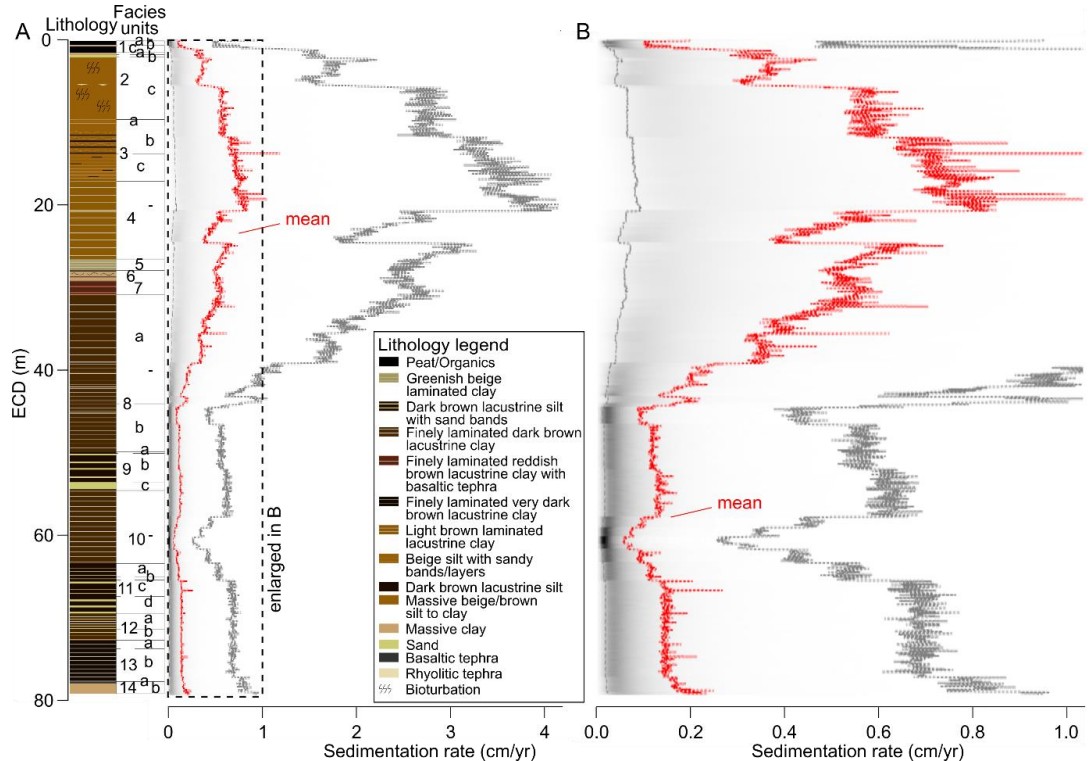

**Figure 9: A: Sedimentation rate variation of the Orakei sediment sequence in cm/yr from Bacon (Blaauw and Christen, 2020); B: Enlarged 0-1 cm/yr interval from A. Red dotted line marks the mean sedimentation rate, grey dotted lines mark the 95% confidence limits obtained by Bacon. Simplified lithology and facies units from Peti and Augustinus (2019).**





**Figure A1: Major element oxide bivariate plots of reference tephra data (black) and Orakei samples (colourful points). Marked insets are enlarged on the right hand side.**



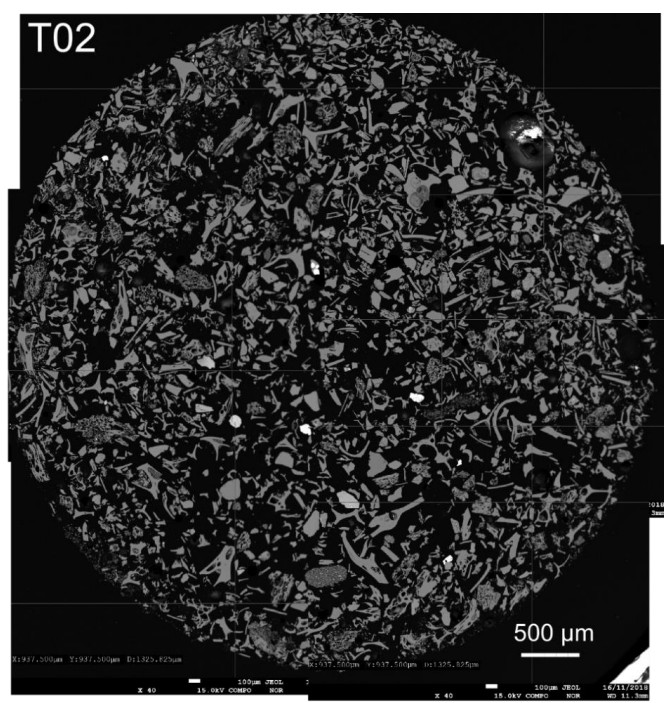

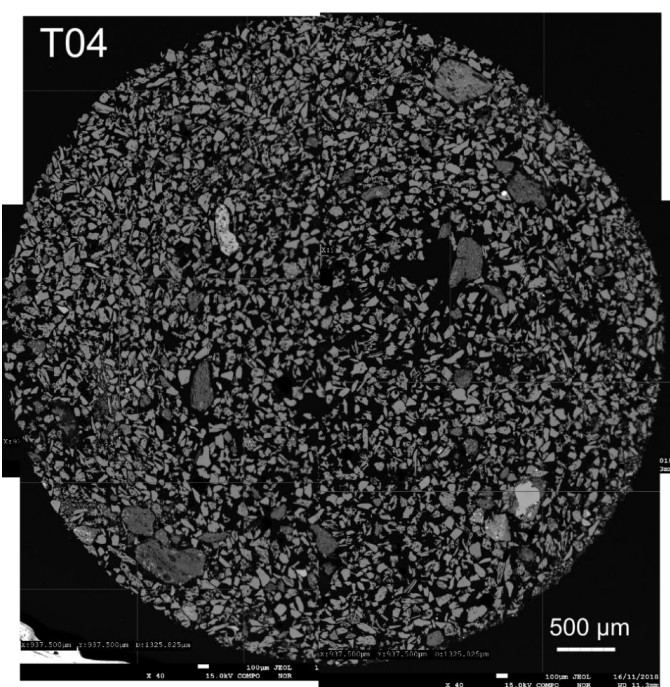

**Figure A2: Backscatter images of samples T02 and T04. The large, cuspate shards with preserved thin bubble walls in T02 signify a very large, very explosive eruption, in comparison the small, blocky shards in T04, suggesting a smaller less explosive eruption.**





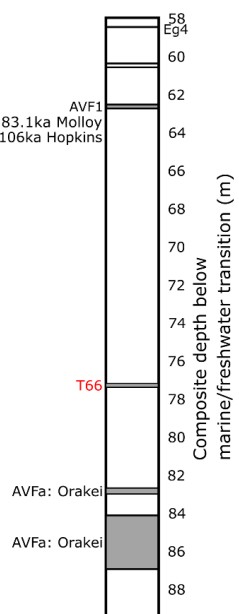

**Figure A3: Lower section of the Orakei 2016 composite sequence highlighting the position of the T66 layers between the stratigraphically matched position of the AVF1 horizon and the primary eruptive material from the Orakei Volcanic Centre correlated to tephra layer AVFa by (Hopkins et al., 2017).**

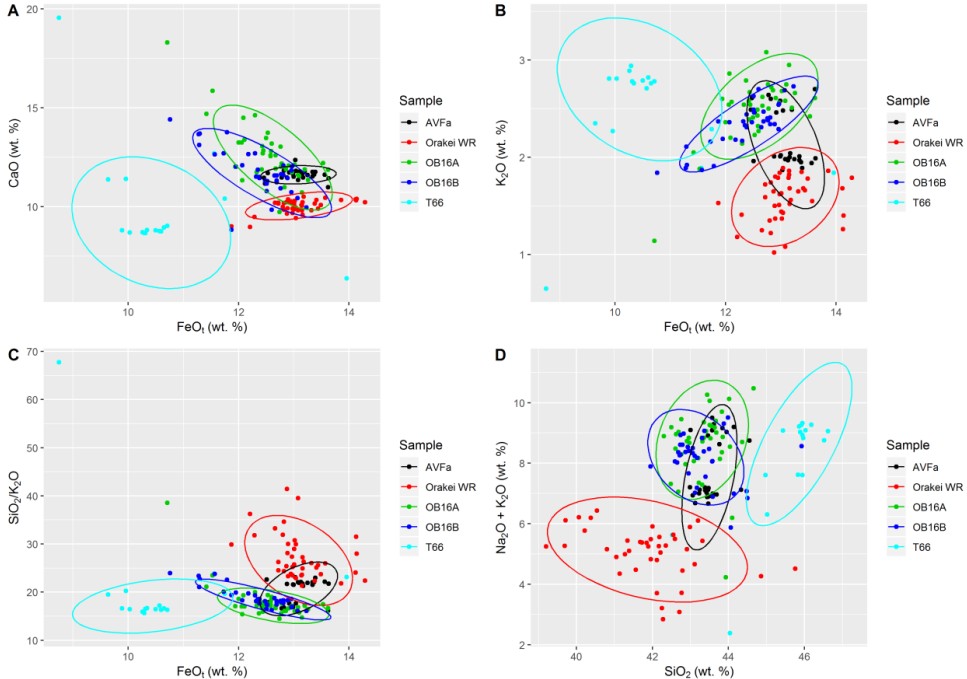

**Figure A4: Comparison of sample T66 to AVFa (tephra layer from Glover Park core; correlated to the Orakei eruptive centre, (Hopkins et al., 2017)), Orakei whole rock (WR) data as well as eruptive material from Orakei in the A and B cores of this study (OB16A, OB16B). Note that T66 overlaps with neither of these samples suggesting that this is a tephra layer from a different eruption. Ellipses follow a multivariate t-distribution (Fox and Weisberg, 2011).**





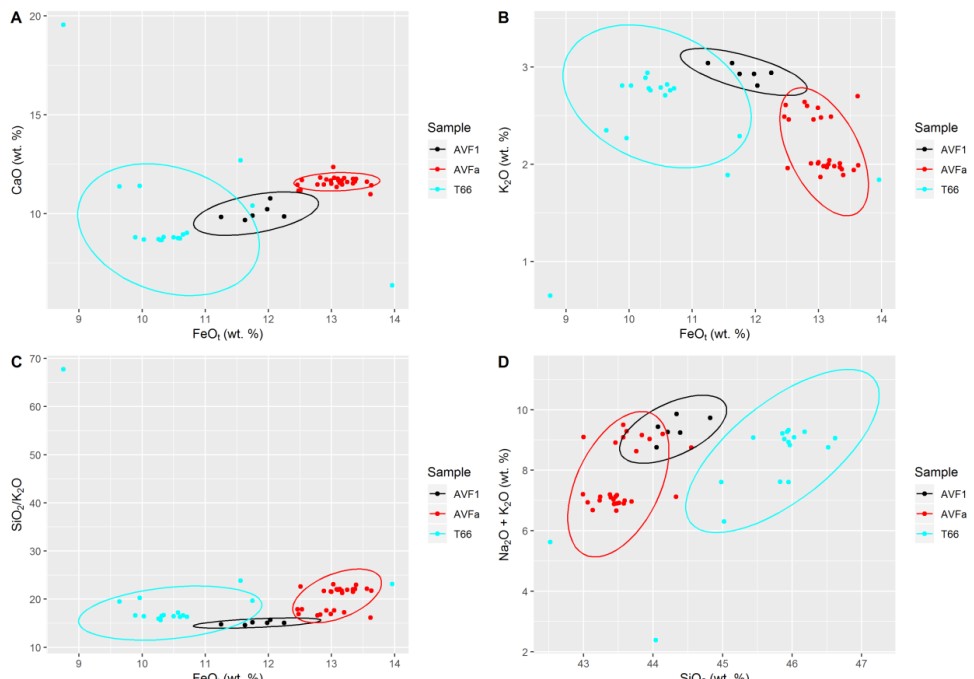

**Figure A5:**
**Comparison of sample T66 to AVF1 (tephra layer in Orakei 2007 core** (Hopkins et al., 2015; Molloy et al., 2009) **correlated to Domain:** **(Hopkins et al., 2017)) and AVFa (tephra layer from Glover Park core correlated to the Orakei Volcanic Centre** (Hopkins et al., 2015, 2017)**). Note that T66 overlaps with neither layer and thus is considered an eruption not previously found in the AVF maar cores. Ellipses follow a multivariate t-distribution (Fox and Weisberg, 2011).**


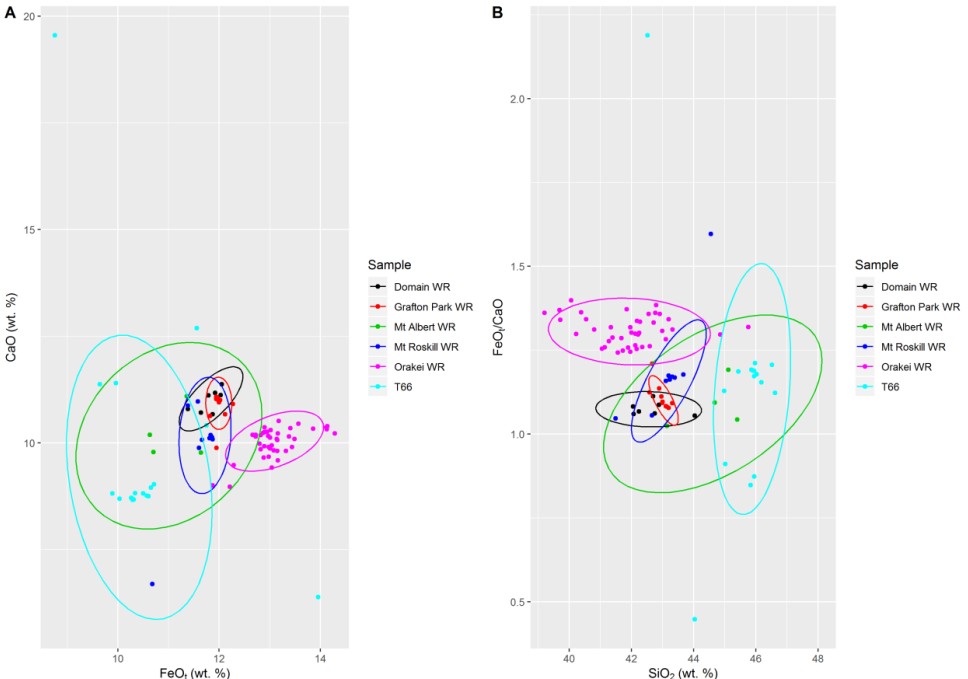

**Figure A6: Matching of T66 to possible source volcanic centres (WR: whole rock data) for this tephra layers. Despite complications of correlating glass shard EMPA chemistry to whole rock geochemistry, the most likely source volcanic centre candidate for tephra**





**T66 is Mt. Albert as only the data and 95% confidence ellipse of Mt Albert WR data overlaps with the T66 tephra EMPA data. Ellipses follow a multivariate t-distribution (Fox and Weisberg, 2011).**

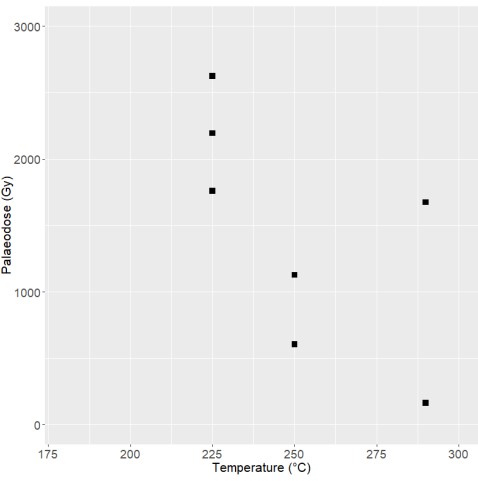

**Figure B1: Preheat plateau for fine-grained sample L15 (A0111), showing substantial scatter between aliquots measured with the**
**same preheat temperature, as well as between all aliquots.**

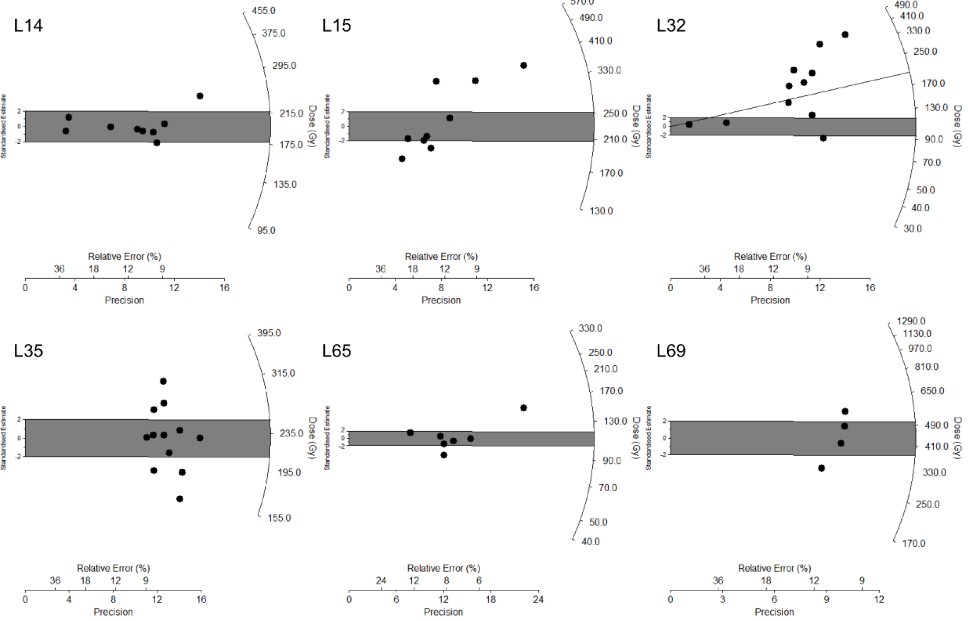

**Figure B2: Radial plots showing the dose distributions of accepted aliquots for all fine-grained samples, as well as the calculated central age (shaded area). In the case of L32 (A0115), the shaded area corresponds to the minimum age calculated using the MAM, and the line corresponds to the equivalent dose which would yield the expected age according to a preliminary core age model.**



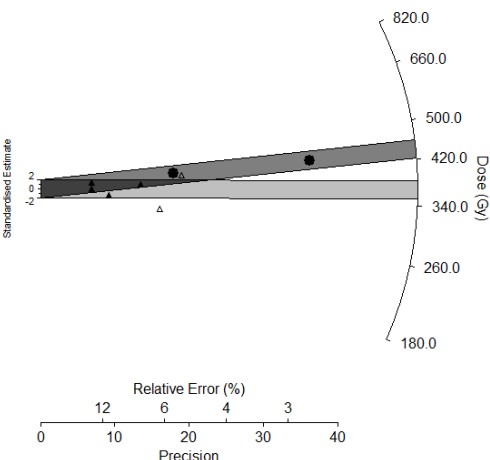


**Figure B3: Radial plot showing the doses of accepted aliquots for the coarser-grained sample L20 (A0113). Aliquots run using the IR50 preheat are shown as black circles, with the darker grey shaded area; those run using the higher temperature preheat are shown as open triangles and the CAM age shaded in lighter grey.**

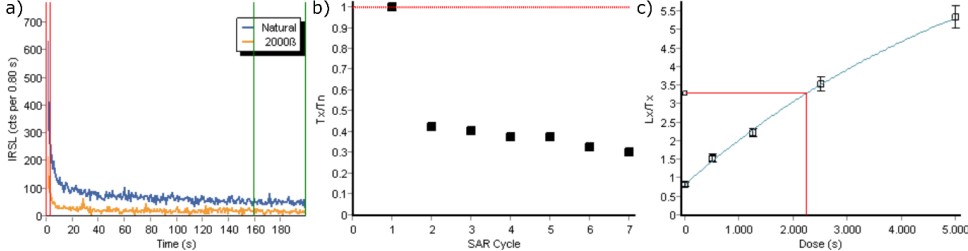

**Figure B4: Illustration of luminescence characteristics of the Orakei core sediments using the post IR-IRSL$_{290}$ protocol: a) typical natural postIR-IRSL$_{290}$ signal decay for dateable sample L14 (A0110); b) change in sensitivity response to test dose over the regenerative dose cycles for sample L35 (A0116), showing reduction in effective charge transfer throughout the protocol in the typical sample; c) dose-response curve for representative aliquot of sample L35 (A0116), showing exponential fit but high Lx/Tx value for the zero dose step illustrating high degree of thermal transfer in the sediments.**

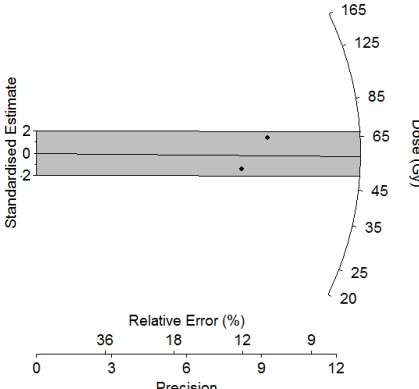


**Figure B5: Dose-recovery distribution from two aliquots of sample L15 (A0111). The shaded region corresponds to the standard deviation of the measured CAM dose; the line corresponds to the applied radiation dose.**







**Figure C1: Orthogonal plots and intensity of magnetisation after step-wise demagnetisation of pilot samples OB16B42-51**





**Figure C2: Orthogonal plots and intensity of magnetisation after step-wise demagnetisation of pilot samples OB16B52-61**





**Figure C3: Orthogonal plots and intensity of magnetisation after step-wise demagnetisation of pilot samples OB16B63-75**





Figure C4: Orthogonal plots and intensity of magnetisation after step-wise demagnetisation of pilot samples OB16B76-84





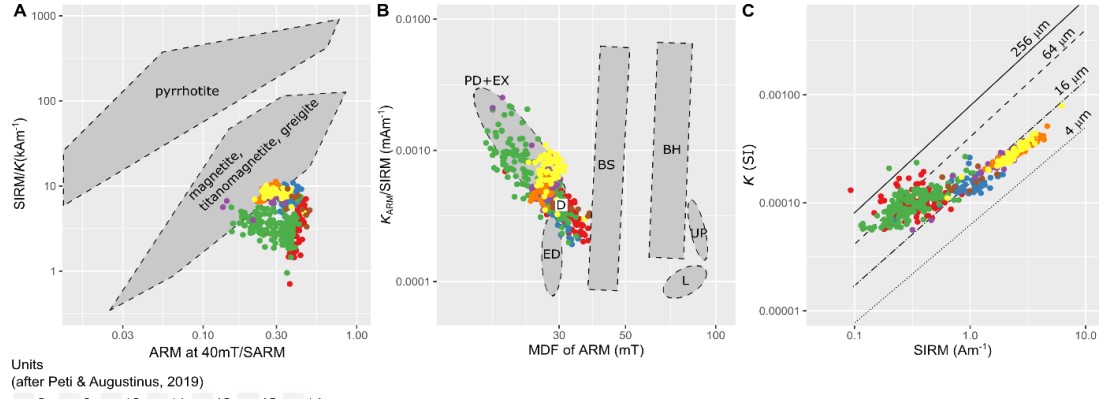

**Figure C5: Rockmagnetic characterisation of the Orakei sediment samples. (A) Qualitative identification of magnetic minerals following Peters and Thompson (1998). SARM: Saturated ARM at 0 mT. We note that the ARM at 40mT/SARM ratios are likely higher than the equivalent measurements of Peters and Thompson (1998) due to the lower maximum AF (80mT) used to induce the**

**ARM. The magnetic carrier in the Orakei sediment sequence is interpreted as dominantly magnetite/titanomagnetite. (B) Magnetic components of Orakei samples in comparison to components defined by Egli (2004). MDF: Mean destructive field. $\kappa_{ARM}$: Susceptibility of ARM (SARM divided by applied bias field). BH: Biogenic hard, high-coercivity magnetosomes, BS: Biogenic soft, low-coercivity magnetosomes, D: detrital particles transported in water systems, ED: eolian dust/wind-blown particles, EX: ultrafine extracellular magnetite, L: maghemite component in loess, PD: pedogenic magnetite, UP: atmospheric particulate matter produced**

**by urban pollution. Orakei samples are classified as detrital and extracellular components. (C) Estimated magnetic grain size indicator following Thompson and Oldfield (1986, p.31). SIRM: Saturated IRM at 0mT. Most of the samples plot in the range of pseudo-single domain magnetite (0.1-20 µm). The increased scatter in the lower left half of the plot, suggesting increasing magnetic grain size, may partly reflect the importance of superparamagnetic (e.g. facies 10) and/or paramagnetic effects in samples with low magnetite concentration.**

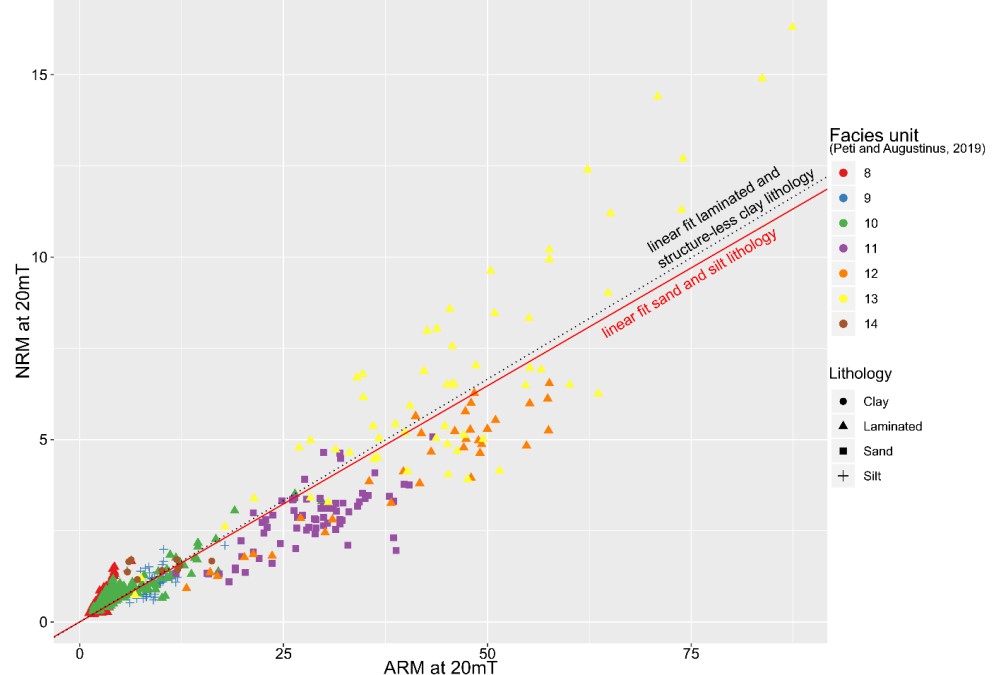


**Figure C6: Orakei ARM vs NRM at 20mT. Note how datapoints in coarser units with sandy and silty lithology follow a linear trend with a slightly shallower slope (red, solid) than clay-dominated lithologies (black dotted; laminated or structure-less).**







**Figure C7: Comparison of Orakei RPI (left) and PISO VADM (right) after the age modelling process in Bacon. Note the good alignment but small offsets are caused by the consideration of absolute ages for the "AVFaa" tephra and luminescence ages as well as the smoothness of the age model introduced by Bacon, which does not always follow the palaeomagnetic tuning points as established by dynamic time warping.**



**Table 1: Bacon prior parameters**

| Accumulation mean (acc.mean) (cm/year) | Accumulation shape (acc.shape) | Memory strength (mem.strength) | Memory mean (mem.mean) | thick (cm) |
|---|---|---|---|---|
| 15 | 1.5 | 4 | 0.7 | 5 |


**Table 2: Identified rhyolitic tephra layers in the Orakei sediment sequence. For source locations see figure 1.**

| Orakei sample code | Tephra name | ECD (m) | Thickness (mm) | Age (cal yr BP) ± 2σ | Source volcanic | Original dating method | Reference |
|---|---|---|---|---|---|---|---|
| T-15 | Rotorua | 5.23 | 20 | 15,635 ± 412 | Okataina | combined $^{14}$C ages above, below and within the tephra layer | (Lowe et al., 2013) |
| T-01 | Okareka | 20.74 | 1 | 22,213 ± 278 | Okataina | $^{14}$C of leaf fragments 10 mm above and below tephra | Original age from Molloy et al. (2009) calibrated with SHCal13 (Hogg et al., 2013). See Appendix A. |
| T-02 | Kawakawa/ Oruanui | 24.62 | 25 | 25,360 ± 160 | Taupo | 22 macrofossil $^{14}$C ages from four sites incorporated in Bayesian model | (Vandergoes et al., 2013) supported by an ice core age (Dunbar et al., 2017) |
| T-04 | Okaia | 30.84 | 5 | 28,621 ± 1,428 | Taupo | $^{14}$C ages | (Lowe et al., 2013) |
| T-71 | Maketu | 40.37 | 18 | 35,866 ± 738 | Okataina | $^{14}$C of bulk sediment and macrofossil charcoal | Original age from (Molloy et al., 2009) freshwater-reservoir effect corrected (for bulk sediment age, see section 3.2, 4.2), and calibrated with SHCal13 (Hogg et al., 2013). See Appendix A. |
| T-73 | Tahuna | 41.99 | 2 | 37,854 ± 940^ | Taupo | $^{14}$C of bulk sediment and macrofossil charcoal | Original age from (Molloy et al., 2009) freshwater-reservoir effect corrected (for bulk sediment age, see section 3.2, 4.2), and calibrated with SHCal13 (Hogg et al., 2013). See Appendix A. |
| T-74 | Rotoehu | 45.14 | >300 (partially reworked) | 45,100 ± 3,300 47,400 ± 3,000 | Okataina | $^{238}$U/$^{230}$Th disequilibrium and (U-Th)/He zircon $^{40}$Ar/$^{39}$Ar | (Danišík et al., 2012)* (Flude and Storey, 2016) |





^ We note that Loame et al. (2019) dated the Tahuna layer to 38,870 ± 1,106 cal yr BP. However, we prefer an age constrained by samples above and below the tephra layer (Molloy et al., 2009) but highlight the agreement between both ages.

* Many ages for the Rotoehu tephra (Rotoiti eruption) have been proposed based on a variety of direct and indirect dating methods spanning approximately the age range 40 – 60 ka. The combined $^{238}U/^{230}Th$ disequilibrium and (U-Th)/He zircon age of 45,100 ± 3,300 years, concordant with radiocarbon ages (Danišík et al., 2012) is in better agreement with the radiocarbon ages of this study (see Table 3) and thus used in the presented chronology.

**Table 3: Overview of radiocarbon samples and ages. Reservoir-corrected ages in italics. Outliers in grey. All are calibrated by SHCal13 (Hogg et al., 2013).**

| Lab Code | Material | ECD (m) | Facies unit (from Peti and Augustinus (2019)) | $^{14}C$ age (years BP ± 1σ) | Reservoir correction (years ± 2σ) | Calibrated mean ± 1σ age (cal. yr BP) (reservoir corrected where required) |
|---|---|---|---|---|---|---|
| NZA28865*^ | Wood | 0.12 | 1a (Peat) | 8,565 ± 30 | | 9,512 ± 19 |
| OZW872 | Wood | 0.22 | 1a (Peat) | 9,175 ± 40 | | 10,303 ± 61 |
| OZW871 | Wood | 0.62 | 1c (Peat) | 10,000 ± 40 | | 11,405 ± 109 |
| OZW870 | Wood | 0.70 | 1c (Peat) | 9,995 ± 40 | | 11,396 ± 107 |
| OZW869 | Wood | 0.71 | 1c (Peat) | 10,065 ± 45 | | 11,519 ± 122 |
| OZW868 | Wood | 0.85 | 1c (Peat) | 10,265 ± 45 | | 11,908 ± 94 |
| OZW876 | Wood | 1.04 | 1c (Peat) | 10,505 ± 45 | | 12,351 ± 142 |
| OZW875 | Wood | 1.20 | 1c (Peat) | 10,815 ± 50 | | 12,700 ± 27 |
| OZW874 | Wood | 1.81 | 2b (Sand) | 9,130 ± 510 | | 10,380 ± 714 |
| OZW873 | Wood | 1.87 | 2b (Sand) | 11,150 ± 45 | | 12,965 ± 73 |
| OZW878 | Wood | 2.27 | 2c (Clay) | 11,535 ± 45 | | 13,342 ± 55 |
| OZW877 | Wood | 2.60 | 2c (Clay) | 11,645 ± 45 | | 13,430 ± 58 |
| OZW879 | Wood | 4.32 | 2c (Clay) | 12,445 ± 50 | | 14,488 ± 183 |
| OZW882 | Wood | 5.49 | 2c (Clay) | 13,405 ± 50 | | 16,074 ± 100 |
| OZW881 | Wood | 5.63 | 2c (Clay) | 13,530 ± 50 | | 16,237 ± 105 |
| OZW880 | Wood | 5.79 | 2c (Clay) | 13,495 ± 50 | | 16,187 ± 101 |
| OZW883 | Wood | 11.76 | 3b (Sand) | 16,370 ± 60 | | 19,718 ± 110 |
| OZW885 | Wood | 12.78 | 3b (Sand) | 51,590 ± 860 | | 51,733 ± 889 |
| OZW884 | Wood | 13.35 | 3b (Sand) | 51,050 ± 710 | | 51,147 ± 726 |
| OZX869 | Bulk sed. | 13.35 | 3b (Sand) | 21,090 ± 110 | 1026 ± 117? | *24,235 ± 208* |
| OZX889 | Bulk sed. | 14.14 | 3c (Sand/Silt) | 23,690 ± 130 | 1026 ± 117? | *26,822 ± 190* |
| OZW886 | Wood | 14.81 | 3c (Sand/Silt) | 48,520 ± 540 | | 48,575 ± 547 |
| OZX888 | Bulk sed. | 15.16 | 3c (Sand/Silt) | 20,540 ± 100 | 1026 ± 117? | *23,604 ± 194* |
| OZW887 | Wood | 16.33 | 3c (Sand/Silt) | 50,640 ± 810 | | 50,767 ± 834 |
| OZX870 | Bulk sed. | 16.61 | 3c (Sand/Silt) | 19,320 ± 130 | 1026 ± 117? | *22,209 ± 206* |
| OZX349 | Bulk sed. | 17.60 | 4 (Clay) | 18,690 ± 60 | 1026 ± 117 | *21,502 ± 164* |





| OZX871 | Bulk sed. | 18.97 | 4 (Clay) | 20,120 ± 90 | 1026 ± 117 | *23,120 ± 190* |
|---|---|---|---|---|---|---|
| OZX887 | Bulk sed. | 19.92 | 4 (Clay) | 21,270 ± 110 | 1026 ± 117 | *24,441 ± 209* |
| OZX348 | Bulk sed. | 20.74 | 4 (Clay) | 19,430 ± 60 | 1026 ± 117 | *22,329 ± 161* |
| OZX872 | Bulk sed. | 22.12 | 4 (Clay) | 19,920 ± 100 | 1026 ± 117 | *22,889 ± 198* |
| OZX886 | Bulk sed. | 22.80 | 4 (Clay) | 20,040 ± 90 | 1026 ± 117 | *23,029 ± 191* |
| OZX885 | Bulk sed. | 23.72 | 4 (Clay) | 21,280 ± 100 | 1026 ± 117 | *24,452 ± 202* |
| OZX350 | Bulk sed. | 23.88 | 4 (Clay) | 21,480 ± 70 | 1026 ± 117 | *24,674 ± 181* |
| OZX884 | Bulk sed. | 25.81 | 4 (Clay) | 21,490 ± 100 | 1026 ± 117 | *24,683 ± 196* |
| OZX351 | Bulk sed. | 26.71 | 5 (Clay) | 22,980 ± 70 | 1026 ± 117 | *26,140 ± 174* |
| OZX883 | Bulk sed. | 27.57 | 5 (Clay) | 22,170 ± 130 | 1026 ± 117? | *25,368 ± 187* |
| OZX882 | Bulk sed. | 28.39 | 6 (Clay) | 34,650 ± 340 | 1026 ± 117? | *38,079 ± 421* |
| OZX873 | Bulk sed. | 29.78 | 7 (Clay) | 26,920 ± 190 | 410 ± 170? | *30,567 ± 231* |
| OZX340 | Wood | 31.44 | 8a (Clay) | 18,210 ± 140 | | *22,027 ± 184* |
| OZX874 | Bulk sed. | 31.55 | 8a (Clay) | 24,870 ± 150 | 410 ± 170 | *28,467 ± 253* |
| OZX875 | Bulk sed. | 32.53 | 8a (Clay) | 25,180 ± 140 | 410 ± 170 | *28,804 ± 255* |
| OZX876 | Bulk sed. | 33.44 | 8a (Clay) | 25,670 ± 140 | 410 ± 170 | *29,375 ± 303* |
| OZX342 | Wood | 33.83 | 8a (Clay) | 25,560 ± 190 | | *29,676 ± 293* |
| OZX877 | Bulk sed. | 35.54 | 8a (Clay) | 26,820 ± 160 | 410 ± 170 | *30,499 ± 228* |
| OZX344 | Bulk sed. | 36.50 | 8a (Clay) | 29,260 ± 120 | 410 ± 170 | *32,990 ± 261* |
| OZX343 | Wood | 37.63 | 8a (Clay) | 25,580 ± 190 | | *29,704 ± 294* |
| OZX341 | Wood | 39.20 | 8a (Clay) | 30,360 ± 140 | | *34,324 ± 164* |
| OZX347 | Bulk sed. | 39.20 | 8a (Clay) | 30,770 ± 160 | 410 ± 170 | *34,261 ± 237* |
| OZX878 | Bulk sed. | 39.63 | 8a (Clay) | 29,140 ± 200 | 410 ± 170 | *32,843 ± 330* |
| LuS9636* | Wood | 42.45 | 8a (Clay) | 32,400 ± 1200 | | *36,859 ± 1420* |
| OZX879 | Bulk sed. | 42.79 | 8a (Clay) | 34,140 ± 300 | 410 ± 170 | *38,155 ± 421* |
| OZX880 | Bulk sed. | 43.91 | 8a (Clay) | 35,650 ± 330 | 410 ± 170 | *39,808 ± 434* |
| OZX881 | Bulk sed. | 44.24 | 8a (Clay) | 36,740 ± 370 | 410 ± 170 | *40,848 ± 397* |
| OZX346 | Bulk sed. | 44.55 | 8b (Clay) | 38,830 ± 330 | 410 ± 170 | *42,339 ± 294* |

* From correlated Orakei 2007 core

^ From (Hayward et al., 2008)

? Reservoir effect assumed from stratigraphically closest facies unit with calculated reservoir effect (see section 4.2).


**Table 4. Equivalent dose ($D_e$), total dose rate data and pIR-IRSL$_{290}$ age estimates for the Orakei 4-11 μm polymineral fine-grained samples.**

| Sample code | ECD (m) | $D_e$ (Gy) | K (%) | Th (ppm) | U (ppm) | Total dose rate (Gy/ka) | Age (years)* |
|---|---|---|---|---|---|---|---|
| *Lab code* | | | | | | | |
| L65 | 50.21 | 111 ± 10[a] | 0.81 ± 0.04 | 4.27 ± 0.21 | 0.86 ± 0.04 | 1.56 ± 0.16 | 71,200 ± 9,800 |
| *A0118* | | | | | | | *55,760 ± 8,630* |
| L32 | 63.54 | 105 ± 11[b] | 0.77 ± 0.04 | 3.75 ± 0.19 | 0.79 ± 0.04 | 1.44 ± 0.15 | 73,100 ± 10,800 |
| *A0115* | | | | | | | *56,400 ± 9,620* |
| L35 | 65.59 | 230 ± 17[a] | 0.96 ± 0.05 | 5.21 ± 0.26 | 1.08 ± 0.05 | 1.89 ± 0.20 | 122,000 ± 16,000 |



| | | | | | | | |
|---|---|---|---|---|---|---|---|
| *A0116* | | | | | | | *109,230 ± 14,500* |
| L14 | 67.11 | 198 ± 12[a] | 0.90 ± 0.09 | 3.92 ± 0.30 | 0.81 ± 0.30 | 1.57 ± 0.18 | 126,000 ± 17,000 |
| *A0110* | | | | | | | *111,860 ± 15,030* |
| L15 | 71.74 | 230 ± 41[a] | 0.87 ± 0.10 | 5.07 ± 0.30 | 1.08 ± 0.40 | 1.79 ± 0.22 | 128,000 ± 28,000 |
| *A0111* | | | | | | | *114,950 ± 26,810* |
| L69[c] | 74.41 | 440 ± 59[a] | 0.83 ± 0.04 | 4.93 ± 0.25 | 1.24 ± 0.06 | 1.81 ± 0.19 | 243,000 ± 41,000 |
| *A0119* | | | | | | | |

[a] Calculated using the CAM.

[b] Calculated using the MAM.

[c] Only 4 aliquots of sample L69 were acceptable but scattered. Hence, we decided against residual dose correction and reject sample L69 as an outlier given insufficient $D_e$ data.

* Ages corrected for residual dose (24 Gy) based on tests on sample L15 are provided in italics. Water contents of 20±10% were used for all samples.

**Table 5. Tuning points obtained by dynamic time warping alignment of the Orakei RPI data to the PISO-1500 VADM (Channell et al., 2009). A standard deviation of ±1000 years is assumed for all points which equals the temporal resolution of the PISO-1500 stack.**

| Tuning point | ECD (m) | Age (years) |
|---|---|---|
| TuP93 | 46.64 | 53,400 |
| TuP175 | 51.61 | 62,000 |
| TuP193 | 52.03 | 64,600 |
| TuP297 | 57.82 | 71,200 |
| TuP343 | 59.10 | 77,200 |
| TuP384 | 60.23 | 82,800 |
| TuP434 | 61.09 | 90,000 |
| TuP464 | 61.79 | 94,800 |
| TuP529 | 63.41 | 100,600 |
| TuP589 | 65.55 | 107,000 |
| TuPex18 | 74.65 | 121,200 |
| TuPex82 | 77.24 | 129,200 |
| TuPex115 | 79.05 | 133,400 |

**Table B1. Summary of luminescence characteristics arising from postIR-IRSL$_{290}$ protocol measurements.**

| Sample | Number of aliquots accepted | Recycling ratio | Overdispersion (%) |
|---|---|---|---|
| L14 | 9/15 | 1.35 ± 0.35 | 12.4 |
| *A0110* | | | |
| L15 | 9/15 | 1.38 ± 0.69 | 52 |
| *A0111* | | | |
| L32 | 11/15 | 1.04 ± 0.09 | 67 |
| *A0115* | | | |





| L35 | 12/15 | 1.10 ± 0.08 | 24.3 |
| *A0116* | | | |
| L65 | 7/15 | 1.10 ± 0.10 | 22.7 |
| *A0118* | | | |
| L69 | 4/10 | 1.00 ± 0.10 | 24.7 |
| *A0119* | | | |

**Table B2: Results for the coarse-grained sample L20 (A0113) of different IRSL preheat temperatures.**

| IRSL preheat (°C) | No. aliquots accepted | Recycling ratio | Recuperation (%) | $D_e$ (Gy) | Total dose rate (Gy/ka) | Age (ka) |
|---|---|---|---|---|---|---|
| 50 | 2/12 | 1.03 ± 0.04 | 0.5 ± 0.1 | 439 ± 11 | 2.19 ± 0.27 | 200 ± 25 |
| 200 | 6/12 | 0.95 ± 0.18 | 1.9 ± 1.3 | 368 ± 28 | | 168 ± 24 |





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
