# Peer review of "Development of a multi-method chronology spanning the Last Glacial Interval from Orakei maar lake, Auckland, New Zealand"

_Geochronology, 2020_

## Referee Comment (RC1) · Anonymous Referee #1 · 3 Sep 2020

General comments

The manuscript presents a very detailed and thorough discussion of the multi-method chronology applied to the Orakei maar lake sequence. Each chronological method is described, and individual results presented, before integrating them using a BACON age model. The manuscript is well structured and well-written. The graphs and figures are all of exceptional quality, clearly labelled and with descriptive legends.

Specific comments

The abstract is very long (spanning two paragraphs). I would suggest to remove the discussion of the Be-10 from the abstract - better to focus on the chronological methods

that were incorporated into the final age model.

With SHCal20 now out I leave it up to the authors whether they choose to update their chronology. I would certainly encourage this, since presumably the next step will be palaeoclimate interpretations.

Discussion of reservoir corrections for radiocarbon dating is brief, and slightly conflates the 'hardwater effect' with the marine reservoir effect, which arise due to separate processes. I wouldn't have thought that there would be much of a hardwater effect as the catchment is presumably basaltic rather than carbonate?

For the tuning of the palaeomagnetic RPI curve, why were the tuning points selected randomly? It would seem better to select parts where there is more confidence in the alignment? Or, perhaps at least explain why a random approach is used for the DTW algorithm.

Is the geomagnetic excursion at ∼62 ka the Greenland-Norwegion Sea excursion? Was this considered to be used in the chronology development? It seems quite well defined in the Orakei RPI (though perhaps the trough is not clear).

Technical corrections

Line 50-52: 'Orakei maar paleolake is of unprecedented quality...' Please quantify this statement.

Line 85: 'improve temporal constraints on regional of palaeoclimatic...' Please rephrase.

Fig 6 and 7: I believe these will need to be reformatted into a portrait format.

Line 151: There is no section 3.6.1?

---

## Referee Comment (RC2) · Anonymous Referee #2 · 9 Sep 2020

Development of a multi-method chronology spanning the Last Glacial Interval from Orakei maar lake, Auckland, New Zealand

In review at Geochronology Discussions

Peti and colleagues present a multi-proxy age model for an exceptional sedimentary sequence spanning the last glacial cycle from the Auckland Volcanic Field. To develop the ago model, they integrate radiocarbon, tephra stratigraphy, luminescence dating, paleomagnetism, and cosmogenic Be. To treat their data objectively and to quantify uncertainty, they employ Dynamic Time Warping (DTW) and Bayesian Age Depth

modeling methods. Overall, the paper is well written, and the data are clearly presented. I was interested in reading more about the archive and the author's approach and perspectives on building their multi-proxy age model. Studies like this are essential for all of us that work on sedimentary sequences and the chronology will likely form the backbone for many future studies that will work on the Orakei (and other regional) maar lake. I feel this study is certainly suitable for publication in Geochronology with some revision. I am not an expert in the luminescence dating methods, and while they seem properly documented and presented in a way I can follow, hopefully another reviewer can evaluate them in more detail.

General Comments

Radiocarbon:

I feel that, while not perfect data (e.g., age reversals, unknown reservoir effects), the treatment of the radiocarbon data is fair and the authors are honest about their uncertainties. I would recommend the authors update the calibration to the SHCal20, now that it is available, and present (and make available) both the SHCal13 and SHCal20 based age models (so that other authors can make direct comparisons to either). Otherwise, the next study that presents Orakei maar lake data on age will need to re-do the age model and this age model will be dated.

Tephra Stratigraphy:

Obviously, the author's identification of the "unidentified" basaltic tephra layer T66 is central to the older part of this age model and the only real constraint beyond the RPI correlation (as the uncertainties in the luminescence data prevent those data from providing strong constraint at the temporal resolution of the final age-depth model). The authors propose that this a newly recognized tephra for the AVF, AVFaa, as it cannot be correlated to previously identified tephra layers. They use the Ar/Ar constraints from their proposed eruptive center, Mt. Albert, to assign an age to this layer. I think this assumption is reasonable, and while it is better explained in the appendix, I think it

deserves a little more attention in the main text (and perhaps the abstract) because of how important this interpretation/assumption it is to the final age model. This should maybe include the data needed to identify the tephra in the main text, as the authors do for their other tephra in Figure 3. The way the treatment of T66 is presented in the results section 4.1 makes it seem like the age of this tephra layer is well known, the eruptive history of Mt. Albert is well known, and the tephra identification has no ambiguity. This new AVFaa tephra may also be important for future studies. See below, but I also am curious if there is an RPI DTW solution that independently supports this age assignment.

I am assuming that AVF1 was not used in the age model because it has two possible ages ∼106 vs ∼83 ka. It seems like the author's age model, while not using the tephra as a constrain, is more consistent with the older of these two ages. I think it would be worthwhile to add a paragraph in the main text to discuss the AVF1 tephra, how the previously published age constraints were derived and how the new age model compares. Does the new age agree with either of the older ages? Why or why not do you think that is the case? Does it provide an addition independent support for the RPI based correlation?

Paleomagnetism:

I liked the authors use and application of DTW in their correlation of the RPI data. We all know that wiggle stratigraphic correlations can be non-unique, so while not always perfect, at least DTW is objective. However, to get a perfect DTW solution requires perfect data (which is never the case and cannot be expected in paleomagnetism). Thus, the result of the DTW solution when using a general DTW algorithm (like the one used in this study) for geologic data is often a stair-step pattern, implying sediment delivery in pulses separated by periods of no deposition. However, we often assume that sedimentary records like these accumulate gradually over time. The authors in a way deal with this by randomly sampling tuning points from the DTW solution and setting hard start/end tie points. However, this is problem that Hay et al, which the authors cite, also

address through their development of a DTW algorithm. In this algorithm, users can work with imperfect data by varying assumptions relevant to geologic data (such as how variable sediment accumulations are) to explore various possible DTW solutions that can be evaluated against independent constraints and/or expert knowledge.

Do the authors think it would be worth trying the Hay et al. DTW approach to explore other possible DTW solutions that may be more reasonable for imperfect geologic data? Why or why not? Can you treat the AVFaa tephra age independent of the RPI DTW solution and find a solution that independently supports the age the authors assign to the AVFaa tephra?

Sedimentation Rates:

It makes me nervous when I see a major change in sedimentation rates at a depth where the main chronometer for the age model changes. In the case of this study the authors find a switch from lower to higher sedimentation rates at around the same depth that the age model changes from being primarily constrained by RPI correlation to radiocarbon. I think this observation should be included in the main text. Why should I, the reader, be convinced that this accumulation rate change is the real signal and not an artifact of a non-unique or problematic RPI correlation? It doesn't appear to exactly line up with the facies unit changes or the lithologic log, but maybe there are other data that show a sedimentological change around the same time?

Data Availability:

Thanks for posting your data to Pangea. I would also recommend including the actual age-depth relationship with uncertainty as an independent contribution.

Specific Comments:

Line 263: Hay et al. aligned chemostratigraphic data, not paleomagnetic data. Their algorithm was modified to work with paleomagnetic vector data by Hagen at al. But, the Hay et al. algorithm would be the appropriate choice for RPI correlations.

Lines 495-515: There is information in this section that seems like it would fit better in the methods section, particularly the choice of DTW algorithm.

Figure 2: Would it be helpful to indicate the stratigraphic position/labels of the tephra layers?

Figure 6: It is difficult to read the small text in this figure. Please make the text larger.

Figure 7: It might help the clarity of the figure to decrease the symbol size so that it is easier to see how the age control points compare to each other.

Figures B1-B2, B4, C1-C6: All of these figures would benefit from increasing the font size of the smaller fonts to make them more legible.

---

## Short Comment (SC1) · 9 Sep 2020

[revised manuscript text omitted]

---

## Referee Comment (RC3) · Anonymous Referee #3 · 16 Sep 2020

Thank you for the opportunity to review gchron-2020-23, "Development of a multi-method chronology spanning the Last Glacial Interval from Orakei maar lake, Auckland, New Zealand"; I really enjoyed reading the manuscript. Peti et al. take on the unenvious task of pulling together radiocarbon, paleomagnetism, meteoric beryllium, luminescence, and tephrochronology with a Bayesian age-depth modelling framework, supplemented by dynamic time warping. The manuscript is very well-written, conclusions do not overextend the data, and the work underpins a continuous 130-ka record that will no doubt foster many proxy records to come. I can easily recommend the content for Geochronology.

[Figure]

I am neither a physicist nor chemist, so I cannot specifically comment on the appropriateness and accuracy of much of the methodology (e.g., paleomagnetism, luminescence). General comments about the age-depth model are followed by specific comments in the text.

General comments

Test of the model: The authors do an admirable job of stitching together the various chronological threads. However, I would like to have seen a test of the age-depth model. If this were published with a pollen record, for instance, we could see if the appearance of critical taxa corresponds with other records from the northern North Island. As is, the reliability of the reconstruction is hard to gauge. One option could be to remove a tephra, run the model, and compare the model's estimated age of the tephra to the tephra's actual age, then repeat.

Dynamic time warping: This is an interesting technique that I have not seen applied to matching proxy records. While creative, I wonder about the heavy-handedness of the warping function on the original data. The stepwise pattern in the RPI data implies the algorithm expands and compresses the record quite regularly. Further, the VADM reference curve is interpolated from a data point every 1000 yr to 200-yr resolution. All of this results in an uncertainty that is seemingly not transferred to the age-depth model. The stock +/- 1000 years does not seem realistic given the uncertainty of the Rotoehu. The authors should consider a meaningful exercise in quantifying this error. Perhaps randomly sampling 13 data points could be repeated multiple times to estimate uncertainty? From a different angle, are there RPI measurements from the top 40 m? If so, the DTW technique could be compared to the chronology established with radiocarbon and tephrochronology.

Changing sedimentation rate: I think strong caveats need to be stated when highlighting the major trends in sedimentation rate. The authors rightly point out that the changes are not strongly related to stratigraphy. However, change in sedimentation

rate is related to a change in dating technique (from RPI matching to radiocarbon and tephrochronology).

Reservoir effect: If this was a known problem, then why only have two couplets of macrofossil/tephra and bulk sediment? It is beyond the scope to resample in the current paper, but perhaps more extensive comparisons between macrofossil and bulk sediment ages would be worth investigating in a future publication.

SHCal20: Given this will be the age-depth model for many proxy records to come, along with associated inter-hemispheric comparisons, I reluctantly suggest the authors recalibrate their age-depth models with this new curve.

Specific comments

Define "high resolution"

Typically, errors are reported as 2 sigma, but here they are reported as 1 sigma. Please explain why this is the case or change to 2 sigma.

Hyphenate units and value when acting as adjective. E.g., change, "...using wireline drilling in 1 m-length sections" to "...using wireline drilling in 1-m sections".

P1L30: Change "spall" to "span"

P2L45: New Zealand does not need to be possessive

P3L85: Delete "of" before "paleoclimatic"

P3 Regional setting: Influx of erosional material is often invoked as a confounding factor throughout the manuscript. However, the catchment of Orakei is very small and crater wall slumps were presumably removed from the stratigraphy. Please explain potential sources of the erosional influx.

P7L274: Add ")" after "Accumulation model"

P8L316: Change "...as identified by (Molloy et al., 2009): to "...as identified by Molloy

et al. (2009)"

P16L648: Delete second "associated"

Figure 8: Interestingly, the age-depth model underestimates most radiocarbon dates between the Rotorua and Okareka tephras and overestimates most ages between the Okareka and Rotoehu tephras. Any thoughts on this?

---

## Author Comment (AC4) · 12 Oct 2020

Gchron-2020-23_QS

| Line number | Comment QS | Comment by authors |
|---|---|---|
| 13 | location (?) | changed accordingly |
| 14-15 | strikethrough "of associated changes" | deleted "of associated changes" |
| 14-15 | I understand what you mean, but I'm not sure about the sentence. | changed as suggested in line 15 |
| 15 | Why not something like this?
"Sediments from the Auckland Volcanic Field maar lakes preserve records of such large-scale climatic influences on regional paleoenvironment changes, as well as past volcanic eruptions." | changed accordingly as suggested |
| 17 | rapidly deposited | changed accordingly |
| 17 | high-resolution | changed accordingly |
| 19 | highlighted "combining" and "combined" | replaced the first "combining" with "using" |
| 23-24 | results suggest major influences of unaccounted catchment processes, preventing straightforward geomagnetic interpretations, | this section (on Be in the abstract) was now removed following comment (RC1-C1) |
| 25 | can you really confirm the presence of the Laschamp based only on 10Be? I'm not sure. | this section (on Be in the abstract) was now removed following comment (RC1-C1) |
| 25-27 | highlighted "We have integrated our absolute chronology with tuning of the relative paleointensity record of the Earth's magnetic field to a global reference curve (PISO-1500)." | not changed as it is unclear why this sentence was highlighted |
| 35 | strikethrough "events" | deleted "events" |
| 36-37 | Convoluted sentence. (...) uncertainties prevent understanding accurately the generation (...)? | changed to "… uncertainties prevent accurate understanding of the generation…" |
| 41 | of | changed accordingly |
| 41 | available | added "available" |
| 41 | spanning | changed accordingly |
| 42-43 | Convoluted sentence. Why not something like: "In this context, the laminated sediment sequences from maar lakes of the Auckland Volcanic Field (AVF) provide key paleoclimate records for the LGI and beyond." | changed as suggested |
| 50 | How does it change through time? | extended the sentence to "This study focuses on the lacustrine sediment sequence contained in Orakei Basin, deposited following the phreatomagmatic eruption forming the maar crater until the post-glacial sea-level rise breached the crater rim and led to the current connection between Orakei Basin and the sea (Fig. 1; Peti and Augustinus, 2019)." |

| 55-57 | In this study, we integrate absolute dating techniques (tephrochronology, radiocarbon, luminescence) and correlative dating (tuning of paleomagnetic field variations established by the relative paleointensity and meteoric 10Be) to develop an original age-depth model of the Orakei maar lake sediments. | changed as suggested |
|---|---|---|
| 55-57 | The following paragraph (or most of it) could be move in the method section. | The segments of the two following paragraphs (with respective edits) have now been moved to the beginning of the respective methods sections. |
| 62-63 | strikethrough "since it is a well-established technique for dating organic macrofossil samples younger than ca. 50,000 years (Bronk Ramsey, 2008)." | deleted "since it is a well-established technique for dating organic macrofossil samples younger than ca. 50,000 years (Bronk Ramsey, 2008)." |
| 71-72 | Do you mean by wiggle-matching of climatic records? | deleted "environmental" in "synchronous changes" given caveats of circularity when wiggle-matching climatic records |
| 72-75 | I don't think this is a good exemple to illustrate previous sentence. Geomagnetic changes are independant of environmental variations (idealy). I'm not sure to understand properly these two highlighted sentences. You could also add another reference dealing with 10Be (the one cited is about RPI) | See above comment. Removed 10Be part of this sentence to solely focus on paleomagnetic data here, and on 10Be in the following sentence. |
| 78 | Carcaillet et al do not used 10Be for dating terrestrial sediments. Simon et al. (2020 QGeo, 10.1016/j.quageo.2020.101081) do such a thing, but not by correlating with paleointensity changes (rather using 10Be radioactive decay). To my knowledge your paper and the one submitted by Lisé-Pronovost et al. (in revision in QGeo) are the first ones to try using 10Be as a relative dating tool by comparing with RPI references. Despite poor results, you could mention this here ;) | We moved this part to section 3.5 (Be methods) and removed the caveat in terrestrial settings and reference to Carcaillet et al as the same statement is repeated in this section. We added "Though the radioactive decay of 10Be has been used to date sediment much older than the Orakei sequence (e.g., Frank et al., 2008; Simon et al., 2020a), no study has been published yet applying 10Be variations in sediment cores as a relative dating tool by comparison to RPI reference data beyond the Laschamp Excursion (Nilsson et al., 2011)." to section 3.5. |
| 83 | because you most likely do not retrieved a 10Be production signal, I agree with your interpretation. | see more detailed comments in results and discussion sections |
| 83 | deposition | changed accordingly |
| 83, 84 | highlighted "robust" twice | changed second instance of "A robust independent chronology" to |

| | | |
|---|---|---|
| | | "This detailed independent chronology" |
| 85 | highlighted "significantly" | deleted "significantly" |
| 118 | strikethrough "visible", added "identified" | changed accordingly |
| 132 | strikethrough "carefully" | deleted "carefully" |
| 135-13 | strikethrough "Marine reservoir age corrections are routinely addressed in the marine realm but more difficult to assess in lake basins due to their different sizes, variable regional lithologies, depths and movement of water masses (Philippsen, 2013)." | deleted entire sentence |
| 138-139 | strikethrough "to increase the age resolution" | deleted "to increase the age resolution" |
| 150-151 | I understand it is annoying, but could you calibrate using the new SHCal20 curve? At least, look at the difference obtained between the ages after using both references. | The age model has been updated using SHCal20 now. |
| 192 | I'm not sure paleomagneticians will like this explanation, but I do understand the following argument ;) | The problem has been observed in nearby Lake Pupuke (Nilsson et al., 2011) so that we assume this to be a real problem… |
| 194-195 | strikethrough "This is a problem especially around the age of the Mono Lake Excursion, which correlates with a flare-up of the basaltic volcanoes of the AVF around 30,000 cal yr BP (Molloy et al., 2009)." | Not adjusted, this sentence is needed to explain why no paleomagnetic data is available above ca. 40,000 cal yr BP which would have been great for a comparison of 14C/tephra derived chronology and RPI DTW based chronology (see later part of comment (RC3-C2)). |
| 195 | strikethrough "Orakei maar sediment" | deleted "Orakei maar sediment" |
| 209-210 | To my opinion, this is very light and you'll need to explain a bit more why you are confident in this RPI proxy if you want to correlate it with references to help building the age model. See for instance the recent paper by Hatfield et al. (2020, Frontiers). What is your magnetic mineralogy? It is very likely that rock mag properties changes through core considering lithological and grain-size changes. It is important to discuss this since you use later RPI to build the age model. Moreover, ARM is also very dependent on grain size at constant magnetic mineralogy. | We added a reference for details to section 4.4 where the more detailed discussion of the NRM/ARM ratio is now placed. See also later comments. |
| 211 | 6 | corrected |
| 213-215 | I would rephrase. | Split into two sentences "Meteoric cosmogenic $^{10}$Be is produced in the atmosphere via nuclear reactions of cosmic ray particles with nuclei such as nitrogen and oxygen. $^{10}$Be readily attaches to aerosols and dust, and |

| | | with a short residence time of ~1 yr, is deposited on the Earth surface mainly via precipitation (Willenbring and von Blanckenburg, 2010)." |
|---|---|---|
| 217 | You could also mention here that you still need to normalized 10Be if you want to obtain records of geomagnetic field strength variations. The two cited references use different approaches, i.e. 230Thxs and flux. Most papers reconstructing past geomagnetic dipole moment from deep marine sediments use the 9Be normalisation. | adapted to "variability in (normalised) $^{10}$Be concentrations" added the "(normalised)" as the specific normalisation with 9Be is mentioned further below we chose not to expand further on this here |
| 216 | strikethrough "to a first order" | deleted "to a first order" |
| 218 | These processes essentially complicate the identification of a 10Be production signal (see also recent papers by Czymzik et al.). | adapted to "complicate $^{10}$Be provenance, delivery and accumulation and hence the identification of a $^{10}$Be production signal (e.g., Czymzik et al., 2015; Nilsson et al., 2011)." |
| 224 | es | changed accordingly |
| 224 | were | changed accordingly |
| 225 | strikethrough "down-core" | deleted "down-core" |
| 225 | strikethrough "the" | deleted "the" |
| 226 | strikethrough "in the Orakei sediment sequence, which cannot be dated with radiocarbon," | deleted "in the Orakei sediment sequence, which cannot be dated with radiocarbon," |
| 231 | measure | changed accordingly |
| 232 | using | changed accordingly |
| 240 | to increase the 10Be resolution | changed accordingly |
| 245-246 | strikethrough "Authigenic 9Be was not analysed for these ten samples and was considered negligible compared to the 9Be spike mass following measurements at UoA and ANSTO (see above)." | not changed as we consider this statement crucial to justify why 10Be/9Be is not presented for the Lund/ETH samples |
| 252 | strikethrough "Orakei maar lake" | deleted "Orakei maar lake" |
| 253-256 | I agree with this, but you might synthesis this part. | Slightly shortened to "It is crucial to avoid circularity in tuning climate proxies based on assumed synchroneity, when the presence or absence of this possible synchroneity is actually an overarching study objective (Blaauw, 2012)." |
| 256 | "relative paleointensity of the Earth magnetic field strength" is strange. Rephrase. | changed to "relative intensity of the Earth magnetic field (RPI)" |
| 257-258 | strikethrough "unlike climate signals" | deleted "unlike climate signals" |
| 259 | You could add the new study by Hatfield et al. (2020) | reference to Hatfield et al., 2020 added |
| 261-262 | strikethrough "uses generalized dynamic programming, in which a complex problem is divided into smaller problems and their solutions are stored for later use. DTW" | deleted "uses generalized dynamic programming, in which a complex problem is divided into smaller |

| | | | |
|---|---|---|---|
| | | | problems and their solutions are stored for later use. DTW" |
| 278-279 | | Identified by which proxy in your sediments? PMAG intensity or direction? 10Be? You should say that you applied the age from Lascu to the identified Laschamp interval in your sediments. | sentence extended to "the U/Th-age of the Laschamp Excursion as identified by paleomagnetic direction and intensity using the age of 41,100 ± 350 (1 σ) years BP from Lascu et al. (2016)" |
| 280 | | RPI | changed accordingly |
| 281 | | strikethrough "reference curve" | deleted "reference curve" |
| 281 | | strikethrough "with" | deleted "with" |
| 281 | | stack | added "stack" |
| 281 | | of radiocarbon ages using | added "of radiocarbon ages using" |
| 282 | | strikethrough "of radiocarbon ages" | deleted "of radiocarbon ages" |
| 282 | | strikethrough "conducted by" | deleted "conducted by" |
| 282 | | done by | added "done by" |
| 298-299 | | highlighted "and substantial thickness (>30 cm) suggest that this layer is the Rotoehu tephra." | Not clear why this was highlighted? |
| 338 | | These outliers were not incorporated in the age-model. | Not added, sentence from next comment moved here instead (slightly adapted to "Since the model recognises these outliers there was no need to remove them manually."). We like to make the difference clear between removing sample ages by the operator ("manually") vs. adding them to the Bacon input and the age model not passing through them at all and thus the model recognising them as outliers. |
| 345-346 | | strikethrough "The Bacon age model recognises all 13 outliers and hence there was no need to remove them manually." | see comment above |
| 356 | | strikethrough "The remaining six samples provided ages, and these results" | deleted "The remaining six samples provided ages, and these results" |
| 356 | | of the remaining six samples | added "of the remaining six samples" |
| 356 | | They are | changed to "They conform" |
| 374-375 | | strikethrough "of magnetic field inclination and reduced intensity of the Earth magnetic field." | deleted "of magnetic field inclination and reduced intensity of the Earth magnetic field." |
| 374 | | geomagnetic | added "geomagnetic" |
| 382 | | Some of the Figure in Appendix C should appear in the main text and be discussed more thoroughly here. This is very important to allow the use of RPI record. An easy way is to discuss if your data respect the Tauxe's criteria. Also, did you removed part of your record due to identified problematic layers? | Problematic layers were removed as part of the construction of the event corrected depth scale removing most problematic paleomagnetic data as well as samples with MAD > 15 as stated in section 3.4. |

| | | We have now moved Figure C5 (and parts of its caption) to the main text (section 4.4) and extend the text by the following discussion regarding Tauxe's criteria: "The magnetic data partially fulfils the loosely defined criteria to assess the reliability of paleointensity data from sediments (Tauxe, 1993). It appears that magnetic concentration variations exceed one order of magnitude at times and the magnetic grain size is likely not confined to a very narrow range, but all other criteria are generally fulfilled. |
|---|---|---|
| 389 | You should probably add other references presenting the Laschamp excursion from sediments or lava flows. | changed to "Laschamp Excursion (e.g., Cassata et al., 2008; Ingham et al., 2017; Laj et al., 2014; Laj and Channell, 2015; Mochizuki et al., 2006; Roperch et al., 1988) dated to 41,400 ± 350 yr by Lascu et al. (2016)." |
| 393-394 | could… | changed to "could correspond to" |
| 394 | Similarly to previous comment, they are numerous (although less numerous than for the Laschamp) papers dealing with the Blake from sediments, cite some of them. Why only referencing results from speleothems? | extended to "Blake Excursion (Smith and Foster, 1969; Thouveny et al., 2004; Tric et al., 1991; Zhu et al., 1994) dated to 116,500 ± 700 to 112,000 ± 1,900 years by Osete et al. (2012)." |
| 404 | Use the slope to calculate RPI. The slope method should give high correlation coefficients if demagnetisation steps look alike, this is good to reinforce trust on your RPI record. | We choose not to apply the slope method as we already provide the information of different demagnetisation steps which all give very similar data. Following Valet and Meynadier (1998) it is mostly not significant which approach is used. |
| 406 | e.g. | added "e.g.," |
| 408 | add also references from lava flows. Some measurements exist from nearby lava flows. See introduction in my recent paper for exemples and a discussion of such low intensity during the Laschamp (10.1016/j.epsl.2020.116547). | adapted to "the Laschamp Excursion as measured in sediments (e.g., Channell et al., 2009) as well as in lava flows from France (e.g., Laj et al., 2014; Roperch et al., 1988) and New Zealand (Cassata et al., 2008; Ingham et al., 2017; Mochizuki et al., 2006)" |
| 410 | not removed by normalization procedure then… | added ", which was not fully removed by the NRM/ARM normalisation procedure" |

| 412-413 | Is any rock mag or environmental proxy correlate with the RPI? If yes, say it and discuss. If no, say it since it strengthen your interpretation. | sentence above extended to "NRM recording in a higher energy depositional environment (compare Fig. 2) and observed in a minor anti-correlation between dry bulk density (not shown) and RPI." |
|---|---|---|
| 416 | Norwegian Greenland Sea Excursion? The RPI low corresponds to a slight shift in inclination. Is it reliable? If yes, say it and discuss. It would be the first NGS-Exc. identified in this area. | The following paragraph has been added:
"The short-duration RPI trough around 52 m aligns with a very shallow inclination of +0.4° at 51.2 m (Fig. 6). The combination of inclination, low RPI and its depth (inferring an age of ca. 61,000 yr) suggests that this may be the Norwegian-Greenland Sea Excursion (Bleil and Gard, 1989; Løvlie, 1989). This probable reversal of the geomagnetic field was considered to be restricted to high latitudes accompanied by a global low in geomagnetic field intensity and has been confirmed in various northern high-latitude sites (Channell et al., 1997; Nowaczyk et al., 1994, 2003; Nowaczyk and Baumann, 1992; Nowaczyk and Frederichs, 1999; Simon et al., 2012; Xuan et al., 2012). However, low field strength and potentially excursional directions have also been interpreted as the Norwegian-Greenland Sea Excursion in Black Sea sediments (Liu et al., 2020; Nowaczyk et al., 2013) and the Western Equatorial Pacific (Lund et al., 2017). The occurrence of the Norwegian-Greenland Sea Excursion in the Orakei maar lake record would thus constitute its first observation this far south although additional samples are needed to confirm its occurrence in the Orakei record." |
| 422 | + ref | changed to "inverse record to the relative paleointensity time-series (Elsasser et al., 1956; Ménabréaz et al., 2011)." |
| 423 | strikethrough "may have" | deleted "may have" |
| 423 | contains | added "contains" |
| 425 | strikethrough "geochemistry" | deleted "geochemistry" |

| | | |
|---|---|---|
| 426 | + ref. Please be more specific! | extended by "as $^9$Be is commonly released by weathering (Wittmann et al., 2015)." |
| 432 | Please look at fig. 3 from Simon et al., 2017 (10.1016/j.epsl.2016.11.052). In that paper, we identified two huge 9Be peaks within tephra layers. More interestingly, an other tephra layer does not bear similar large 9Be signature. Likely influenced by the nature of the eruption. In your study, there is only one 9Be peak while you have other tephra layers, why? Any idea. | Interesting. Contrary to your study we find the large peak below the position of the Rotoehu tephra layer (quite sharp base of the tephra but some cracks extend material below its base). Note that no samples were taken in the tephra layer and the actual layer itself has been excluded from the event corrected depth scale too. In this record, the Rotoehu tephra layer is clearly the thickest and from a very large eruption which may explain why the same or similar 9Be peaks have not been observed at other tephra layers. |
| 437 | Why so? Induced by very heterogenic lithologies and a sampling artefact? Normalising by 9Be would have likely reduce these deviations (if of lithological origins). | Added the sentence "The reason for this discrepancy is unclear but may be due to very heterogenic lithologies or represent a sampling/analytical artefact." |
| 447 | Bourlès et al., 1989 | reference changed to Bourlès et al., 1989 |
| 460 | The reason why the Be ratio likely does not work is because it does not respect the homogeneous mixing of both isotopes prior to scavenging. | added "as the ratio does not respect the homogeneous mixing of both isotopes prior to scavenging." |
| 462-464 | Please consider rewriting this sentence. What is enhanced? "galactic cosmic-ray production of 10Be" looks weird. | Changed to "Elevated $^{10}$Be deposition…" |
| 473 | Don't look further, this is explaining data deviation in some intervals. | see below |
| 475-476 | It seems very unlikely that you sediments could bear a 11 year solar modulation signal and not a large-scale event associated with the Laschamp. | We agree, we corrected this to "Again, we have no clear explanation of this discrepancy but it likely is due to heterogenic lithologies and/or represents a sampling/analytical artefact." |
| 479 | Does it compare favorably with records from the Pupuke Lake by Nilsson et al. (2011)? | added ", as also observed at nearby Lake Pupuke (Nilsson et al., 2011)," |
| 481 | Most importantly I think is: does your record show coherent features with available 10Be (Be ratio) records? Compare with records presented in Figs. 5 & 6 of Simon et al. (2016; 10.1002/2016JB013335). | we added "– a pattern not observed in the previous $^{10}$Be records (Simon et al., 2016)." |
| 483 | What did you expected? directional deviation or RPI low? I guess the second which presents a long duration… say it. | added "as an RPI low, hence a peak in $^{10}$Be" |

| 484 | Is it significant? It looks to me the Be ratio show the same pattern. | revised to "Two small peaks in 10Be at 73.6 m and 74.6 m may correspond to the inferred level of the Blake Excursion". As we cannot be sure whether it is significant or not, we do not use the Blake Excursion age in the age model. |
|---|---|---|
| 490 | and marine sediments (e.g. Simon et al., 2020, EPSL). | added accordingly |
| 497 | strikethrough "s" | deleted "s" |
| 497 | These similarities | "This correlation" changed to "these similarities" |
| 501 | Why not the opposite? It looks more correct to me since you don't gain anything to sample PISO at 200 year and, at the opposite, you might smooth unreliable RPI feature doing the opposite (Orakei RPI sample to 1 ka). Considering DRM it looks more correct to me. | Thank you for this observation, this indeed also improves the fit. We have updated the DTW application with the Orakei RPI smoothed to match the 1000 yr resolution of PISO and hence updated the age model as well as all related text. |
| 502 | strikethrough "between the equivalent ages" | deleted "between the equivalent ages" |
| 512 | What is the age uncertainty of PISO? | No age uncertainty is given in Channell et al., 2009. We use ±1000 years given the temporal resolution of PISO-1500. |
| 514 | Carcaillet is dealing with marine sediments, not lacustrine catchment problem. | reference deleted |
| 528 | You mentioned just above that the chronology for the lower part of the Orakei sequence is mainly guided by the "AVFAA" tephra... I hope your age model agrees with this age then. It seems very circular to me. | we clarified the above statement to "the "AVFaa" tephra provides an age for the chronology development close to the position of the possible Blake Excursion." |
| 538 | I don't get it. | This sentence refers to fig 9. The following description follows the mean line and ignores the related uncertainties presented in the figure.
We conclude that this is too confusing to state and potentially self-exploratory. |
| 551 | strikethrough "VADM" | deleted "VADM" |
| 575-577 | PISO has a resolution of 1 ka because it's a global stack, not because of measurements resolution. The huge advantage is that PISO mainly extracts a dipole variations proxy, useful for global correlation. Orakei RPI can averages a theoretical average resolution of 168 years, but this is likely smoothed by magnetisation acquisition in the sediments. | changed to "Orakei RPI record has a theoretical average resolution one measurement per 168 years although it is likely smoothed by magnetisation acquisition in the sediments" |

---

## Author Response (AR1)

We thank all three anonymous referees, Quentin Simon and Richard Staff as editor for the valuable comments which have improved the manuscript and the age model. Please find our responses here (in blue) with new page and line numbers of applied changes followed by the manuscript with tracked changes.

**Editor**

Minor comments/suggestions:

L45 (of original submission): "…at the top of…" – do you mean "…towards the north of…"?
changed to "towards the north of" P2 L41.

L55: This could be a lengthy discussion(!), but is "tephrochronology" itself an "absolute dating technique"? It is if "absolute" ages are associated with the "tephrostratigraphy"… But those ages come from other "absolute" chronological techniques don't they (e.g., 40Ar/39Ar on the volcanic material itself, or 14C or luminescence from previously dated sedimentary archives containing the tephras in question)?
We agree, this is somewhat "in between". We changed this from "absolute" to "numerical" as this should highlight the difference between the techniques we use here. P2 L53.

L58: "Taupo Volcanic Zone (TVZ)" abbreviation already introduced on L50.
Changed to "TVZ" only (note this section has been moved to the methods (3.1)). P3 L98.

L77: invert "…may which disturb…".
This half-sentence has been removed during restructuring (moving this from the introduction to section 3.5).

L82: re-word as (something like) "In comparison to this earlier model, the present study…" (for clarification).
Changed as suggested. P2 L58.

L94-95: is there a way to avoid repetition of "crater rim"?
replaced the second "crater rim" with "the tuff ring surrounding the maar lake" P2 L7.

Section 3.1: Personally, I would like the inclusion at this stage of how many tephra layers were analysed, and also a note that these were all visible (rather than crypto-) tephra layers. (This would be consistent with following sections that note, e.g., how many plant macros were collected for 14C dating.)
extended to "Tephra identification of eight visible layers" P3 L102.

L134 & 139: change "anomalous" to "anomalously".
changed accordingly. P4 L121 & 125.

L236: change "were performed" to "was "performed" (or "measurement" to "measurements").
changed to "measurements". P7 L240.

L260: is there a reason that "Virtual" is capitalised but not "…axial dipole moment"?
removed the capitalisation of Virtual. P7 L262.

L265: Presumably, "strata" (plural) should be "stratum" (singular)?
True. Changed accordingly. P7 L265.

L314: insert "…[a] similar depth…".
Changed accordingly. P9 L329.

L374: should this be "…of Earth's magnetic field"?
This part of the sentence has been removed as part of restructuring the sentence.

L408: change to "These facies unit[s]…".
changed accordingly. P12 L447.

L438: delete "a".
Deleted. P13 L491.

L564: insert "…through [the] chosen mean…"
Added accordingly. P16 L623.

L583: "…unrealistically low confidence ranges…" – this is unclear. Do you instead mean "…unrealistically precise confidence ranges…"?
Yes. As in "small" or "narrow". Changed to "precise". P16 L645.

Response to Reviewer 3 (RC3-C1): "In order to avoid inflating the current manuscript even more, we decided to use the investigation of the resulting sedimentation rates in comparison with the lithology as the test for the age model's reliability." I think that Reviewer 3 was not asking for an (e.g.) pollen record to precisely align (climatically wiggle-match) your record to others, as this would prevent the assessment of environmental synchrony/asynchrony as you say. Rather, I think that the comment was saying that if such an environmental record were also presented, then this would allow "rough" assessment of the reliability of the chronology (e.g., is the Last Interglacial approximately "in the right place"). Anyway, I fully accept that the present manuscript if focussing solely on chronology (rather than palaeoenvironment), and that no such comparison data are therefore included. However, the current wording of the sedimentological argument (L594-596: "The slowest sedimentation rate of the entire sequence is observed in an interval of very fine laminations (facies unit 10; Fig. 9), which is consistent with slow sedimentation. Fine laminations generally indicate slower accumulation rates because only small amounts of sediment are slowly deposited at the lake bottom." seems circular as written. Please could you clarify this line of argument.
Updated to "The slowest sedimentation rate of the entire sequence is observed in an interval of very fine laminations (facies unit 10; Fig. 11), which is consistent with slow sedimentation in a quiet depositional environment dominated by in-lake production of biological particles and characterised by the absence of mass movement deposits indicative of instantaneous deposition." P17 L657-660.

**Referee#1**

We thank Referee #1 for their constructive and helpful review and address the raised below. RC1 = reviewer comment from reviewer 1. C1-C9 = comments 1 to 9 followed by our response.

Specific comments

(RC1-C1) The abstract is very long (spanning two paragraphs). I would suggest to remove the discussion of the Be-10 from the abstract - better to focus on the chronological methods that were incorporated into the final age model.

Response to (RC1-C1): We have removed the Be-10 part from the abstract and reduced it to one paragraph as suggested.

(RC1-C2) With SHCal20 now out I leave it up to the authors whether they choose to update their chronology. I would certainly encourage this, since presumably the next step will be palaeoclimate interpretations.
Response to (RC1-C2): The age model has been updated with SHCal20.

(RC1-C3) Discussion of reservoir corrections for radiocarbon dating is brief, and slightly conflates the 'hardwater effect' with the marine reservoir effect, which arise due to separate processes. I wouldn't have thought that there would be much of a hardwater effect as the catchment is presumably basaltic rather than carbonate?
Response to (RC1-C3): Surprisingly we have observed higher $CaCO_3$ contents than expected in places within the sequence. We have removed the sentence on marine reservoir corrections to eliminate the confusion.

(RC1-C4) For the tuning of the palaeomagnetic RPI curve, why were the tuning points selected randomly? It would seem better to select parts where there is more confidence in the alignment? Or, perhaps at least explain why a random approach is used for the DTW algorithm.
Response to (RC1-C4): To explain this, we have added: "We chose to select these tuning points randomly (apart from the basal point) in order to prevent any bias that involved selecting points to arrive at a favoured solution" P7 L281 – P8 283.

(RC1-C5) Is the geomagnetic excursion at _62 ka the Greenland-Norwegion Sea excursion? Was this considered to be used in the chronology development? It seems quite well defined in the Orakei RPI (though perhaps the trough is not clear).
Response to (RC1-C5): Maybe, given that Quentin Simon has raised the same observation we added a whole paragraph on this possibility: "The short-duration RPI trough around 52 m aligns with a very shallow inclination of +0.4° at 51.2 m (Fig. 6). The combination of inclination, low RPI and its depth (inferring an age of ca. 61,000 yr) suggests that this may be the Norwegian-Greenland Sea Excursion (Bleil and Gard, 1989; Løvlie, 1989). This probable reversal of the geomagnetic field was considered to be restricted to high latitudes accompanied by a global low in geomagnetic field intensity and has been confirmed in various northern high-latitude sites (Channell et al., 1997; Nowaczyk et al., 1994, 2003; Nowaczyk and Baumann, 1992; Nowaczyk and Frederichs, 1999; Simon et al., 2012; Xuan et al., 2012). However, low field strength and potentially excursional directions have also been interpreted as the Norwegian-Greenland Sea Excursion in Black Sea sediments (Liu et al., 2020; Nowaczyk et al., 2013) and the Western Equatorial Pacific (Lund et al., 2017). The occurrence of the Norwegian-Greenland Sea Excursion in the Orakei maar lake record would thus constitute its first observation this far south although additional samples are needed to confirm its occurrence in the Orakei record." P12 L461-470.

Technical corrections
(RC1-C6) Line 50-52: 'Orakei maar paleolake is of unprecedented quality...' Please quantify this statement.
Response to (RC1-C6): We are not sure how this is supposed to be quantified but we updated the sentence to "The sediment record from the Orakei maar paleolake is unprecedented in its combination of length, resolution, and completeness in the context of the terrestrial south-west (SW) Pacific." P2 L48-50.

(RC1-C7) Line 85: 'improve temporal constraints on regional of palaeoclimatic...' Please

rephrase.
Response to (RC1-C7): Deleted "of", see comment (RC3-C11). P2 L61.

(RC1-C8) Fig 6 and 7: I believe these will need to be reformatted into a portrait format.
Response to (RC1-C8): Figures are reformatted (and updated) into portrait format.

(RC1-C9) Line 151: There is no section 3.6.1?
Response to (RC1-C9): Corrected to "section 3.7". P4 L136.

**Referee#2**

We thank Referee #2 for their constructive and helpful review and address the raised points below.
RC2 = reviewer comment from reviewer 2. C1-C12 = comments 1 to 12 followed by our response.

General Comments

(RC2-C1) Radiocarbon:
I feel that, while not perfect data (e.g., age reversals, unknown reservoir effects), the
treatment of the radiocarbon data is fair and the authors are honest about their uncertainties.
I would recommend the authors update the calibration to the SHCal20, now
that it is available, and present (and make available) both the SHCal13 and SHCal20
based age models (so that other authors can make direct comparisons to either). Otherwise,
the next study that presents Orakei maar lake data on age will need to re-do
the age model and this age model will be dated.
Response to (RC2-C1): The age model has now been updated to use SHCal20. The difference to the
earlier version using SHCal13 is max. 200 yr, usually less than 100 yr. No work from this record using
the earlier age model (with SHCal13) has been published yet so that we refrain from a comparison
between both age models and urge all co-workers to use the updated age model for future work.

(RC2-C2) Tephra Stratigraphy:
Obviously, the author's identification of the "unidentified" basaltic tephra layer T66 is
central to the older part of this age model and the only real constraint beyond the
RPI correlation (as the uncertainties in the luminescence data prevent those data from
providing strong constraint at the temporal resolution of the final age-depth model). The
authors propose that this a newly recognized tephra for the AVF, AVFaa, as it cannot
be correlated to previously identified tephra layers. They use the Ar/Ar constraints from
their proposed eruptive center, Mt. Albert, to assign an age to this layer. I think this
assumption is reasonable, and while it is better explained in the appendix, I think it deserves a little
more attention in the main text (and perhaps the abstract) because of
how important this interpretation/assumption it is to the final age model. This should
maybe include the data needed to identify the tephra in the main text, as the authors
do for their other tephra in Figure 3. The way the treatment of T66 is presented in
the results section 4.1 makes it seem like the age of this tephra layer is well known,
the eruptive history of Mt. Albert is well known, and the tephra identification has no
ambiguity. This new AVFaa tephra may also be important for future studies. See below,
but I also am curious if there is an RPI DTW solution that independently supports this
age assignment.

Response to (RC2-C2): We have moved the text from the appendix to a new section "4.1.2 Basaltic tephra sample T66" (P9 L346 – P10 L364) and added a new figure 4 summarising the relevant figures from the appendix (A4-A6). See comment below (RC2-C4) regarding the RPI DTW solution for AVFaa.

(RC2-C3) I am assuming that AVF1 was not used in the age model because it has two possible ages _106 vs _83 ka. It seems like the author's age model, while not using the tephra as a constrain, is more consistent with the older of these two ages. I think it would be worthwhile to add a paragraph in the main text to discuss the AVF1 tephra, how the previously published age constraints were derived and how the new age model compares. Does the new age agree with either of the older ages? Why or why not do you think that is the case? Does it provide an addition independent support for the RPI based correlation?

Response (RC2-C3): AVF1 was not used in the age model because it has not been identified via EMPA in the new Orakei 2016 cores that are mostly used in the composite stratigraphy and age model. Its depth is correlated from the core presented in Molloy et al 2009 and could be used but its investigation (along with the same question for all other tephra layers) is part of a separate study in review (with minor revision requested) in New Zealand Journal of Geology and Geophysics. Actually, the updated age model produces an age for AVF1 of ca. 90.4 ka falling somewhat between both ages. As this is discussed in the upcoming NZJGG paper we chose not to discuss it here.

(RC2-C4) Paleomagnetism:
I liked the authors use and application of DTW in their correlation of the RPI data. We all know that wiggle stratigraphic correlations can be non-unique, so while not always perfect, at least DTW is objective. However, to get a perfect DTW solution requires perfect data (which is never the case and cannot be expected in paleomagnetism). Thus, the result of the DTW solution when using a general DTW algorithm (like the one used in this study) for geologic data is often a stair-step pattern, implying sediment delivery in pulses separated by periods of no deposition. However, we often assume that sedimentary records like these accumulate gradually over time. The authors in a way deal with this by randomly sampling tuning points from the DTW solution and setting hard start/end tie points. However, this is problem that Hay et al, which the authors cite, also address through their development of a DTW algorithm. In this algorithm, users can work with imperfect data by varying assumptions relevant to geologic data (such as how variable sediment accumulations are) to explore various possible DTW solutions that can be evaluated against independent constraints and/or expert knowledge. Do the authors think it would be worth trying the Hay et al. DTW approach to explore other possible DTW solutions that may be more reasonable for imperfect geologic data? Why or why not? Can you treat the AVFaa tephra age independent of the RPI DTW solution and find a solution that independently supports the age the authors assign to the AVFaa tephra?

Response to (RC2-C4): Obtaining various possible DTW solutions and evaluating them might become an interesting exercise once we have a better understanding of the depositional environment of the sediments. We are expecting ongoing multi-proxy environmental reconstructions and high-resolution micro-facies work to shed light in this matter. A future study may find a better DTW solution but this goes beyond the scope of the current paper.
It is true that the alignment path looks very "staircase"-like which is not expected for the mode of sediment delivery (at least on the resolution we can study it here) but is necessary to allow enough stretching and compressing between the PISO and Orakei RPI to happen in the alignment. For this

reason, we do not use the alignment path itself as an age model but allow for smoothness again by integrating the tuning points into Bacon.

As for the AVFaa tephra age. We have now updated the DTW procedure (section 3.6) and removed the split at the level of the AVFaa tephra. Thus we're treating the tephra age independently and note that it agrees with the DTW solution within +/- 2 sd.

(RC2-C5) Sedimentation Rates:
It makes me nervous when I see a major change in sedimentation rates at a depth where the main chronometer for the age model changes. In the case of this study the authors find a switch from lower to higher sedimentation rates at around the same depth that the age model changes from being primarily constrained by RPI correlation to radiocarbon. I think this observation should be included in the main text. Why should I, the reader, be convinced that this accumulation rate change is the real signal and not an artifact of a non-unique or problematic RPI correlation? It doesn't appear to exactly line up with the facies unit changes or the lithologic log, but maybe there are other data that show a sedimentological change around the same time?
Response to (RC2-C5): We have added the following paragraph to section 4.8 "The stepwise increase in sedimentation rate at ~45 m nearly coincides with the change in chronometer from RPI tuning points to tephra and 14C ages. Whilst we cannot entirely disprove an influence of the chronometer change on the increase in sedimentation rate, we do note several observations that support this sedimentation rate change to be method-independent: (1) It is a stepwise change not a sudden change at the exact change point in chronometer. (2) In the interval where both chronometers overlap, albeit very short, the Rotoehu tephra and the uppermost RPI running point agree well (Fig. 10). (3) The increase in sedimentation rate does occur at the transition from facies unit 8b to 8a. These sub-facies differ in their colour contrasts between the laminations potentially indicating slightly different chemical composition, thus a slightly different depositional context which may well agree with a different sedimentation rate.  (4) Further changes in sedimentation rate, even larger in magnitude than at ~45 m occur at other positions in the sediment sequence independent of strong lithological/facies changes (and independent of chronometer changes) such as at ~39 m and within facies unit 4 (Fig. 11)." P15 L596-605.

(RC2-C6) Data Availability:
Thanks for posting your data to Pangea. I would also recommend including the actual age-depth relationship with uncertainty as an independent contribution.
Response to (RC2-C6): Thank you for this suggestion. We post the updated age-depth relationship (on a cm-resolution) as a supplementary to this publication.

Specific Comments:

(RC2-C7) Line 263: Hay et al. aligned chemostratigraphic data, not paleomagnetic data. Their algorithm was modified to work with paleomagnetic vector data by Hagen at al. But, the Hay et al. algorithm would be the appropriate choice for RPI correlations.
Response to (RC2-C7): Corrected the sentence to "Dynamic time warping (DTW) aligns time series datasets through generalized dynamic programming (Hay et al., 2019) and has been adapted for paleomagnetic vector data by Hagen et al. (2020)." P7 L263-265.

(RC2-C8) Lines 495-515: There is information in this section that seems like it would fit better in the methods section, particularly the choice of DTW algorithm.
Response to (RC2-C8): We moved most of section 4.6 into the methods section 3.6 and extended the

results section on the aligned curves to better focus on the match between RPI from Orakei and PISO-1500.

(RC2-C9) Figure 2: Would it be helpful to indicate the stratigraphic position/labels of the tephra layers?
Response to (RC2-C9): We have added the tephra layers to this figure.

(RC2-C10) Figure 6: It is difficult to read the small text in this figure. Please make the text larger.
Response to (RC2-C10): We have made the text larger (now Fig. 8).

(RC2-C11) Figure 7: It might help the clarity of the figure to decrease the symbol size so that it is easier to see how the age control points compare to each other.
Response to (RC2-C11): We have decreased the symbol size (now Fig. 9).

(RC2-C12) Figures B1-B2, B4, C4: All of these figures would benefit from increasing the font size of the smaller fonts to make them more legible.
Response to (RC2-C12): We have increased the font size in all mentioned figures.

**Referee#3**

We thank Referee #3 for their constructive and helpful review and address the raised points below. RC3 = reviewer comment from reviewer 3. C1-C16 = comments 1 to 16 followed by our response.

General comments
(RC3-C1) Test of the model: The authors do an admirable job of stitching together the various chronological threads. However, I would like to have seen a test of the age-depth model. If this were published with a pollen record, for instance, we could see if the appearance of critical taxa corresponds with other records from the northern North Island. As is, the reliability of the reconstruction is hard to gauge. One option could be to remove a tephra, run the model, and compare the model's estimated age of the tephra to the tephra's actual age, then repeat.
Response to (RC3-C1): Unfortunately, publishing a pollen record alongside the age model is beyond the scope of this paper. However, this is also somewhat circular since if we were to only trust the age model if it matches the ages inferred from expected ages of changes in proxies (i.e., MIS stages) we would fail to recognise local variability so that the inferred model age-proxy correlations are in error.
The test via removing tephra ages and comparing the modelled age to the published age is part of a paper currently in review in New Zealand Journal of Geology and Geophysics (alongside resulting new tephra ages for previously undated tephra layers) but gives a high degree of confidence in the age model. It also doesn't work for a long section of the record where no (previously dated) tephra layers were found. In order to avoid inflating the current manuscript even more, we decided to use the investigation of the resulting sedimentation rates in comparison with the lithology as the test for the age model's reliability.

(RC3-C2) Dynamic time warping: This is an interesting technique that I have not seen applied to matching proxy records. While creative, I wonder about the heavy-handedness of the warping function on the original data. The stepwise pattern in the RPI data implies the algorithm expands and compresses the record quite regularly. Further, the VADM reference curve is interpolated from a data point every 1000 yr to 200-yr resolution.

All of this results in an uncertainty that is seemingly not transferred to the age-depth model. The stock +/- 1000 years does not seem realistic given the uncertainty of the Rotoehu. The authors should consider a meaningful exercise in quantifying this error. Perhaps randomly sampling 13 data points could be repeated multiple times to estimate uncertainty? From a different angle, are there RPI measurements from the top 40 m? If so, the DTW technique could be compared to the chronology established with radiocarbon and tephrochronology.

Response to (RC3-C2): The manuscript urges the need for more realistic errors already in point 2 of section 6 (conclusions) but we expand on this in section 5.1 (as a weakness of the age model). We have updated the errors on the tuning points to reflect the match from the DTW alignment (compare new Fig. 8; Tab. 5) such that points that are matched to more than one age have a higher uncertainty spanning the range of ages that they are matched to.
Repeated sampling of the points will result in various points over the same curve but not produce different age estimates for the same depth point (which would be necessary for the uncertainty). As stated in section 3.4 there are no RPI measurements above 40 m depth as the sequence contains frequent basaltic tephra layers that obviate development of a reliable paleomagnetic signal from this section of the sequence.

(RC3-C3) Changing sedimentation rate: I think strong caveats need to be stated when highlighting the major trends in sedimentation rate. The authors rightly point out that the changes are not strongly related to stratigraphy. However, change in sedimentation rate is related to a change in dating technique (from RPI matching to radiocarbon and tephrochronology).

Response to (RC3-C3): We have added the following paragraph to section 4.8 "The stepwise increase in sedimentation rate at ~ 45 m nearly coincides with the change in chronometer from RPI tuning points to tephra and 14C ages. Whilst we cannot entirely disprove an influence of the chronometer change on the increase in sedimentation rate, we do note several observations that support this sedimentation rate change to be method-independent: (1) It is a stepwise change not a sudden change at the exact change point in chronometer. (2) In the interval where both chronometers overlap, albeit very short, the Rotoehu tephra and the uppermost RPI running point agree well (Fig. 10). (3) The increase in sedimentation rate does occur at the transition from facies unit 8b to 8a. These sub-facies differ in their colour contrasts between the laminations potentially indicating slightly different chemical composition, thus a slightly different depositional context which may well agree with a different sedimentation rate. (4) Further changes in sedimentation rate, even larger in magnitude than at ~45 m occur at other positions in the sediment sequence independent of strong lithological/facies changes (and independent of chronometer changes) such as at ~39 m and within facies unit 4 (Fig. 11)." P15 L596-605

(RC3-C4) Reservoir effect: If this was a known problem, then why only have two couplets of macrofossil/tephra and bulk sediment? It is beyond the scope to resample in the current paper, but perhaps more extensive comparisons between macrofossil and bulk sediment ages would be worth investigating in a future publication.

Response to (RC3-C4): This was, unfortunately, not known beforehand. The couplets of dates were intended to show that this is no/a minor problem but gave a larger difference than expected. More couplets however are very difficult to achieve (over the entire sequence/in representative places) as no macrofossils were found in long parts of the record within the limit of radiocarbon dating.

(RC3-C5) SHCal20: Given this will be the age-depth model for many proxy records to come, along with associated inter-hemispheric comparisons, I reluctantly suggest the authors recalibrate their age-depth models with this new curve.

Response to (RC3-C5): The age model has been updated using SHCal20.

Specific comments

(RC3-C6) Define "high resolution"
Response to (RC3-C6): (sedimentation rate above ~1m/ka) added to abstract. P1 L17.

(RC3-C7) Typically, errors are reported as 2 sigma, but here they are reported as 1 sigma. Please explain why this is the case or change to 2 sigma.
Response to (RC3-C7): Thank you for catching this, they were actually reported as a mix of both depending on how the respective literature reported them and because input values to Bacon are required to be 1 sigma. We have now changed them all to 2 sigma (or 95% confidence ranges).

(RC3-C8) Hyphenate units and value when acting as adjective. E.g., change, ": : :using wireline drilling in 1 m-length sections" to ": : :using wireline drilling in 1-m sections".
Response to (RC3-C8): Changed. P3 L90.

(RC3-C9) P1L30: Change "spall" to "span"
Response to (RC3-C9): Changed. P1 L26.

(RC3-C10) P2L45: New Zealand does not need to be possessive
Response to (RC3-C10): Removed possessive " 's ". P2 L41.

(RC3-C11) P3L85: Delete "of" before "paleoclimatic"
Response to (RC3-C11): Deleted "of". P2 L61.

(RC3-C12) P3 Regional setting: Influx of erosional material is often invoked as a confounding factor throughout the manuscript. However, the catchment of Orakei is very small and crater wall slumps were presumably removed from the stratigraphy. Please explain potential sources of the erosional influx.
Response to (RC3-C12): The crater is the catchment but undetected (small) debris flows from the crater wall were invoked as a reason for potentially problematic data whilst larger flows have been removed from the stratigraphy.

(RC3-C13) P7L274: Add ")" after "Accumulation model"
Response to (RC3-C13): Added ")"  P8 L288.

(RC3-C14) P8L316: Change ": : :as identified by (Molloy et al., 2009): to ": : :as identified by Molloy et al. (2009)"
Response to (RC3-C14): Changed as suggested. P9 L331.

(RC3-C15) P16L648: Delete second "associated"
Response to (RC3-C15): deleted first "associated" as it preserves the flow of the sentence better. P18 L716.

(RC3-C16) Figure 8: Interestingly, the age-depth model underestimates most radiocarbon dates between the Rotorua and Okareka tephras and overestimates most ages between the Okareka and Rotoehu tephras. Any thoughts on this?
Response to (RC3-C16): See discussion of the outliers in the manuscript. The interval between Rotorua and Okareka is dominated by fluvial inwash following a crater rim breach by a stream which has likely transported macrofossils of an older age into the lake basin. The younger-than-the-model

ages between the Okareka and Rotoehu tephras may have to do with small sample masses as larger macrofossils could not be found and thus smaller samples had to be used and/or with not fully captured reservoir corrections.

The Okareka age is not as well constrained as the KOT age, an adjustment of the Okareka age would potentially allow the model to include more of the older-than-the-model ages but 1) at this stage we have no indication that the used Okareka tephra age is in error and 2) it would still mean that several clearly too old outliers remain in the fluvial facies.

**Quentin Simon**

We thank Quentin Simon for his detailed comments from the attached pdf which we address in table form.

| Line number | Comment QS | Comment by authors | New page/ line |
|---|---|---|---|
| 13 | location (?) | changed accordingly | P1 L13 |
| 14-15 | strikethrough "of associated changes" | deleted "of associated changes" | P1 L14 |
| 14-15 | I understand what you mean, but I'm not sure about the sentence. | changed as suggested in line 15 | P1 L15-16 |
| 15 | Why not something like this? "Sediments from the Auckland Volcanic Field maar lakes preserve records of such large-scale climatic influences on regional paleoenvironment changes, as well as past volcanic eruptions." | changed accordingly as suggested | P1 L15-16 |
| 17 | rapidly deposited | changed accordingly | P1 L17 |
| 17 | high-resolution | changed accordingly | P1 L17 |
| 19 | highlighted "combining" and "combined" | replaced the first "combining" with "using" | P1 L19 |
| 23-24 | results suggest major influences of unaccounted catchment processes, preventing straightforward geomagnetic interpretations, | this section (on Be in the abstract) was now removed following comment (RC1-C1) | |
| 25 | can you really confirm the presence of the Laschamp based only on 10Be? I'm not sure. | this section (on Be in the abstract) was now removed following comment (RC1-C1) | |
| 25-27 | highlighted "We have integrated our absolute chronology with tuning of the relative paleointensity record of the Earth's magnetic field to a global reference curve (PISO-1500)." | not changed as it is unclear why this sentence was highlighted | P1 L22 |
| 35 | strikethrough "events" | deleted "events" | P1 L31 |
| 36-37 | Convoluted sentence. (...) uncertainties prevent | changed to "... uncertainties prevent accurate | P1 L32-33 |

| | understanding accurately the generation (...)? | understanding of the generation…" | |
|---|---|---|---|
| 41 | of | changed accordingly | P1 L37 |
| 41 | available | added "available" | P1 L37 |
| 41 | spanning | changed accordingly | P1 L37 |
| 42-43 | Convoluted sentence. Why not something like: "In this context, the laminated sediment sequences from maar lakes of the Auckland Volcanic Field (AVF) provide key paleoclimate records for the LGI and beyond." | changed as suggested | P2 L38-40 |
| 50 | How does it change through time? | extended the sentence to "This study focuses on the lacustrine sediment sequence contained in Orakei Basin, deposited following the phreatomagmatic eruption forming the maar crater until the post-glacial sea-level rise breached the crater rim and led to the current connection between Orakei Basin and the sea (Fig. 1; Peti and Augustinus, 2019)." | P2 L45-48. |
| 55-57 | In this study, we integrate absolute dating techniques (tephrochronology, radiocarbon, luminescence) and correlative dating (tuning of paleomagnetic field variations established by the relative paleointensity and meteoric 10Be) to develop an original age-depth model of the Orakei maar lake sediments. | changed as suggested | P2 L53-55 |
| 55-57 | The following paragraph (or most of it) could be move in the method section. | The segments of the two following paragraphs (with respective edits) have now been moved to the beginning of the respective methods sections. | |
| 62-63 | strikethrough "since it is a well-established technique for dating organic macrofossil samples younger than ca. 50,000 years (Bronk Ramsey, 2008)." | deleted "since it is a well-established technique for dating organic macrofossil samples younger than ca. 50,000 years (Bronk Ramsey, 2008)." | P4 L119 |
| 71-72 | Do you mean by wiggle-matching of climatic records? | deleted "environmental" in "synchronous changes" given caveats of circularity when | P5 L183 |

| | | wiggle-matching climatic records | |
|---|---|---|---|
| 72-75 | I don't think this is a good exemple to illustrate previous sentence. Geomagnetic changes are independant of environmental variations (idealy). I'm not sure to understand properly these two highlighted sentences. You could also add another reference dealing with 10Be (the one cited is about RPI) | See above comment. Removed 10Be part of this sentence to solely focus on paleomagnetic data here, and on 10Be in the following sentence. | |
| 78 | Carcaillet et al do not used 10Be for dating terrestrial sediments. Simon et al. (2020 QGeo, 10.1016/j.quageo.2020.101081) do such a thing, but not by correlating with paleointensity changes (rather using 10Be radioactive decay). To my knowledge your paper and the one submitted by Lisé-Pronovost et al. (in revision in QGeo) are the first ones to try using 10Be as a relative dating tool by comparing with RPI references. Despite poor results, you could mention this here ;) | We moved this part to section 3.5 (Be methods) and removed the caveat in terrestrial settings and reference to Carcaillet et al as the same statement is repeated in this section. We added "Though the radioactive decay of $^{10}$Be has been used to date sediment much older than the Orakei sequence (e.g., Frank et al., 2008; Simon et al., 2020a), no study has been published yet applying 10Be variations in sediment cores as a relative dating tool by comparison to RPI reference data beyond the Laschamp Excursion (Nilsson et al., 2011)." to section 3.5. | P6 L213-216 |
| 83 | because you most likely do not retrieved a 10Be production signal, I agree with your interpretation. | see more detailed comments in results and discussion sections | |
| 83 | deposition | changed accordingly | P2 L59 |
| 83, 84 | highlighted "robust" twice | changed second instance of "A robust independent chronology" to "This detailed independent chronology" | P2 L61 |
| 85 | highlighted "significantly" | deleted "significantly" | P2 L61 |
| 118 | strikethrough "visible", added "identified" | changed accordingly | P3 L95 |
| 132 | strikethrough "carefully" | deleted "carefully" | P4 L120 |
| 135-13 | strikethrough "Marine reservoir age corrections are routinely addressed | deleted entire sentence | |

| | | | |
|---|---|---|---|
| | in the marine realm but more difficult to assess in lake basins due to their different sizes, variable regional lithologies, depths and movement of water masses (Philippsen, 2013)." | | |
| 138-139 | strikethrough "to increase the age resolution" | deleted "to increase the age resolution" | P4 L124 |
| 150-151 | I understand it is annoying, but could you calibrate using the new SHCal20 curve? At least, look at the difference obtained between the ages after using both references. | The age model has been updated using SHCal20 now. | |
| 192 | I'm not sure paleomagneticians will like this explanation, but I do understand the following argument ;) | The problem has been observed in nearby Lake Pupuke (Nilsson et al., 2011) so that we assume this to be a real problem… | |
| 194-195 | strikethrough "This is a problem especially around the age of the Mono Lake Excursion, which correlates with a flare-up of the basaltic volcanoes of the AVF around 30,000 cal yr BP (Molloy et al., 2009)." | Not adjusted, this sentence is needed to explain why no paleomagnetic data is available above ca. 40,000 cal yr BP which would have been great for a comparison of 14C/tephra derived chronology and RPI DTW based chronology (see later part of comment (RC3-C2). | P5 L191 |
| 195 | strikethrough "Orakei maar sediment" | deleted "Orakei maar sediment" | P5 L192 |
| 209-210 | To my opinion, this is very light and you'll need to explain a bit more why you are confident in this RPI proxy if you want to correlate it with references to help building the age model. See for instance the recent paper by Hatfield et al. (2020, Frontiers). What is your magnetic mineralogy? It is very likely that rock mag properties changes through core considering lithological and grain-size changes. It is important to discuss this since you use later RPI to build the age model. Moreover, ARM is also very dependent on grain size at constant magnetic mineralogy. | We added a reference for details to section 4.4 where the more detailed discussion of the NRM/ARM ratio is now placed. See also later comments. | P6 L210 |

| | | | |
|---|---|---|---|
| 211 | 6 | sentence deleted | |
| 213-215 | I would rephrase. | Split into two sentences "Meteoric cosmogenic $^{10}$Be is produced in the atmosphere via nuclear reactions of cosmic ray particles with nuclei such as nitrogen and oxygen. $^{10}$Be readily attaches to aerosols and dust, and with a short residence time of ~1 yr, is deposited on the Earth surface mainly via precipitation (Willenbring and von Blanckenburg, 2010)." | P6 L216-219 |
| 217 | You could also mention here that you still need to normalized 10Be if you want to obtain records of geomagnetic field strength variations. The two cited references use different approaches, i.e. 230Thxs and flux. Most papers reconstructing past geomagnetic dipole moment from deep marine sediments use the 9Be normalisation. | adapted to "variability in (normalised) $^{10}$Be concentrations" added the "(normalised)" as the specific normalisation with 9Be is mentioned further below we chose not to expand further on this here | P6 L219 |
| 216 | strikethrough "to a first order" | deleted "to a first order" | P6 L218 |
| 218 | These processes essentially complicate the identification of a 10Be production signal (see also recent papers by Czymzik et al.). | adapted to "complicate $^{10}$Be provenance, delivery and accumulation and hence the identification of a $^{10}$Be production signal (e.g., Czymzik et al., 2015; Nilsson et al., 2011)." | P6 L221-223 |
| 224 | es | changed accordingly | P6 L228 |
| 224 | were | changed accordingly | P6 L228 |
| 225 | strikethrough "down-core" | deleted "down-core" | P6 L229 |
| 225 | strikethrough "the" | deleted "the" | P6 L229 |
| 226 | strikethrough "in the Orakei sediment sequence, which cannot be dated with radiocarbon," | deleted "in the Orakei sediment sequence, which cannot be dated with radiocarbon," | P6 L230-231 |
| 231 | measure | changed accordingly | P6 L235 |
| 232 | using | changed accordingly | P6 L236 |
| 240 | to increase the 10Be resolution | changed accordingly | P7 L244 |
| 245-246 | strikethrough "Authigenic 9Be was not analysed for these ten samples and was considered negligible compared to the 9Be spike mass following | not changed as we consider this statement crucial to justify why 10Be/9Be is not presented for the Lund/ETH samples | P7 L249-250 |

| | | | |
|---|---|---|---|
| | measurements at UoA and ANSTO (see above)." | | |
| 252 | strikethrough "Orakei maar lake" | deleted "Orakei maar lake" | P7 L255 |
| 253-256 | I agree with this, but you might synthesis this part. | Slightly shortened to "It is crucial to avoid circularity in tuning climate proxies based on assumed synchroneity, when the presence or absence of this possible synchroneity is actually an overarching study objective (Blaauw, 2012)." | P7 L256-258 |
| 256 | "relative paleointensity of the Earth magnetic field strength" is strange. Rephrase. | changed to "relative intensity of the Earth's magnetic field (RPI)" | P7 L258-259 |
| 257-258 | strikethrough "unlike climate signals" | deleted "unlike climate signals" | P7 L259 |
| 259 | You could add the new study by Hatfield et al. (2020) | reference to Hatfield et al., 2020 added | P7 L261 |
| 261-262 | strikethrough "uses generalized dynamic programming, in which a complex problem is divided into smaller problems and their solutions are stored for later use. DTW" | deleted "uses generalized dynamic programming, in which a complex problem is divided into smaller problems and their solutions are stored for later use. DTW" | P7 L263-265 |
| 278-279 | Identified by which proxy in your sediments? PMAG intensity or direction? 10Be? You should say that you applied the age from Lascu to the identified Laschamp interval in your sediments. | sentence extended to "the U/Th-age of the Laschamp Excursion as identified by paleomagnetic direction and intensity using the age of 41,100 ± 350 (2 σ) years BP from Lascu et al. (2016)" | P8 L292-293 |
| 280 | RPI | changed accordingly | P8 L295 |
| 281 | strikethrough "reference curve" | deleted "reference curve" | P8 L295 |
| 281 | strikethrough "with" | deleted "with" | P8 L296 |
| 281 | stack | added "stack" | P8 L295 |
| 281 | of radiocarbon ages using | added "of radiocarbon ages using" | P8 L296 |
| 282 | strikethrough "of radiocarbon ages" | deleted "of radiocarbon ages" | P8 L296 |
| 282 | strikethrough "conducted by" | deleted "conducted by" | P8 L296 |
| 282 | done by | added "done by" | P8 L296 |
| 298-299 | highlighted "and substantial thickness (>30 cm) suggest that this layer is the Rotoehu tephra." | Not clear why this was highlighted? | P9 L312 |
| 338 | These outliers were not incorporated in the age-model. | Not added, sentence from next comment moved here instead (slightly adapted to "Since the model recognises these outliers there was no | P10 L310 |

| | | need to remove them manually.”). We like to make the difference clear between removing sample ages by the operator (“manually”) vs. adding them to the Bacon input and the age model not passing through them at all and thus the model recognising them as outliers. | |
|---|---|---|---|
| 345-346 | strikethrough “The Bacon age model recognises all 13 outliers and hence there was no need to remove them manually.” | see comment above | P10 L310 |
| 356 | strikethrough “The remaining six samples provided ages, and these results” | deleted “The remaining six samples provided ages, and these results” | P10 L387 |
| 356 | of the remaining six samples | added “of the remaining six samples” | P10 L387 |
| 356 | They are | changed to “They conform” | P10 L387 |
| 374-375 | strikethrough “of magnetic field inclination and reduced intensity of the Earth magnetic field.” | deleted “of magnetic field inclination and reduced intensity of the Earth magnetic field.” | P11 L404 |
| 374 | geomagnetic | added “geomagnetic” | P11 L404 |
| 382 | Some of the Figure in Appendix C should appear in the main text and be discussed more thoroughly here. This is very important to allow the use of RPI record. An easy way is to discuss if your data respect the Tauxe's criteria. Also, did you removed part of your record due to identified problematic layers? | Problematic layers were removed as part of the construction of the event corrected depth scale removing most problematic paleomagnetic data as well as samples with MAD > 15 as stated in section 3.4.

We have now moved Figure C5 (and parts of its caption) to the main text (section 4.4) and extend the text by the following discussion regarding Tauxe's criteria: “The magnetic data partially fulfils the loosely defined criteria to assess the reliability of paleointensity data from sediments (Tauxe, 1993). It appears that magnetic concentration variations exceed one order of magnitude at times and the magnetic grain size is likely not confined to a very | P11 L414-417 |

| | | narrow range, but all other criteria are generally fulfilled. | |
|---|---|---|---|
| 389 | You should probably add other references presenting the Laschamp excursion from sediments or lava flows. | changed to "Laschamp Excursion (e.g., Cassata et al., 2008; Ingham et al., 2017; Laj et al., 2014; Laj and Channell, 2015; Mochizuki et al., 2006; Roperch et al., 1988) dated to 41,400 ± 350 yr by Lascu et al. (2016)." | P11 L424-426 |
| 393-394 | could… | changed to "could correspond to" | P11 L430 |
| 394 | Similarly to previous comment, they are numerous (although less numerous than for the Laschamp) papers dealing with the Blake from sediments, cite some of them. Why only referencing results from speleothems? | extended to "Blake Excursion (Smith and Foster, 1969; Thouveny et al., 2004; Tric et al., 1991; Zhu et al., 1994) dated to 116,500 ± 700 to 112,000 ± 1,900 years by Osete et al. (2012)." | P11 L430-431 |
| 404 | Use the slope to calculate RPI. The slope method should give high correlation coefficients if demagnetisation steps look alike, this is good to reinforce trust on your RPI record. | We choose not to apply the slope method as we already provide the information of different demagnetisation steps which all give very similar data. Following Valet and Meynadier (1998) it is mostly not significant which approach is used. | |
| 406 | e.g. | added "e.g.," | P12 L445 |
| 408 | add also references from lava flows. Some measurements exist from nearby lava flows. See introduction in my recent paper for exemples and a discussion of such low intensity during the Laschamp (10.1016/j.epsl.2020.116547). | adapted to "the Laschamp Excursion as measured in sediments (e.g., Channell et al., 2009) as well as in lava flows from France (e.g., Laj et al., 2014; Roperch et al., 1988) and New Zealand (Cassata et al., 2008; Ingham et al., 2017; Mochizuki et al., 2006)" | P12 L445-446 |
| 410 | not removed by normalization procedure then… | added ", which was not fully removed by the NRM/ARM normalisation procedure" | P12 L451 |
| 412-413 | Is any rock mag or environmental proxy correlate with the RPI? If yes, say it and discuss. If no, say it since it strengthen your interpretation. | sentence above extended to "NRM recording in a higher energy depositional environment (compare Fig. 2) and observed in a minor anti-correlation between dry bulk density (not shown) and RPI." | P12 L448-450 |
| 416 | Norwegian Greenland Sea Excursion? The RPI low | The following paragraph has been added: | P12 L461-470 |

| | | | |
|---|---|---|---|
| | corresponds to a slight shift in inclination. Is it reliable? If yes, say it and discuss. It would be the first NGS-Exc. identified in this area. | "The short-duration RPI trough around 52 m aligns with a very shallow inclination of +0.4° at 51.2 m (Fig. 6). The combination of inclination, low RPI and its depth (inferring an age of ca. 61,000 yr) suggests that this may be the Norwegian-Greenland Sea Excursion (Bleil and Gard, 1989; Løvlie, 1989). This probable reversal of the geomagnetic field was considered to be restricted to high latitudes accompanied by a global low in geomagnetic field intensity and has been confirmed in various northern high-latitude sites (Channell et al., 1997; Nowaczyk et al., 1994, 2003; Nowaczyk and Baumann, 1992; Nowaczyk and Frederichs, 1999; Simon et al., 2012; Xuan et al., 2012). However, low field strength and potentially excursional directions have also been interpreted as the Norwegian-Greenland Sea Excursion in Black Sea sediments (Liu et al., 2020; Nowaczyk et al., 2013) and the Western Equatorial Pacific (Lund et al., 2017). The occurrence of the Norwegian-Greenland Sea Excursion in the Orakei maar lake record would thus constitute its first observation this far south although additional samples are needed to confirm its occurrence in the Orakei record." | |
| 422 | + ref | changed to "inverse record to the relative paleointensity time-series (Elsasser et al., 1956; Ménabréaz et al., 2011)." | P12 L475-476 |
| 423 | strikethrough "may have" | deleted "may have" | P12 L476 |

| | | | |
|---|---|---|---|
| 423 | contains | added "contains" | P12 L476 |
| 425 | strikethrough "geochemistry" | deleted "geochemistry" | P12 L478 |
| 426 | + ref. Please be more specific! | extended by "as $^9$Be is commonly released by weathering (Wittmann et al., 2015)." | P12 L479-480 |
| 432 | Please look at fig. 3 from Simon et al., 2017 (10.1016/j.epsl.2016.11.052). In that paper, we identified two huge 9Be peaks within tephra layers. More interestingly, an other tephra layer does not bear similar large 9Be signature. Likely influenced by the nature of the eruption. In your study, there is only one 9Be peak while you have other tephra layers, why? Any idea. | Interesting. Contrary to your study we find the large peak below the position of the Rotoehu tephra layer (quite sharp base of the tephra but some cracks extend material below its base). Note that no samples were taken in the tephra layer and the actual layer itself has been excluded from the event corrected depth scale too. In this record, the Rotoehu tephra layer is clearly the thickest and from a very large eruption which may explain why the same or similar 9Be peaks have not been observed at other tephra layers. | |
| 437 | Why so? Induced by very heterogenic lithologies and a sampling artefact? Normalising by 9Be would have likely reduce these deviations (if of lithological origins). | Added the sentence "The reason for this discrepancy is unclear but may be due to very heterogenic lithologies or represent a sampling/analytical artefact." | P13 L491-492 |
| 447 | Bourlès et al., 1989 | reference changed to Bourlès et al., 1989 | P13 L502 |
| 460 | The reason why the Be ratio likely does not work is because it does not respect the homogeneous mixing of both isotopes prior to scavenging. | added "as the ratio does not respect the homogeneous mixing of both isotopes prior to scavenging." | P13 L514-515 |
| 462-464 | Please consider rewriting this sentence. What is enhanced? "galactic cosmic-ray production of 10Be" looks weird. | Changed to "Elevated $^{10}$Be deposition…" | P13 L518 |
| 473 | Don't look further, this is explaining data deviation in some intervals. | see below | |
| 475-476 | It seems very unlikely that you sediments could bear a 11 year solar modulation signal and not a | We agree, we corrected this to "Again, we have no clear explanation of this | P14 L530-531 |

| | | | |
|---|---|---|---|
| | large-scale event associated with the Laschamp. | discrepancy but it likely is due to heterogenic lithologies and/or represents a sampling/analytical artefact." | |
| 479 | Does it compare favorably with records from the Pupuke Lake by Nilsson et al. (2011)? | added ", as also observed at nearby Lake Pupuke (Nilsson et al., 2011)," | P14 L533-534 |
| 481 | Most importantly I think is: does your record show coherent features with available 10Be (Be ratio) records? Compare with records presented in Figs. 5 & 6 of Simon et al. (2016; 10.1002/2016JB013335). | we added "– a pattern not observed in the previous $^{10}$Be records (Simon et al., 2016)." | P14 L536-537 |
| 483 | What did you expected? directional deviation or RPI low? I guess the second which presents a long duration... say it. | added "as an RPI low, hence a peak in $^{10}$Be" | P14 L540 |
| 484 | Is it significant? It looks to me the Be ratio show the same pattern. | revised to "Two small peaks in 10Be at 73.6 m and 74.6 m may correspond to the inferred level of the Blake Excursion". As we cannot be sure whether it is significant or not, we do not use the Blake Excursion age in the age model. | P14 L540-541 |
| 490 | and marine sediments (e.g. Simon et al., 2020, EPSL). | added accordingly | P14 L548-549 |
| 497 | strikethrough "s" | deleted "s" | P7 L274 |
| 497 | These similarities | "This correlation" changed to "these similarities" | P7 L274 |
| 501 | Why not the opposite? It looks more correct to me since you don't gain anything to sample PISO at 200 year and, at the opposite, you might smooth unreliable RPI feature doing the opposite (Orakei RPI sample to 1 ka). Considering DRM it looks more correct to me. | Thank you for this observation, this indeed also improves the fit. We have updated the DTW application with the Orakei RPI smoothed to match the 1000 yr resolution of PISO and hence updated the age model as well as all related text. | |
| 502 | strikethrough "between the equivalent ages" | deleted "between the equivalent ages" | |
| 512 | What is the age uncertainty of PISO? | No age uncertainty is given in Channell et al., 2009. We use ±1000 years given the temporal resolution of PISO-1500. | P7 L276 |

| 514 | Carcaillet is dealing with marine sediments, not lacustrine catchment problem. | reference deleted | |
|------|------|------|------|
| 528 | You mentioned just above that the chronology for the lower part of the Orakei sequence is mainly guided by the "AVFAA" tephra... I hope your age model agrees with this age then. It seems very circular to me. | we clarified the above statement to "the "AVFaa" tephra provides an age for the chronology development close to the position of the possible Blake Excursion." | P15 L565 |
| 538 | I don't get it. | This sentence refers to fig 9. The following description follows the mean line and ignores the related uncertainties presented in the figure.
We conclude that this is too confusing to state and potentially self-exploratory so we deleted this. | |
| 552 | strikethrough "VADM" | deleted "VADM" | P16 L611 |
| 575-577 | PISO has a resolution of 1 ka because it's a global stack, not because of measurements resolution. The huge advantage is that PISO mainly extracts a dipole variations proxy, useful for global correlation. Orakei RPI can averages a theoretical average resolution of 168 years, but this 
[revised manuscript text omitted]

1000

[Figure]

**Lithology**
from Peti and Augustinus (2019)

**Facies description**

Event corrected depth (ECD, m) below marine/freshwater transition

| | |
|---|---|
| 1 a b c b a | Peat |
| 2 c | Light brown massive clay with bioturbation |
| 3 a b c | Banded sand/silt with wood fragments |
| 4 | Light brown clay with fine laminations |
| 5 | Greyish fine clay with light wavy laminations |
| 6 | Massive light brown clay with basaltic tephra |
| 7 | Reddish brown clay with fine laminations and abundant basaltic tephra |
| 8 a b | Dark brown fine clay with fine laminations and thin mass movement deposits |
| 9 a b c | Massive to banded silt to sand/silt |
| 10 | Fine brown with lighter laminations frequent thin mass movement deposits |
| 11 a b c d | Silt with coarse sand layers and big organics fragments (wood and compressed leafs, twigs) |
| 12 a b | Finely laminated dark brown clay, thin sand bands, abundant mass movement deposits |
| 13 a b | Finely laminated very dark brown clay, high frequency laminations, abundant (thick) mass movement deposits |
| 14 a b | Massive light brown clay (layered basaltic ash) |

Legend:
- Peat/Organics
- Greenish beige laminated clay
- Dark brown lacustrine silt with sand bands
- Finely laminated dark brown lacustrine clay
- Finely laminated reddish brown clay with basaltic tephra
- Finely laminated very dark brown lacustrine clay
- Light brown laminated lacustrine clay
- Beige silt with sandy bands/layers
- Dark brown lacustrine silt
- Massive beige/brown silt to clay
- Massive clay
- Sand
- Basaltic tephra
- Rhyolitic tephra
- Bioturbation

[revised manuscript text omitted]

| OZX882 | Bulk sed. | 28.39 | 6 (Clay) | 34,650 ± 34580 | 1026 ± 117? | 38, 615 ± 836 |
| OZX873 | Bulk sed. | 29.78 | 7 (Clay) | 26,920 ± 38190 | 410 ± 170? | 30,611 ± 478 |
| OZX340 | Wood | 31.44 | 8a (Clay) | 18,210 ± 28140 | | 22, 107 ± 340 |
| OZX874 | Bulk sed. | 31.55 | 8a (Clay) | 24,870 ± 30150 | 410 ± 170 | 28, 666 ± 484 |
| OZX875 | Bulk sed. | 32.53 | 8a (Clay) | 25,180 ± 28140 | 410 ± 170 | 29, 003 ± 255 |
| OZX876 | Bulk sed. | 33.44 | 8a (Clay) | 25,670 ± 28140 | 410 ± 170 | 29, 481 ± 476 |
| OZX342 | Wood | 33.83 | 8a (Clay) | 25,560 ± 19380 | | 29, 741 ± 478 |
| OZX877 | Bulk sed. | 35.54 | 8a (Clay) | 26,820 ± 32160 | 410 ± 170 | 30, 538 ± 464 |
| OZX344 | Bulk sed. | 36.50 | 8a (Clay) | 29,260 ± 24120 | 410 ± 170 | 33, 343 ± 540 |
| OZX343 | Wood | 37.63 | 8a (Clay) | 25,580 ± 38190 | | 29, 762 ± 480 |
| OZX341 | Wood | 39.20 | 8a (Clay) | 30,360 ± 28140 | | 34, 738 ± 392 |
| OZX347 | Bulk sed. | 39.20 | 8a (Clay) | 30,770 ± 32160 | 410 ± 170 | 34, 710 ± 502 |

[revised manuscript text omitted]